# Non-Asymptotic Guarantees for Average-Reward Q-Learning with Adaptive Stepsizes

**Zaiwei Chen**
School of Industrial Engineering, Purdue University
chen5252@purdue.edu

## Abstract

This work presents the first finite-time analysis of average-reward $Q$-learning with an asynchronous implementation. A key feature of the algorithm we study is the use of adaptive stepsizes that act as local clocks for each state-action pair. We show that the mean-square error of this $Q$-learning algorithm, measured in the span seminorm, converges at a rate of $\tilde{\mathcal{O}}(1/k)$. To establish this result, we demonstrate that adaptive stepsizes are necessary: without them, the algorithm fails to converge to the correct target. Moreover, adaptive stepsizes can be viewed as a form of implicit importance sampling that counteracts the effect of asynchronous updates. Technically, the use of adaptive stepsizes causes each $Q$-learning update to depend on the full sample history, introducing strong correlations and making the algorithm a non-Markovian stochastic approximation (SA) scheme. Our approach to overcoming this challenge involves (1) a time-inhomogeneous Markovian reformulation of non-Markovian SA, and (2) a combination of almost-sure time-varying bounds, conditioning arguments, and Markov chain concentration inequalities to break the strong correlations between the adaptive stepsizes and the iterates.

## 1   Introduction

Reinforcement Learning (RL) has become as a powerful framework for solving sequential decision-making problems, as demonstrated by its growing impact across a range of real-world applications, including autonomous robotics [64], game-playing AI [63], and the development of large language models [15]. Given the promising potential of RL, establishing strong theoretical foundations to guide its practical implementation is of significant importance.

An RL problem is typically modeled as a Markov decision process (MDP) [67], but its objective can vary by application. Finite-horizon RL maximizes cumulative rewards over a fixed time, while infinite-horizon discounted RL introduces a discount factor $\gamma \in (0, 1)$ to prioritize immediate rewards over future rewards. However, selecting an appropriate discount factor can be challenging. For instance, it is often unclear how far into the future decisions should account for rewards. Moreover, in high-frequency decision-making tasks such as robotic control, queuing systems, and financial trading, introducing a discount factor may not be appropriate. The infinite-horizon average-reward setting addresses these challenges by optimizing the long-run average reward without requiring a discount factor. However, the absence of discounting introduces unique challenges for both algorithm design and theoretical analysis. For example, the associated Bellman operator is no longer a norm-contractive mapping [58], sample-based updates add additional complexities to the structure of the algorithm [1], and optimality depends on value differences rather than absolute values.

These challenges are particularly evident in $Q$-learning [75], one of the most classical and practically impactful RL algorithms. Due to its popularity and its role as a major milestone in RL [53], substantial efforts have been dedicated to providing theoretical guarantees, especially in terms of convergence rates, for $Q$-learning. In the discounted setting, the first finite-time analysis of $Q$-learning was

conducted in the early 2000s [28], followed by a series of works over the past two decades that eventually led to matching upper and lower bounds [4, 45]. In contrast, in the average-reward setting, due to the aforementioned challenges, existing results are largely limited to asymptotic convergence [1, 37, 66, 73, 74, 77, 78] and regret analysis [3, 34, 76, 83]. Regarding finite-time analysis, even for $Q$-learning with synchronous updates (which requires a generative model for i.i.d. sampling), results have only appeared very recently [14, 35, 80]. To the best of our knowledge, no existing work provides a finite-time analysis for the last-iterate convergence of $Q$-learning with asynchronous updates based on a single trajectory of Markovian samples.

*In this work, we provide the first principled study on the finite-time analysis of average-reward $Q$-learning. Specifically, we make the following contributions.*

**Finite-Time Analysis for the Last-Iterate Convergence.** We study a natural variant of average-reward $Q$-learning (cf. Algorithm 1). The only difference compared to its discounted counterpart (aside from setting the discount factor to one) is the use of stepsizes $\alpha_k(s, a)$ that adapt to individual state-action pairs $(s, a)$. Specifically, the adaptive stepsize $\alpha_k(s, a)$ is inversely proportional to the number of visits to the state-action pair $(s, a)$, thereby serving as a local clock for each pair. We establish finite-time convergence bounds for this $Q$-learning algorithm, showing that $\mathbb{E}[p_{\text{span}}(Q_k - Q^*)^2]$ (where $p_{\text{span}}(\cdot)$ denotes the span seminorm, $Q_k$ is the $k$-th iterate, and $Q^*$ is the optimal relative $Q$-function) converges at a rate of $\tilde{\mathcal{O}}(1/k)$. To the best of our knowledge, this result provides the first finite-time analysis for the last-iterate convergence of average-reward $Q$-learning.

**The Necessity of Using Adaptive Stepsizes.** Since the use of adaptive stepsizes is the only major difference compared to the existing finite-time analysis of discounted $Q$-learning,[1] we conduct a thorough investigation into both the necessity and the fundamental reasons for using adaptive stepsizes in average-reward $Q$-learning. Specifically, we show that if universal stepsizes are used (e.g., $\alpha_k(s, a) = 1/k$ for all $(s, a)$), the algorithm is, in general, guaranteed *not* to converge in $p_{\text{span}}(\cdot)$ to $Q^*$ (cf. Proposition 4.1). We then identify that the use of adaptive stepsizes can be interpreted as a form of importance sampling that counteracts the effect of asynchronous updates based on a single sample trajectory, with the importance sampling ratios estimated using empirical frequencies derived from the historical number of visits to each state-action pair.

**Technical Contributions.** The use of adaptive stepsizes in $Q$-learning causes each update to depend on the entire history of visited state-action pairs, introducing strong correlations and making the algorithm a non-Markovian stochastic approximation (SA). This poses significant challenges for the finite-time analysis, which have not been addressed in the existing literature. To overcome this challenge, we first reformulate the algorithm as a Markovian SA by incorporating the empirical frequencies of visited state-action pairs into the stochastic process that drives the algorithm. However, the resulting Markovian noise is time-inhomogeneous and does not exhibit the geometric mixing property typically used to handle correlations between the iterates and the noise. To resolve this issue, we develop an approach that begins by showing that the $Q$-learning iterates, while not uniformly bounded by a constant (in stark contrast to the discounted setting [29]), nonetheless satisfy an almost-sure, time-varying bound that grows at most logarithmically with $k$. We then leverage this result, along with conditioning arguments and Markov chain concentration inequalities, to break the correlation. See Section 5 for a more detailed discussion of our proof techniques.

## 1.1 Related Literature

In this section, we discuss related literature on average-reward Q-learning, discounted $Q$-learning, and Markovian SA.

**Average-Reward Q-Learning.** The first provably convergent algorithms for average-reward $Q$-learning are the relative value iteration (RVI) $Q$-learning and stochastic shortest path (SSP) $Q$-learning algorithms [1]. Since then, many variants have been proposed, including differential $Q$-learning [72, 73], dynamic horizon $Q$-learning [35], among others. The asymptotic convergence of RVI $Q$-learning and its variants was established in [1] by leveraging results from general asynchronous SA [11], and was later extended under relaxed assumptions [2, 74, 78] and to MDPs with continuous state spaces [37, 77]. In contrast, non-asymptotic results are much more limited. Specifically,

---

[1]Although adaptive stepsizes have been used in the existing asymptotic analysis of discounted $Q$-learning [1, 68], they have been less explored in the context of finite-time analysis.

even for $Q$-learning with a synchronous implementation (which requires a generative model for i.i.d. sampling), finite-time analysis has not been conducted until very recently [24, 35, 80]. A related but distinct line of work designs online $Q$-learning-based algorithms that aim to balance the exploration–exploitation trade-off, with performance measured in terms of regret; see [3, 76, 83] and the references therein. However, since the performance metrics are fundamentally different (regret versus last-iterate convergence in the span seminorm), these results are not directly comparable.

**Discounted $Q$-Learning.** The celebrated $Q$-learning algorithm for discounted MDPs was first proposed in [75], and its almost sure convergence was established through various approaches in [13, 33, 68]. The first finite-time analysis of $Q$-learning (for both synchronous and asynchronous implementations) was performed in [28]. Since then, a sequence of works has aimed to provide refined characterizations of its convergence rate, including mean-square bounds [5, 6, 20, 21, 42, 70] and high-probability concentration bounds [44–46, 60, 71]. Variants of $Q$-learning, including Zap $Q$-learning, $Q$-learning with Polyak averaging, and $Q$-learning with Richardson - Romberg extrapolation, have been proposed and studied in [25], [48], and [82], respectively. To overcome the curse of dimensionality, $Q$-learning is often implemented with function approximation in practice. A rich body of literature provides theoretical guarantees for $Q$-learning with function approximation [19, 23, 49–51, 79]. Compared with $Q$-learning in the discounted setting, the finite-time analysis of average-reward $Q$-learning is significantly more challenging due to the following reasons: (1) the lack of discounting makes the Bellman operator non-contractive in any norm, (2) the iterates are not uniformly bounded, and (3) the use of adaptive stepsizes (which is necessary, as will be illustrated in Section 4) introduces strong correlations and makes the algorithm a non-Markovian SA. As a result, none of the existing approaches for discounted $Q$-learning are applicable here.

**Stochastic Approximation.** At a high level, $Q$-learning can be viewed as iteratively solving the Bellman equation, a fixed-point equation, using the SA method [61]. The asymptotic convergence of SA for solving fixed-point equations has been established under fairly general assumptions in [7, 12, 13, 39, 41], among many others. For finite-time analysis, the properties of the fixed-point operator and the nature of the noise sequence play crucial roles. In particular, when the operator is linear or contractive with respect to some norm, and the noise process is either i.i.d. or forms a uniformly ergodic Markov chain, there is a rich body of literature establishing mean-square and high-probability bounds [9, 21, 22, 26, 32, 54, 55, 59, 60, 65, 70]. Beyond these standard settings, finite-time analysis of SA with seminorm contractive operators and non-expansive operators have been developed recently in [24] and [10, 14], respectively. A main feature of the average-reward $Q$-learning algorithms studied in this work is that, as SA algorithms, they are inherently non-Markovian due to the use of adaptive stepsizes that depend on the entire history of visited state-action pairs. Through a novel reformulation, we cast the algorithm as a Markovian SA; however, the resulting Markovian noise is time-inhomogeneous and, more importantly, does not exhibit geometric mixing, or even mixing faster than $1/k$. This presents a unique challenge that has not been addressed in the existing finite-time analysis of SA.

## 2 From Average-Reward RL to the Seminorm Bellman Equation

In this section, we first provide background on average-reward RL and then introduce the seminorm Bellman equation, which plays a central role in motivating both the algorithm design and the analysis of $Q$-learning in subsequent sections.

### 2.1 Background on Average-Reward RL

Consider an infinite-horizon, undiscounted MDP defined by the tuple $(\mathcal{S}, \mathcal{A}, \mathcal{P}, \mathcal{R})$ [58], where $\mathcal{S}$ and $\mathcal{A}$ denote the finite state and action spaces, respectively. The transition dynamics are given by $\mathcal{P} = \{p(s'|s, a)\}_{(s,a,s') \in \mathcal{S} \times \mathcal{A} \times \mathcal{S}}$, where $p(s'|s, a)$ denotes the probability of transitioning to state $s'$ after taking action $a$ in state $s$. The stage-wise reward function is denoted by $\mathcal{R} : \mathcal{S} \times \mathcal{A} \to [-1, 1]$. The transition probabilities and the reward function are unknown to the agent, but the agent can interact with the environment by taking actions, observing transitions, and receiving rewards.

Given a policy $\pi : \mathcal{S} \to \Delta(\mathcal{A})$ (where $\Delta(\mathcal{A})$ denotes the set of probability distributions supported on $\mathcal{A}$), the average reward $r^\pi \in \mathbb{R}^{|\mathcal{S}|}$ is defined as $r^\pi(s) = \lim_{K \to \infty} \mathbb{E}_\pi[\sum_{k=1}^K \mathcal{R}(S_k, A_k)/K \mid S_1 = s]$

for all $s \in \mathcal{S}$, where the expectation $\mathbb{E}_\pi[\,\cdot\,]$ is taken over the randomness in the trajectory generated by the policy $\pi$. It is shown in standard MDP theory [58] that if the Markov chain induced by the policy $\pi$ has a single recurrent class, the limit exists and is independent of the initial state. In this work, we operate under this setting. Consequently, we slightly abuse notation and use $r^\pi$ to denote the scalar average reward associated with policy $\pi$. The goal is to find an optimal policy $\pi^*$ that maximizes the average reward. Define the optimal relative $Q$-function (also known as the $Q$-value bias function) $Q^* \in \mathbb{R}^{|\mathcal{S}||\mathcal{A}|}$ as $Q^*(s,a) = \mathbb{E}_{\pi^*}[\sum_{k=1}^\infty (\mathcal{R}(S_k, A_k) - r^*) \mid S_1 = s, A_1 = a]$ for all $(s,a) \in \mathcal{S} \times \mathcal{A}$, where $r^* \in \mathbb{R}$ is the optimal average reward. It is known that any policy $\pi$ that satisfies $\pi(s) \in \arg\max_{a \in \mathcal{A}} Q^*(s,a)$ for all state $s \in \mathcal{S}$ is an optimal policy [58]. Therefore, the problem reduces to finding the optimal relative $Q$-function $Q^*$.

To find $Q^*$, we next introduce the Bellman equation. Let $\mathcal{H} : \mathbb{R}^{|\mathcal{S}||\mathcal{A}|} \to \mathbb{R}^{|\mathcal{S}||\mathcal{A}|}$ denote the Bellman operator defined as $[\mathcal{H}(Q)](s,a) = \mathcal{R}(s,a) + \mathbb{E}[\max_{a' \in \mathcal{A}} Q(S_2, a') \mid S_1 = s, A_1 = a]$ for all $(s,a)$ and $Q \in \mathbb{R}^{|\mathcal{S}||\mathcal{A}|}$. The Bellman equation is then given by

$$\mathcal{H}(Q) - Q = r^* e, \tag{1}$$

where $e$ denotes the all-ones vector. It is well known that $Q^*$ is a solution to Eq. (1) [58]. However, such a solution is not unique. To see this, observe that for any $c \in \mathbb{R}$, it follows directly from the definition of $\mathcal{H}(\cdot)$ that $\mathcal{H}(Q^* + ce) - (Q^* + ce) = \mathcal{H}(Q^*) + ce - Q^* - ce = \mathcal{H}(Q^*) - Q^* = r^* e$, implying that $Q^* + ce$ is also a solution to Eq. (1). Fortunately, for the purpose of finding an optimal policy, errors in the direction of the all-ones vector have no impact on the result, because $\pi^*(s) \in \arg\max_{a \in \mathcal{A}} Q^*(s,a) = \arg\max_{a \in \mathcal{A}} (Q^*(s,a) + c)$ for all $c \in \mathbb{R}$. Therefore, it suffices to find any point in the set $\mathcal{Q} := \{Q^* + ce \mid c \in \mathbb{R}\}$.

## 2.2 The Seminorm Bellman Equation

Due to the lack of discounting, the Bellman operator $\mathcal{H}(\cdot)$ is not a contraction mapping under any norm. However, it has been shown in [58] that the operator $\mathcal{H}(\cdot)$ can be a contraction mapping with respect to the span seminorm under certain additional assumptions on the underlying stochastic model (to be discussed shortly). Since this property forms the foundation of the $Q$-learning algorithm we will present, we begin by formally introducing the span seminorm and discussing its properties.

For any $x \in \mathbb{R}^d$, the span seminorm of $x$, denoted by $p_{\text{span}}(x)$, is defined as $p_{\text{span}}(x) = (\max_i x_i - \min_j x_j)/2$. Similar to a norm, the span seminorm is non-negative and satisfies the triangle inequality: $p_{\text{span}}(x + y) \leq p_{\text{span}}(x) + p_{\text{span}}(y)$ for all $x, y \in \mathbb{R}^d$; and absolute homogeneity: $p_{\text{span}}(\alpha x) = |\alpha| p_{\text{span}}(x)$ for all $\alpha \in \mathbb{R}$ and $x \in \mathbb{R}^d$ [57, Section 6.6.1]. However, unlike a norm, $p_{\text{span}}(x) = 0$ does not imply $x = 0$. In fact, the set $\{x \in \mathbb{R}^d \mid p_{\text{span}}(x) = 0\}$ is called the kernel of the span seminorm and is denoted by $\ker(p_{\text{span}})$. Since $p_{\text{span}}(x) = 0$ if and only if $\max_i x_i = \min_j x_j$, in which case all entries of $x$ must be identical, we have $\ker(p_{\text{span}}) = \{ce \mid c \in \mathbb{R}\}$. Another important property of the span seminorm is that $p_{\text{span}}(x)$ can be interpreted as the distance from $x$ to the linear subspace $\{ce \mid c \in \mathbb{R}\}$ with respect to $\|\cdot\|_\infty$. Since this result will be used frequently throughout the paper, we formally state it in the following lemma. The proof is provided in Appendix C.1.

**Lemma 2.1.** *For any $x \in \mathbb{R}^d$, we have $\arg\min_{c \in \mathbb{R}} \|x - ce\|_\infty = (\max_i x_i + \min_j x_j)/2$. As a result, the span seminorm of $x$ can be equivalently written as $p_{span}(x) = \min_{c \in \mathbb{R}} \|x - ce\|_\infty$.*

With $p_{\text{span}}(\cdot)$ properly introduced, we next state our assumption on the Bellman operator $\mathcal{H}(\cdot)$.

**Assumption 2.1.** The Bellman operator $\mathcal{H}(\cdot)$ is a $\beta$-contraction mapping with respect to $p_{\text{span}}(\cdot)$.

A sufficient condition for Assumption 2.1 to hold is $\max_{(s,a),(s',a')} \|p(\cdot \mid s,a) - p(\cdot \mid s',a')\|_{\text{TV}} < 1$, where $\|\cdot\|_{\text{TV}}$ denotes the total variation distance. In this case, Assumption 2.1 is satisfied with $\beta = \max_{(s,a),(s',a')} \|p(\cdot \mid s,a) - p(\cdot \mid s',a')\|_{\text{TV}}$. This condition is adopted from standard MDP theory textbooks [58], where it was used to study the convergence of relative value iteration [57, Proposition 6.6.1 and Theorem 6.6.2]. Since the goal of this work is to establish the convergence rate for $Q$-learning in the span seminorm contraction setting, we do not attempt to relax Assumption 2.1. Nevertheless, we include a discussion in Section 6 on potential approaches for analyzing $Q$-learning when Assumption 2.1 is not satisfied.

Under Assumption 2.1, we have the following result, whose proof is presented in Appendix C.2.

**Lemma 2.2.** $\{Q^* + ce \mid c \in \mathbb{R}\} = \{Q \mid \mathcal{H}(Q) - Q = r^* e\} = \{Q \mid p_{span}(\mathcal{H}(Q) - Q) = 0\}$.

As a result of Lemma 2.2, the seminorm fixed-point equation

$$p_{\text{span}}(\mathcal{H}(Q) - Q) = 0 \tag{2}$$

and the Bellman equation (1) are equivalent in the sense that they have the same set of solutions: $\mathcal{Q} = \{Q^* + ce \mid c \in \mathbb{R}\}$. Therefore, it suffices to find a solution to Eq. (2) in order to compute an optimal policy. For this reason, we shall refer to Eq. (2) as the seminorm Bellman equation, or simply the Bellman equation.

## 3 Main Results

To solve the Bellman equation (2), we next introduce a $Q$-learning algorithm with finite-time convergence guarantees. Recall that in the RL setting, the parameters of the stochastic model (i.e., the transition dynamics and the reward function) are unknown to the agent, but the agent can interact with the environment by taking actions, receiving rewards, and observing environment transitions.

### 3.1 Algorithm

Let $\pi$ denote the policy used by the agent to interact with the environment, commonly referred to as the behavior policy. Our $Q$-learning algorithm is represented in Algorithm 1.

---
**Algorithm 1** $Q$-Learning
---
1: **Input:** Initializations $Q_1 = 0 \in \mathbb{R}^{|\mathcal{S}||\mathcal{A}|}$, $S_1 \in \mathcal{S}$, and a behavior policy $\pi$.
2: **for** $k = 1, 2, \cdots$, **do**
3:      Take $A_k \sim \pi(\cdot \mid S_k)$, observe $S_{k+1} \sim p(\cdot \mid S_k, A_k)$, and receive $\mathcal{R}(S_k, A_k)$.
4:      Compute the temporal difference: $\delta_k = R(S_k, A_k) + \max_{a'} Q_k(S_{k+1}, a') - Q_k(S_k, A_k)$.
5:      Update the $Q$-function: $Q_{k+1}(s, a) = Q_k(s, a) + \alpha_k(s, a)\mathbb{1}_{\{(s,a)=(S_k,A_k)\}}\delta_k$ for all $(s, a)$,
       where $\alpha_k(s, a) = \alpha/(N_k(s, a) + h)$, and $N_k(s, a) := \sum_{i=1}^{k} \mathbb{1}_{\{(S_i, A_i)=(s,a)\}}$ denotes the total
       number of visits to the state-action pair $(s, a)$ up to the $k$-th iteration.
6: **end for**
7: **Output:** $\{Q_k\}_{k \geq 1}$

---

Note that Algorithm 1 is surprisingly simple and represents the most natural extension of $Q$-learning in the discounted setting [75]. In fact, the only modification (aside from setting the discount factor to one) is using adaptive stepsizes of the form $\alpha_k(s, a) = \alpha/(N_k(s, a) + h)$, where $\alpha, h > 0$ are tunable parameters. Importantly, the stepsize $\alpha_k(s, a)$ depends on the specific state-action pair through the counter $N_k(s, a)$ and is therefore not universal. Throughout, we refer to such stepsizes as *adaptive stepsizes*. Although adaptive stepsizes of this form have been used in the existing asymptotic analysis of $Q$-learning and temporal-difference learning in both the discounted and average-reward settings [1, 68], they have been less explored in the context of finite-time analysis. The necessity and theoretical motivation behind this choice will be discussed in detail in Section 4.

In the existing literature, the algorithm most closely related to Algorithm 1 is RVI $Q$-learning [1]. In RVI $Q$-learning, the temporal difference is defined as $\delta_k = R(S_k, A_k) + \max_{a'} Q_k(S_{k+1}, a') - Q_k(S_k, A_k) - f(Q_k)$, where $f(\cdot)$ is a Lipschitz function satisfying $f(e) = 1$ and $f(Q + ce) = f(Q) + c$ for any $c \in \mathbb{R}$. Subtracting the additional term $f(Q_k)$ in the temporal difference ensures almost sure convergence to a particular solution $Q$ of Eq. (1) that satisfies $f(Q) = r^*$ [1]. In contrast, Algorithm 1 does not include such a step, as we will show that it guarantees convergence to $Q^*$ in $p_{\text{span}}(\cdot)$, or equivalently, convergence to the set $\mathcal{Q} = \{Q^* + ce \mid c \in \mathbb{R}\}$ with respect to $\|\cdot\|_\infty$. From a theoretical standpoint, such a set convergence is sufficient for computing an optimal policy.

### 3.2 Finite-Time Analysis

To present our main theoretical result, we first state our assumption regarding the behavior policy.

**Assumption 3.1.** The behavior policy satisfies $\pi(a|s) > 0$ for all $(s, a)$, and the Markov chain $\{S_k\}$ induced by the behavior policy $\pi$ is irreducible and aperiodic.

Assumption 3.1 is standard in the existing studies of both value-based and policy-based RL algorithms [46, 47, 65, 68, 69], and guarantees that all state-action pairs are visited infinitely often during learning.

Specifically, under Assumption 3.1, the Markov chain $\{S_k\}$ induced by $\pi$ has a unique stationary distribution, denoted by $\mu \in \Delta(\mathcal{S})$, which satisfies $\min_{s \in \mathcal{S}} \mu(s) > 0$ [43]. Moreover, there exist $C > 1$ and $\rho \in (0,1)$ such that $\max_{s \in \mathcal{S}} \|p_\pi^k(S_k = \cdot | S_1 = s) - \mu(\cdot)\|_{\mathrm{TV}} \leq C\rho^{k-1}$ for all $k \geq 1$ [43], where $p_\pi$ denotes the transition kernel of the Markov chain $\{S_k\}$ induced by $\pi$.

To aid in the statement of our main theorem, we introduce the following notation. Let $\tau_k = \min\{t : C\rho^{t-1} \leq \alpha/(k+h)\}$, $b_k = \alpha|\mathcal{S}||\mathcal{A}|\log(\lceil \frac{k-1}{|\mathcal{S}||\mathcal{A}|}\rceil/h + 1)$, and $m_k = b_k + p_{\mathrm{span}}(Q^*)$, where $\lceil x \rceil$ returns the smallest integer greater than or equal to $x$. Note that $\tau_k$, $b_k$, and $m_k$ all grow at most logarithmically in $k$. Let $D_{\min} = \min_{s,a} \mu(s)\pi(a|s)$, which is positive under Assumption 3.1.

**Theorem 3.1.** *Consider $\{Q_k\}$ generated by Algorithm 1. Suppose that Assumptions 2.1 and 3.1 are satisfied. Then, there exists $K > 0$ such that for any $k \in \{1, 2, \cdots, K\}$, we have $p_{\mathrm{span}}(Q_k - Q^*) \leq b_k + p_{\mathrm{span}}(Q^*)$ almost surely (a.s.), and for any $k \geq K+1$, we have*

$$
\mathbb{E}[p_{\mathrm{span}}(Q_k - Q^*)^2] \leq
\begin{cases}
3m_K^2 \left( \dfrac{K+h}{k+h} \right)^{\frac{\alpha(1-\beta)}{2}} + \dfrac{C_1 \tau_k (m_k+1)^2}{(k+h)^{\frac{\alpha(1-\beta)}{2}}}, & \text{if } \alpha(1-\beta) < 2, \\[2ex]
3m_K^2 \left( \dfrac{K+h}{k+h} \right) + \dfrac{C_2 \tau_k (m_k+1)^2 \log(k+h)}{(k+h)}, & \text{if } \alpha(1-\beta) = 2, \\[2ex]
3m_K^2 \left( \dfrac{K+h}{k+h} \right)^{\frac{\alpha(1-\beta)}{2}} + \dfrac{C_1 \tau_k (m_k+1)^2}{(k+h)}, & \text{if } \alpha(1-\beta) > 2.
\end{cases}
$$

*Here, $C_1$ and $C_2$ are problem-dependent constants defined as*

$$
C_1 = \frac{C\alpha^2|\mathcal{S}||\mathcal{A}|\log(|\mathcal{S}||\mathcal{A}|)}{(1-\rho)(1-\beta)D_{\min}^2 \min(1,\alpha)|2-(1-\beta)\alpha|}, \text{ and } C_2 = \frac{C|\mathcal{S}||\mathcal{A}|\log(|\mathcal{S}||\mathcal{A}|)}{(1-\rho)(1-\beta)^3 D_{\min}^2},
$$

*where $c_1$ and $c_2$ are absolute constants.*

A proof sketch of Theorem 3.1 is provided in Section 5, with the full proof deferred to Appendix A. Due to the presence of Markovian noise and the use of adaptive stepsizes, the convergence bound in Theorem 3.1 does not hold from the initial iteration. Specifically, prior to iteration $K$, we establish an almost-sure bound that grows at most logarithmically with $k$. After iteration $K$, the "averaging" effect becomes dominant, and the mean-square error begins to decay, with the rate of convergence depending critically on the choice of the constant $\alpha$ in Algorithm 1. In particular, if $\alpha$ is below the threshold $2/(1-\beta)$, the convergence rate is $\mathcal{O}(k^{-\alpha(1-\beta)/2})$, which can be arbitrarily slow. Conversely, if $\alpha$ exceeds the threshold, the convergence rate improves to the optimal $\mathcal{O}(1/k)$ up to logarithmic factors. A qualitatively similar phenomenon has been observed in norm-contractive SA algorithms [21], linear SA algorithms [9], and stochastic gradient descent/ascent algorithms [40].

Based on Theorem 3.1, we have the following corollary for the sample complexity.

**Corollary 3.1.1.** *Given $\epsilon > 0$, to achieve $\mathbb{E}[p_{\mathrm{span}}(Q_k - Q^*)] \leq \epsilon$ with Algorithm 1, the sample complexity is $\tilde{\mathcal{O}}\left(|\mathcal{S}|^3|\mathcal{A}|^3 D_{\min}^{-2}(1-\beta)^{-5}\epsilon^{-2}\right)$.*

The proof of Corollary 3.1.1 is provided in Appendix A.5. Notably, the dependence on the desired accuracy level is $\tilde{\mathcal{O}}(\epsilon^{-2})$, which is unimprovable in general. While we make the dependence on the size of the state–action space and the seminorm contraction factor explicit, these terms are by no means optimal in light of existing information-theoretic lower bounds [36]. It is worth noting, however, that the lower bound in [36] was derived under a generative model that provides i.i.d. samples, whereas our analysis considers the more challenging Markovian sampling setting. Despite this discrepancy, tightening the dependencies on $|\mathcal{S}||\mathcal{A}|$, $1/(1-\beta)$, and $D_{\min}$, whether through refined analysis or improved algorithmic techniques such as Polyak averaging or variance reduction, remains an important direction for future research. That being said, we emphasize that this is the first result to establish last-iterate convergence rate guarantees for model-free $Q$-learning with an asynchronous implementation.

## 4  The Necessity of Using Adaptive Stepsizes

As discussed in the previous section, the most important feature of Algorithm 1 is its use of adaptive stepsizes. Theoretically breaking their strong correlation with the iterates $Q_k$ will be our primary

technical innovation in the proof of Theorem 3.1. However, before delving into the proofs, it is important to first answer the following two questions: (1) Is the use of adaptive stepsizes necessary, and how does the algorithm behave if we instead use universal stepsizes (e.g., $\alpha_k(s,a) = 1/k$ for all $(s,a)$)? (2) If adaptive stepsizes are indeed necessary, how do we interpret their role? In this section, we provide clear answers to these questions. The insights developed in this section will play a central role in guiding our proof of Theorem 3.1.

### 4.1 Q-Learning with Universal Stepsizes: Provable Convergence to the Wrong Target

In this section, we will show that if we use universal stepsizes in Algorithm 1, the algorithm fails to converge in $p_{\text{span}}(\cdot)$ to $Q^*$. To this end, we first present the main update equation for $Q$-learning with universal stepsizes in the following:

$$Q_{k+1}(s,a) = Q_k(s,a) + \alpha_k \mathbb{1}_{\{(S_k,A_k)=(s,a)\}} \left( R(S_k,A_k) + \max_{a'} Q_k(S_{k+1},a') - Q_k(S_k,A_k) \right) \quad (3)$$

for all $(s,a)$, where the asynchronous nature of the update equation is captured by the indicator function. Here, the stepsize $\alpha_k$ depends only on $k$ and is independent of the state-action pairs.

The $Q$-learning algorithm described in Eq. (3) takes the typical form of a Markovian SA algorithm. We begin with the reformulation and identify the target equation it aims to solve. Let $Y_k = (S_k, A_k, S_{k+1})$ for all $k \geq 1$. Note that $\{Y_k\}$ forms a Markov chain, with state space $\mathcal{Y} = \mathcal{S} \times \mathcal{A} \times \mathcal{S}$. In addition, under Assumption 3.1, the Markov chain $\{Y_k\}$ admits a unique stationary distribution $\nu \in \Delta(\mathcal{Y})$ satisfying $\nu(s,a,s') = \mu(s)\pi(a|s)p(s'|s,a)$ for all $(s,a,s') \in \mathcal{Y}$. Let $G : \mathbb{R}^{|\mathcal{S}||\mathcal{A}|} \times \mathcal{Y} \to \mathbb{R}^{|\mathcal{S}||\mathcal{A}|}$ be an operator defined such that given input arguments $Q \in \mathbb{R}^{|\mathcal{S}||\mathcal{A}|}$ and $y = (s_0, a_0, s_1) \in \mathcal{Y}$, the $(s,a)$-th entry of the output of the operator is given by

$$[G(Q,y)](s,a) = \mathbb{1}_{\{(s_0,a_0)=(s,a)\}} \left( \mathcal{R}(s_0,a_0) + \max_{a' \in \mathcal{A}} Q(s_1,a') - Q(s_0,a_0) \right) + Q(s,a).$$

With $\{Y_k\}$ and $G(\cdot)$ defined above, Eq. (3) can be formulated as a Markovian SA:

$$Q_{k+1} = Q_k + \alpha_k(G(Q_k, Y_k) - Q_k). \quad (4)$$

Let $\bar{\mathcal{H}} : \mathbb{R}^{|\mathcal{S}||\mathcal{A}|} \to \mathbb{R}^{|\mathcal{S}||\mathcal{A}|}$ be the "expected" operator defined as:

$$\bar{\mathcal{H}}(Q) := \mathbb{E}_{Y \sim \nu}[G(Q,Y)] = [(I-D) + D\mathcal{H}](Q), \quad \forall Q \in \mathbb{R}^{|\mathcal{S}||\mathcal{A}|}, \quad (5)$$

where $\mathcal{H}(\cdot)$ is the Bellman operator, and $D$ is an $|\mathcal{S}||\mathcal{A}| \times |\mathcal{S}||\mathcal{A}|$ diagonal matrix with diagonal entries $\{\mu(s)\pi(a|s)\}_{(s,a)\in\mathcal{S}\times\mathcal{A}}$. Inspired by [21] (which studies $Q$-learning in the discounted setting), we refer to $\bar{\mathcal{H}}(\cdot)$ as the *asynchronous Bellman operator*. The reason is that for each $(s,a)$, the output $[\bar{\mathcal{H}}(Q)](s,a)$, according to its definition, can be viewed as the expectation of a random variable that takes $[\mathcal{H}(Q)](s,a)$ with probability $D(s,a)$ and takes $Q(s,a)$ with probability $1-D(s,a)$, capturing the asynchronous nature of $Q$-learning.

The following lemma shows that $\bar{\mathcal{H}}(\cdot)$ is also a contraction mapping with respect to $p_{\text{span}}(\cdot)$.

**Lemma 4.1.** *Under Assumptions 2.1 and 3.1, the asynchronous Bellman operator $\bar{\mathcal{H}}(\cdot)$ is a contraction mapping with respect to $p_{span}(\cdot)$, with contraction factor $\bar{\beta} := 1 - (1-\beta)D_{\min}$.*

The proof of Lemma 4.1 is presented in Appendix C.3. In light of Lemma 4.1, we identify that the Markovian SA algorithm described in Eq. (4) is designed to solve the seminorm fixed-point equation

$$p_{\text{span}}(\bar{\mathcal{H}}(Q) - Q) = 0, \quad (6)$$

which we refer to as the *asynchronous Bellman equation*, in contrast to the original Bellman equation (2). Therefore, applying recent results on Markovian SA with seminorm contractive operators [24], we show in the following proposition that $Q$-learning with universal stepsizes (cf. Eq. (3)) provably achieves mean-square convergence in $p_{\text{span}}(\cdot)$ to a *wrong target*. See Appendix C.4 for its proof.

**Proposition 4.1.** *Suppose that Assumptions 2.1 and 3.1 are satisfied. Then, we have*

$$\begin{cases} \{Q \mid p_{span}(\bar{\mathcal{H}}(Q) - Q) = 0\} = \{Q \mid p_{span}(\mathcal{H}(Q) - Q) = 0\}, & \text{if } D = I/(|\mathcal{S}||\mathcal{A}|), \\ \{Q \mid p_{span}(\bar{\mathcal{H}}(Q) - Q) = 0\} \cap \{Q \mid p_{span}(\mathcal{H}(Q) - Q) = 0\} = \emptyset, & \text{otherwise.} \end{cases} \quad (7)$$

*Moreover, when $\alpha_k = \alpha/(k+h)$ with appropriately chosen $\alpha$ and $h$, the Q-learning algorithm described in Eq. (3) achieves $\mathbb{E}[p_{span}(Q_k - \bar{Q}^*)^2] \leq \tilde{\mathcal{O}}(1/k)$, where $\bar{Q}^*$ is a particular solution to the asynchronous Bellman equation $p_{span}(\bar{\mathcal{H}}(Q) - Q) = 0$.*

Since the sets of solutions to the asynchronous Bellman equation and the original Bellman equation are completely disjoint, except in the degenerate case where $D = I/(|\mathcal{S}||\mathcal{A}|)$, Proposition 4.1 reveals the following critical issue: *as long as $D \neq I/(|\mathcal{S}||\mathcal{A}|)$, Q-learning with universal stepsizes is guaranteed not to converge to any point in the desired set of solutions $\mathcal{Q} = \{Q^* + ce \mid c \in \mathbb{R}\}$ of the original Bellman equation $p_{span}(\mathcal{H}(Q) - Q) = 0$.*

In summary, the fundamental issue for $Q$-learning with universal stepsizes in the average-reward setting is that the combination of a seminorm contraction mapping and asynchronous updates can make the set of solutions to the asynchronous Bellman equation $p_{\text{span}}(\bar{\mathcal{H}}(Q) - Q) = 0$ completely different from that of the original Bellman equation $p_{\text{span}}(\mathcal{H}(Q) - Q) = 0$.

Before illustrating how adaptive stepsizes resolve this issue, we quantify the discrepancy through a sensitivity analysis, presented in the following lemma. The proof of Lemma 4.2 is presented in Appendix C.5.

**Lemma 4.2.** *There exists a constant $c \in \mathbb{R}$ such that $p_{span}(\bar{Q}^* - Q^*) \leq \frac{|c_1|\, p_{span}(D^{-1}e)}{1-\beta}$.*

To interpret this result, consider the special case $D = I/(|\mathcal{S}||\mathcal{A}|)$, in which the right-hand side vanishes and hence $p_{\text{span}}(\bar{Q}^* - Q^*) = 0$, implying that the sets of solutions to $p_{\text{span}}(\bar{\mathcal{H}}(Q) - Q) = 0$ and $p_{\text{span}}(\mathcal{H}(Q) - Q) = 0$ coincide. This agrees with Proposition 4.1. More generally, when $D \neq I/(|\mathcal{S}||\mathcal{A}|)$, the above inequality provides a sensitivity bound.

**The Discounted Setting.**   One might ask: why is discounted $Q$-learning able to achieve provable convergence to the optimal $Q$-function with universal stepsizes? To illustrate, consider the $\gamma$-discounted MDP that shares the same transition kernel and reward function as the average-reward MDP studied in this work. Let $Q_\gamma^*$ be the optimal $Q$-function, and let $\mathcal{H}_\gamma(\cdot)$ denote the Bellman operator, which is a $\gamma$-contraction mapping with respect to $\|\cdot\|_\infty$ [8, 58, 67]. Following the same line of reasoning, discounted $Q$-learning (with asynchronous updates) can be formulated as a Markovian SA algorithm for solving the fixed-point equation $\bar{\mathcal{H}}_\gamma(Q) = Q$, where $\bar{\mathcal{H}}_\gamma(\cdot)$ is the *asynchronous Bellman operator* in the discounted setting, defined as $\bar{\mathcal{H}}_\gamma(Q) = [(I - D) + D\mathcal{H}_\gamma](Q)$. It has been shown in the existing literature [12, 21] that, under Assumption 3.1, the asynchronous Bellman operator $\bar{\mathcal{H}}_\gamma(\cdot)$ maintains the following two important properties of the original Bellman operator $\mathcal{H}_\gamma(\cdot)$: (1) $\bar{\mathcal{H}}_\gamma(\cdot)$ is a contraction mapping with respect to $\|\cdot\|_\infty$, and (2) the asynchronous Bellman equation $\bar{\mathcal{H}}_\gamma(Q) = Q$ admits a unique solution $Q_\gamma^*$, which is the optimal $Q$-function in the discounted setting. Consequently, discounted $Q$-learning with universal stepsizes converges to $Q_\gamma^*$ by standard results on Markovian SA with norm-contractive operators [7, 13].

### 4.2   Q-Learning with Adaptive Stepsizes: Implicit Importance Sampling to the Rescue

In view of Proposition 4.1 and the definition of the asynchronous Bellman operator in Eq. (5), suppose that the triple $(S_k, A_k, S_{k+1})$ were drawn from the distribution $\nu' \in \Delta(\mathcal{Y})$ defined by $\nu'(y) = p(s' \mid s, a)/(|\mathcal{S}||\mathcal{A}|)$ for all $y = (s, a, s') \in \mathcal{Y}$. Note that $\nu'(\cdot)$ corresponds to the joint distribution induced by sampling $S_k \sim \text{Unif}(\mathcal{S})$, $A_k \sim \text{Unif}(\mathcal{A})$, and $S_{k+1} \sim p(\cdot \mid S_k, A_k)$. Then, we would have $D = I/(|\mathcal{S}||\mathcal{A}|)$, and the asynchronous Bellman equation would share the same set of solutions as the original Bellman equation. However, when following the trajectory of the Markov chain $\{(S_k, A_k, S_{k+1})\}$, even asymptotically, we only receive samples from its stationary distribution $\nu(\cdot)$, which satisfies $\nu(y) = \mu(s)\pi(a \mid s)p(s' \mid s, a)$ for any $y = (s, a, s') \in \mathcal{Y}$. Phrased this way, the solution to fixing $Q$-learning with universal stepsizes becomes natural: *importance sampling*.

Importance sampling is a technique widely used in statistics [38], rare-event simulation [16], and off-policy RL [27]. Specifically, when direct sampling from a target distribution is difficult, importance sampling enables the estimation of expectations with respect to the target distribution by instead drawing samples from a different, more accessible distribution, referred to as the behavior distribution. The key idea is to reweight the samples drawn from the behavior distribution according to the likelihood ratio between the target and behavior distributions.

Coming back to the problem of fixing $Q$-learning with universal stepsizes (cf. Eq. (3)), we want to perform importance sampling with $\nu'(\cdot)$ being the target distribution and $\nu(\cdot)$ being the behavior distribution. Therefore, using $\delta_k = (R(S_k, A_k) + \max_{a'} Q_k(S_{k+1}, a') - Q_k(S_k, A_k))$ to denote

the temporal difference for simplicity of notation, the algorithm becomes

$$Q_{k+1}(s,a) = Q_k(s,a) + \alpha_k \frac{\mathbb{1}_{\{(S_k,A_k)=(s,a)\}} \nu'(s,a,s')}{\nu(s,a,s')} \delta_k$$

$$= Q_k(s,a) + \alpha_k \frac{\mathbb{1}_{\{(S_k,A_k)=(s,a)\}}}{|\mathcal{S}||\mathcal{A}|D(s,a)} \delta_k$$

$$= Q_k(s,a) + \tilde{\alpha}_k \frac{\mathbb{1}_{\{(S_k,A_k)=(s,a)\}}}{D(s,a)} \delta_k, \quad \forall (s,a) \in \mathcal{S} \times \mathcal{A}, \tag{8}$$

where the last equality follows by absorbing the $1/(|\mathcal{S}||\mathcal{A}|)$ factor into the stepsizes, i.e., by redefining $\tilde{\alpha}_k = \alpha_k/(|\mathcal{S}||\mathcal{A}|)$.

While the algorithm described in Eq. (8) seems promising, it is impractical because we do not have access to the stationary distribution of the Markov chain $\{(S_k, A_k)\}$ induced by $\pi$. A natural way to obtain an estimate of $D(s,a)$ is by using the empirical frequency. Specifically, recall that we have denoted $N_k(s,a)$ as the total number of visits to state-action pair $(s,a)$ up to the $k$-th iteration. Then, the estimator $N_k(s,a)/k$, or more generally, $D_k(s,a) := (N_k(s,a) + h)/(k + h)$ for any $h \geq 0$, is an asymptotically unbiased estimator of $D(s,a)$ because under Assumption 3.1, the law of large numbers holds for functions of the Markov chain $\{(S_k, A_k)\}$ [52]. Substituting the estimator $D_k(s,a)$ of $D(s,a)$ into Eq. (8), when the universal stepsize is set to $\tilde{\alpha}_k = \alpha/(k + h)$, we obtain

$$Q_{k+1}(s,a) = Q_k(s,a) + \frac{\tilde{\alpha}_k \mathbb{1}_{\{(S_k,A_k)=(s,a)\}}}{D_k(s,a)} \delta_k = Q_k(s,a) + \frac{\alpha \mathbb{1}_{\{(S_k,A_k)=(s,a)\}}}{N_k(s,a) + h} \delta_k, \tag{9}$$

for all $(s,a) \in \mathcal{S} \times \mathcal{A}$. Note that this is exactly the main update equation of the $Q$-learning algorithm presented in Algorithm 1, which uses adaptive stepsizes of the form $\alpha_k(s,a) = \alpha/(N_k(s,a) + h)$.

In summary, the use of adaptive stepsizes in $Q$-learning (which is necessary, as illustrated in Section 4.1) can be viewed as a form of implicit importance sampling that counteracts the effect of asynchronous updates, where the importance sampling weights are effectively constructed from the empirical frequency of visits to each state-action pair.

## 5 Proof Sketch of Theorem 3.1

Although the underlying stochastic process $\{(S_k, A_k)\}$ driving Algorithm 1 is a Markov chain, the $k$-th update depends on the entire sample history $(S_1, A_1, S_2, A_2, \ldots, S_k, A_k)$ due to the use of adaptive stepsizes. As a result, the algorithm, viewed as a SA, is non-Markovian. This presents the main technical challenge in establishing a finite-time analysis. To address it, our proof proceeds through the following key steps.

**A Markovian Reformulation of Non-Markovian SA:** In light of the discussion in Section 4.2, especially Eq. (9), the $k$-th iteration of Algorithm 1 depends deterministically on the following random variables: the current iterate $Q_k$, the $k$-th transition $(S_k, A_k, S_{k+1})$, and the empirical frequency matrix $D_k$, which is an $|\mathcal{S}||\mathcal{A}|$-by-$|\mathcal{S}||\mathcal{A}|$ diagonal matrix with diagonal entries $\{(N_k(s,a)+h)/(k+h)\}_{(s,a)\in\mathcal{S}\times\mathcal{A}}$. This motivates us to define a stochastic process $\{Z_k = (D_k, S_k, A_k, S_{k+1})\}_{k\geq 1}$ and to reformulate Algorithm 1 as an SA driven by $\{Z_k\}$:

$$Q_{k+1} = Q_k + \alpha_k(F(Q_k, Z_k) - Q_k), \tag{10}$$

where $\alpha_k = \alpha/(k+h)$ is the stepsize and $F : \mathbb{R}^{|\mathcal{S}||\mathcal{A}|} \times \mathcal{Z} \to \mathbb{R}^{|\mathcal{S}||\mathcal{A}|}$ is an operator defined such that given input arguments $Q \in \mathbb{R}^{|\mathcal{S}||\mathcal{A}|}$ and $z = (\tilde{D}, s_0, a_0, s_1) \in \mathcal{Z}$, the $(s,a)$-th entry of the output of the operator is given by

$$[F(Q,z)](s,a) = \frac{\mathbb{1}_{\{(s_0,a_0)=(s,a)\}}}{\tilde{D}(s,a)} \left( \mathcal{R}(s_0,a_0) + \max_{a'\in\mathcal{A}} Q(s_1,a') - Q(s_0,a_0) \right) + Q(s,a). \tag{11}$$

Although the reformulation casts the $Q$-learning algorithm as an SA with universal stepsizes, we emphasize that the stochastic process $\{Z_k\}$ forms a time-inhomogeneous Markov chain and therefore does not exhibit the geometric mixing property typically assumed in the SA literature [9, 65, 68].

**A Lyapunov Framework for the Analysis:** After reformulating Algorithm 1 as an SA with time-inhomogeneous Markovian noise, as presented in Eq. (10), we employ a Lyapunov-drift approach

to carry out the finite-time analysis. Inspired by [18, 21, 24, 81], we construct a Lyapunov function $M(\cdot)$ as the generalized Moreau envelope of $p_{\text{span}}(Q^2)/2$. Our goal is to show that the sequence $\{Q_k\}$ generated by the SA algorithm in Eq. (10) exhibits a negative drift with respect to $M(\cdot)$:

$$\mathbb{E}[M(Q_{k+1} - Q^*)] \leq (1 - \mathcal{O}(\alpha_k))\mathbb{E}[M(Q_k - Q^*)] + o(\alpha_k). \tag{12}$$

To establish a recursive inequality of the form (12), the high-level idea is to show that the "deterministic" part of the update in Eq. (10) induces a negative drift of $-\mathcal{O}(\alpha_k)\mathbb{E}[M(Q_k - Q^*)]$, while the error due to stochasticity is at most $o(\alpha_k)$.

Establishing the negative drift is not particularly challenging in our case due to the span seminorm contraction property of the Bellman operator $\mathcal{H}(\cdot)$ and the construction of the Lyapunov function $M(\cdot)$. The main challenge lies in controlling the stochastic error, particularly in handling the correlation between the iterates $\{Q_k\}$ and the stochastic process $\{Z_k\}$. In the existing literature studying Markovian SA, if the underlying noise sequence is i.i.d. or forms a uniformly ergodic Markov chain, the correlation between the iterate and the noise can be handled using either an approach based on conditioning and mixing time [65] or a more recently developed approach based on the Poisson equation [17, 30, 31]. Unfortunately, neither approach is applicable here due to the fact that $\{Z_k\}$ is time-inhomogeneous and lacks geometric mixing.

**Breaking the Correlation:** To break the correlation between $Z_k$ and $Q_k$, we develop an approach consisting of the following two steps:

- *Time-Varying Almost-Sure Bounds:* The first step is to show that the iterate $Q_k$, while not uniformly bounded by a constant, satisfies a time-varying *almost-sure* bound that grows at most *logarithmically* with $k$ (cf. Proposition A.1). These two properties together enable us to decouple the iterate $Q_k$ from the time-inhomogeneous Markovian noise $Z_k$.

- *Conditioning Arguments + Markov Concentration Inequalities:* After decoupling, we handle the randomness in $Z_k = (D_k, S_k, A_k, S_{k+1})$ separately, where two challenges arise: (1) the correlation between $D_k$ and $(S_k, A_k, S_{k+1})$, and (2) the presence of the empirical frequency $D_k(s, a)$ in the denominator of the algorithm update (cf. Eq. (9) and Eq. (11)), which breaks linearity. To address the first challenge, we use a conditioning argument based on the fast mixing of $(S_k, A_k, S_{k+1})$ (cf. Assumption 3.2). To tackle the second, we employ Markov chain concentration inequalities and the $\mathcal{L}^2$ weak law of large numbers for functions of Markov chains.

In the end, we obtain a Lyapunov-drift inequality of the form (12), which can be recursively applied to derive the finite-time bound stated in Theorem 3.1.

## 6 Conclusion

In this work, we present the first study on the last-iterate convergence rates of average-reward $Q$-learning with an asynchronous implementation. Moreover, we show that the key to achieving the desired convergence is the use of adaptive stepsizes, which can be viewed as a form of implicit importance sampling to counteract the asynchronous updates. We believe that our analysis is broadly applicable to the study of general SA algorithms with adaptive stepsizes.

Regarding future work, we identify three directions. First, the sample complexity dependence on the size of the state–action space and the seminorm contraction factor presented in this work is generally not tight. Developing refined analyses or improved algorithmic designs (such as incorporating Polyak averaging or variance reduction) to tighten these bounds is an interesting research direction. Second, investigating the convergence behavior of $Q$-learning with other variants of adaptive stepsizes (as illustrated in Section 4.2) is another promising direction. Finally, while this work focuses on the setting where the Bellman operator is a seminorm contraction mapping, it is worth exploring how to conduct a finite-time analysis of $Q$-learning without this assumption. Since the Bellman operator is always non-expansive with respect to $\|\cdot\|_\infty$, studying SA with non-expansive operators and (time-inhomogeneous) Markovian noise will likely play an important role in advancing this direction.

## Acknowledgments

I would like to thank Mr. Shaan Ul Haque from Georgia Tech for his help in identifying the seminorm contraction property of the asynchronous Bellman operator presented in Lemma 4.1. I would also like to thank Dr. Siva Theja Maguluri from Georgia Tech for his valuable feedback on an early draft of this work.

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

# Appendices

## A   Proof of Theorem 3.1

We follow the road map described in Section 5 to prove Theorem 3.1.

### A.1   A Markovian Reformulation of Non-Markovian Stochastic Approximation

Let $Z_k = (D_k, S_k, A_k, S_{k+1})$ for all $k \geq 1$, whose state space is denoted by $\mathcal{Z}$. To see that $\{Z_k\}$ forms a time-inhomogeneous Markov chain, given $Z_k = (D_k, S_k, A_k, S_{k+1})$, consider the distribution of $Z_{k+1} = (D_{k+1}, S_{k+1}, A_{k+1}, S_{k+2})$. Since $S_{k+1}$ is given in $Z_k$, $A_{k+1} \sim \pi(\cdot|S_{k+1})$, $S_{k+2} \sim p(\cdot|S_{k+1}, A_{k+1})$, and

$$D_{k+1}(s,a) = \frac{N_k(s,a) + \mathbb{1}_{\{(S_{k+1},A_{k+1})=(s,a)\}} + h}{k+1+h}$$
$$= \frac{(k+h)D_k(s,a) + \mathbb{1}_{\{(S_{k+1},A_{k+1})=(s,a)\}} + h}{k+1+h}, \quad \forall (s,a), \tag{13}$$

which is a deterministic function of $k$, $(S_{k+1}, A_{k+1})$, and $D_k$, the stochastic process $\{Z_k\}$ is a time-inhomogeneous Markov chain, where the time inhomogeneity arises from the fact that the transition of $D_k$ depends on $k$.

Although $\{Z_k\}$ is time-inhomogeneous, it admits a unique limiting distribution $\mu_z$ satisfying $\mu_z(\tilde{D}, s_0, a_0, s_1) = \mathbb{1}_{\{\tilde{D}=D\}}\mu(s)\pi(a_0|s_0)p(s_1|s_0, a_0)$, which follows from Assumption 3.1 and the strong law of large numbers for functions of Markov chains [52]. However, since it is well known that the convergence rate from $D_k$ to $D$ is $\tilde{\mathcal{O}}(k^{-1/2})$ (measured in $\mathbb{E}[\|D_k - D\|_2]$), which is sublinear, the Markov chain $\{Z_k\}$ does not exhibit the geometric mixing property typically used in the existing study of Markovian SA.

With $Z_k$ defined above, to reformulate Algorithm 1 as a Markovian SA, let $F : \mathbb{R}^{|\mathcal{S}||\mathcal{A}|} \times \mathcal{Z} \to \mathbb{R}^{|\mathcal{S}||\mathcal{A}|}$ be an operator defined such that given input arguments $Q \in \mathbb{R}^{|\mathcal{S}||\mathcal{A}|}$ and $z = (\tilde{D}, s_0, a_0, s_1) \in \mathcal{Z}$, the $(s,a)$-th entry of the output of the operator is given by

$$[F(Q,z)](s,a) = \frac{\mathbb{1}_{\{(s_0,a_0)=(s,a)\}}}{\tilde{D}(s,a)}\left(\mathcal{R}(s_0,a_0) + \max_{a'\in\mathcal{A}} Q(s_1,a') - Q(s_0,a_0)\right) + Q(s,a). \tag{14}$$

Using the definition of $Z_k$ and $F(\cdot)$, the main update equation from Algorithm 1 can be written as

$$Q_{k+1} = Q_k + \alpha_k(F(Q_k, Z_k) - Q_k), \tag{15}$$

where $\alpha_k = \alpha/(k+h)$. The properties of the operator $F(\cdot)$, along with its connections to the Bellman operator $\mathcal{H}(\cdot)$ are summarized in the following lemma, whose proof is presented in Appendix B.1.

**Lemma A.1.** *The following properties hold regarding the operator $F(\cdot)$.*

*(1) For any $Q_1, Q_2, \tilde{D}$, and $y = (s_0, a_0, s_1) \in \mathcal{Y}$, we have*

$$p_{span}(F(Q_1, \tilde{D}, y) - F(Q_2, \tilde{D}, y)) \leq \frac{2}{\tilde{D}(s_0, a_0)} p_{span}(Q_1 - Q_2).$$

*(2) For any $Q, \tilde{D}$, and $y = (s_0, a_0, s_1) \in \mathcal{Y}$, we have*

$$p_{span}(F(Q, \tilde{D}, y)) \leq \frac{2}{\tilde{D}(s_0, a_0)}(p_{span}(Q) + 1).$$

*(3) For any $Q$, we have $\mathbb{E}_{Y\sim\nu}[F(Q, D, Y)] = \mathcal{H}(Q)$.*

Among the properties stated in Lemma A.1, Parts (1) and (2) present the Lipschitz continuity of $F(\cdot)$ and its at-most-affine growth with respect to the estimated $Q$-function. However, we point out that the Lipschitz constant is inherently random in our analysis, as it depends on the input argument $\tilde{D}$, which corresponds to the empirical frequency matrix $D_k$. Lemma A.1 (3) further states that the operator

$F(\cdot)$ is an asymptotically unbiased estimator of the Bellman operator $\mathcal{H}(\cdot)$, thereby justifying that Eq. (15) represents an SA algorithm for solving the Bellman equation (2).

Since the Bellman operator $\mathcal{H}(\cdot)$ will also frequently appear in our analysis, we summarize its properties in the following lemma. See Appendix B.2 for the proof.

**Lemma A.2.** *The following properties hold regarding the Bellman operator* $\mathcal{H}(\cdot)$.

*(1) For any $Q_1, Q_2$, we have $p_{span}(\mathcal{H}(Q_1) - \mathcal{H}(Q_2)) \leq p_{span}(Q_1 - Q_2)$.*

*(2) For any $Q$, we have $p_{span}(\mathcal{H}(Q)) \leq p_{span}(Q) + 1$.*

*Remark.* The proof of Lemma A.2 also implies $\|\mathcal{H}(Q_1) - \mathcal{H}(Q_2)\|_\infty \leq \|Q_1 - Q_2\|_\infty$ for any $Q_1$ and $Q_2$. This suggests that a possible direction for relaxing the seminorm contraction mapping assumption (cf. Assumption 2.1) is to consider (time-inhomogeneous) Markovian SA under non-expansive operators, as discussed in Section 6.

## A.2 A Lyapunov Framework for the Analysis

After reformulating Algorithm 1 in the form of Eq. (15), we will use a Lyapunov-drift approach to perform the finite-time analysis. Inspired by [18, 21, 24, 81], we construct the Lyapunov function as the generalized Moreau envelope, defined as the informal convolution between the square of the span seminorm and the square of the $\ell_q$-norm:

$$M_{q,\theta}(Q) := \min_{u \in \mathbb{R}^{|\mathcal{S}||\mathcal{A}|}} \left\{ \frac{1}{2} p_{span}^2(u) + \frac{1}{2\theta} \|Q - u\|_q^2 \right\}, \quad \forall Q \in \mathbb{R}^{|\mathcal{S}||\mathcal{A}|},$$

where both $q \geq 1$ and $\theta > 0$ are tunable parameters yet to be chosen. For simplicity of notation, we will write $M(\cdot)$ for $M_{q,\theta}(\cdot)$ throughout the rest of the proof.

Properties of this type of Lyapunov function have been thoroughly investigated in [21, 24], and are restated in the following lemma for the special case of the span seminorm, for completeness. Let $\ell_q = (|\mathcal{S}||\mathcal{A}|)^{-1/q}$ and $u_q = 1$. Note that we have $\ell_q \|Q\|_q \leq \|Q\|_\infty \leq u_q \|Q\|_q$ for any $Q$.

**Lemma A.3** (Proposition 4.1 from [24]). *The Lyapunov function $M(\cdot)$ has the following properties:*

*(1) The function $M(\cdot)$ is convex, differentiable, and satisfies*

$$M(Q_2) \leq M(Q_1) + \langle \nabla M(Q_1), Q_2 - Q_1 \rangle + \frac{L}{2} p_{span}(Q_2 - Q_1)^2, \quad \forall Q_1, Q_2 \in \mathbb{R}^d,$$

*where $L = (q-1)/(\ell_q^2 \theta)$.*

*(2) There exists a seminorm $p_m(\cdot)$, which satisfies $p_m(Q) = \min_{c \in \mathbb{R}} \|Q - ce\|_m$ for some norm $\|\cdot\|_m$, such that $M(Q) = p_m(Q)^2/2$.*

*(3) There exist $\ell_m = (1 + \theta \ell_q^2)^{1/2}$ and $u_m = (1 + \theta u_q^2)^{1/2}$ such that $\ell_m p_m(Q) \leq p_{span}(Q) \leq u_m p_m(Q)$ for all $Q \in \mathbb{R}^{|\mathcal{S}||\mathcal{A}|}$.*

*(4) It holds for all $Q \in \mathbb{R}^{|\mathcal{S}||\mathcal{A}|}$ and $c \in \mathbb{R}$ that $\langle \nabla M(Q), ce \rangle = 0$.*

*(5) It holds for all $Q_1, Q_2, Q_3 \in \mathbb{R}^{|\mathcal{S}||\mathcal{A}|}$ that $\langle \nabla M(Q_1) - \nabla M(Q_2), Q_3 \rangle \leq L p_{span}(Q_1 - Q_2) p_{span}(Q_3)$.*

Lemma A.3 establishes several key properties of $M(\cdot)$. Specifically, Part (1) states that $M(\cdot)$ is a smooth function with respect to the span seminorm. Part (2) states that $M(\cdot)$ itself can be written as the square of a seminorm with kernel space $\{ce \mid c \in \mathbb{R}\}$. Part (3) states that $M(\cdot)$ can approximate the seminorm-square function $p_{span}(Q)^2/2$ arbitrarily closely, since $\lim_{\theta \to 0} u_m/\ell_m = 1$. This property, together with Part (1), implies that $M(Q)$ is a smooth approximation of $p_{span}(Q^2)/2$. Part (4) states that the gradient of $M(\cdot)$ is always orthogonal to $\ker(p_{span})$. Finally, Part (5) follows as a consequence of Parts (1) and (4).

Using the smoothness of $M(\cdot)$ (cf. Lemma A.3 (1)) together with the reformulated update equation (15), we have for all $k \geq 1$ that

$$\mathbb{E}[M(Q_{k+1} - Q^*)] \leq \mathbb{E}[M(Q_k - Q^*)] + \mathbb{E}[\langle \nabla M(Q_k - Q^*), Q_{k+1} - Q_k \rangle]$$

$$\begin{aligned}
&+ \frac{L}{2}\mathbb{E}[p_{\text{span}}(Q_{k+1} - Q_k)^2] \\
&= \mathbb{E}[M(Q_k - Q^*)] + \alpha_k \mathbb{E}[\langle \nabla M(Q_k - Q^*), F(Q_k, Z_k) - Q_k \rangle] \\
&\quad + \frac{L\alpha_k^2}{2}\mathbb{E}[p_{\text{span}}(F(Q_k, Z_k) - Q_k)^2] \qquad\qquad \text{(Lemma A.3 (4))} \\
&= \mathbb{E}[M(Q_k - Q^*)] + \alpha_k \underbrace{\mathbb{E}[\langle \nabla M(Q_k - Q^*), \mathcal{H}(Q_k) - Q_k \rangle]}_{:=T_1} \\
&\quad + \alpha_k \underbrace{\mathbb{E}[\langle \nabla M(Q_k - Q^*), F(Q_k, D, Y_k) - \mathcal{H}(Q_k) \rangle]}_{:=T_2} \\
&\quad + \alpha_k \underbrace{\mathbb{E}[\langle \nabla M(Q_k - Q^*), F(Q_k, D_k, Y_k) - F(Q_k, D, Y_k) \rangle]}_{:=T_3} \\
&\quad + \frac{L\alpha_k^2}{2} \underbrace{\mathbb{E}[p_{\text{span}}(F(Q_k, Z_k) - Q_k)^2]}_{:=T_4},
\end{aligned} \tag{16}$$

where $Y_k = (S_k, A_k, S_{k+1})$ and $D$ is the $|\mathcal{S}||\mathcal{A}|$ by $|\mathcal{S}||\mathcal{A}|$ diagonal matrix with diagonal entries $\{\mu(s)\pi(a|s)\}_{(s,a)\in\mathcal{S}\times\mathcal{A}}$. Recall that $\{Y_k\}$ forms a time-homogeneous Markov chain, with state space denoted by $\mathcal{Y}$. Moreover, under Assumption 3.1, the Markov chain $\{Y_k\}$ admits a unique stationary distribution $\nu \in \Delta(\mathcal{Y})$, which satisfies $\nu(s, a, s') = \mu(s)\pi(a|s)p(s'|s, a)$ for all $y = (s, a, s') \in \mathcal{Y}$.

In view of Eq. (16), it remains to bound the terms $T_1, T_2, T_3,$ and $T_4$. We begin with the term $T_1$, which can be viewed as the "deterministic" part of Algorithm 1 because $F(\cdot, Z_k)$ is an asymptotically unbiased estimator of $\mathcal{H}(\cdot)$. We show in the following lemma that the term $T_1$ provides a negative drift. The proof of Lemma A.4 is presented in Appendix B.3.

**Lemma A.4.** *It holds for all $k \geq 1$ that $T_1 \leq -2\phi_1 \mathbb{E}[M(Q_k - Q^*)]$, where $\phi_1 = 1 - \beta u_m/\ell_m$.*

Since $\lim_{\theta \to 0} u_m/\ell_m = 1$ and $\beta \in (0, 1)$, we can make $\phi_1$ strictly positive (thereby ensuring a negative drift) by choosing $\theta$ appropriately.

Moving to the terms $T_2, T_3,$ and $T_4$ in Eq. (16), the term $T_2$ accounts for the error due to the Markovian noise $\{Y_k\}$, the term $T_3$ accounts for the error in estimating the matrix $D$ using the empirical frequency matrix $D_k$, and the term $T_4$ arises due to the fact that Eq. (15) is a discrete-time algorithm. To show that all of them are dominated by the negative drift provided by the term $T_1$, the analysis is different from and significantly more challenging than the existing literature on Markovian SA. Specifically, the main challenge in bounding these terms lies in handling the correlation among the iterate $Q_k$, the empirical frequency matrix $D_k$, and the Markov chain $\{Y_k\}$. In the existing literature, if the underlying noise sequence is i.i.d. or forms a uniformly ergodic Markov chain, such correlations can be handled using either an approach based on conditioning and mixing time [65] or a more recently developed approach based on the Poisson equation [17, 30, 31]. Unfortunately, neither approach is applicable here due to the fact that $\{Z_k = (D_k, Y_k)\}$ is time-inhomogeneous and lacks geometric mixing.

## A.3  Breaking the Correlation

Our approach to breaking the correlation relies on a combination of almost-sure time-varying bounds, conditioning arguments, and concentration inequalities for Markov chains. To illustrate this approach, we use the term $T_4$ as an example. Before proceeding, we note that among the error terms $T_2, T_3,$ and $T_4$, the term $T_4$ is the easiest to handle. Specifically, since the term $T_4$ is multiplied by $\alpha_k^2$ in Eq. (16), it suffices to show that $T_4 = \tilde{\mathcal{O}}(1)$ for it to be dominated by the negative drift (cf. Lemma A.4). In contrast, for the terms $T_2$ and $T_3$, we need to establish that they are $o(1)$, which is more challenging.

According to the definition of $F(\cdot)$ in Eq. (14), the vector $F(Q_k, Z_k) - Q_k$ has only one non-zero entry, i.e., the $(S_k, A_k)$-th one. Therefore, we have by the definition of $p_{\text{span}}(\cdot)$ that

$$\begin{aligned}
T_4 &= \mathbb{E}[p_{\text{span}}(F(Q_k, D_k, Y_k) - Q_k)^2] \\
&= \frac{1}{4}\mathbb{E}\left[\left(\frac{1}{D_k(S_k, A_k)}\right)^2 \left(\mathcal{R}(S_k, A_k) + \max_{a' \in \mathcal{A}} Q_k(S_{k+1}, a') - Q_k(S_k, A_k)\right)^2\right]
\end{aligned}$$

$$\leq \frac{1}{4}\mathbb{E}\left[\left(\frac{1}{D_k(S_k, A_k)}\right)^2 \left(|\mathcal{R}(S_k, A_k)| + \left|\max_{a' \in \mathcal{A}} Q_k(S_{k+1}, a') - Q_k(S_k, A_k)\right|\right)^2\right]$$

$$\leq \frac{1}{4}\mathbb{E}\left[\left(\frac{1}{D_k(S_k, A_k)}\right)^2 (1 + 2p_{\text{span}}(Q_k))^2\right]. \tag{17}$$

To proceed, the immediate challenge we face is that the random variable $D_k(S_k, A_k)$, which represents the frequency of visiting the state-action pair $(S_k, A_k)$ in the first $k$ time steps, is strongly correlated with $Q_k$.

**Step One: Time-Varying Almost-Sure Bounds.** The first step of our approach to break the correlation between $D_k(S_k, A_k)$ and $Q_k$ is to show that the iterates $Q_k$, while not uniformly bounded by a constant (which is a key difficulty compared to the discounted counterpart [29]), admit a time-varying almost-sure bound. Specifically, we have the following proposition.

**Proposition A.1.** *The following inequality holds for all $k \geq 1$:*

$$p_{\text{span}}(Q_k) \leq b_k,$$

*where $b_k = \alpha|\mathcal{S}||\mathcal{A}| \log(\frac{\lceil (k-1)/(|\mathcal{S}||\mathcal{A}|)\rceil + h}{h})$.*

*Remark.* Note that the proof of Theorem 3.1 for $k \in \{1, 2, \cdots, K-1\}$ is complete, since Proposition A.1 implies $p_{\text{span}}(Q_k - Q^*) \leq p_{\text{span}}(Q_k) + p_{\text{span}}(Q^*) \leq b_k + p_{\text{span}}(Q^*)$ a.s. for all $k$.

Proposition A.1 is a non-trivial observation, as the bound holds independent of the randomness in the sample trajectory and grows logarithmically in $k$. These two features together enable us to decouple the iterate $Q_k$ and the time-inhomogeneous Markovian noise $Z_k$. The proof of Proposition A.1 (presented in Appendix B.4) relies on a combination of induction and a combinatorial argument.

Apply the almost-sure time-varying bound from Proposition A.1 to Eq. (17), we have

$$T_4 \leq \frac{1}{4}\mathbb{E}\left[\left(\frac{1}{D_k(S_k, A_k)}\right)^2 (1 + 2p_{\text{span}}(Q_k))^2\right]$$

$$\leq (1 + b_k)^2 \mathbb{E}\left[\frac{1}{D_k(S_k, A_k)^2}\right]$$

$$= (1 + b_k)^2 \sum_{s,a} \mathbb{E}\left[\frac{\mathbb{1}_{\{(S_k, A_k)=(s,a)\}}}{D_k(s, a)^2}\right]. \tag{18}$$

**Step Two: A Conditioning Argument.** In view of Eq. (18), it remains to bound the quantity $\mathbb{E}[\mathbb{1}_{\{(S_k, A_k)=(s,a)\}}/D_k(s,a)^2]$ for any $(s,a)$. The immediate challenge lies in handling the correlation between the empirical frequency $D_k(s,a)$ and the indicator function $\mathbb{1}_{\{(S_k, A_k)=(s,a)\}}$. To address this issue, we apply a conditioning argument, which is inspired by [65]. Specifically, since $D_k(s,a) = (N_k(s,a) + h)/(k+h)$, we have for any $(s,a)$ and $\tilde{\tau} \leq k-1$ that

$$\mathbb{E}\left[\frac{\mathbb{1}_{\{(S_k, A_k)=(s,a)\}}}{D_k(s,a)^2}\right] = \mathbb{E}\left[\frac{\mathbb{1}_{\{(s,a)=(S_k, A_k)\}}(k+h)^2}{(N_k(s,a)+h)^2}\right]$$

$$= \mathbb{E}\left[\frac{\mathbb{1}_{\{(s,a)=(S_k, A_k)\}}(k+h)^2}{(N_{k-1}(s,a)+1+h)^2}\right]$$

(The pair $(S_k, A_k)$ is visited at time step $k$.)

$$\leq \mathbb{E}\left[\frac{\mathbb{1}_{\{(s,a)=(S_k, A_k)\}}(k+h)^2}{(N_{k-\tilde{\tau}-1}(s,a)+1+h)^2}\right] \quad (N_k(s,a) \text{ is an increasing function of } k.)$$

$$= \mathbb{E}\left[\mathbb{P}(S_k = s, A_k = a \mid S_{k-\tilde{\tau}-1}, A_{k-\tilde{\tau}-1})\frac{(k+h)^2}{(N_{k-\tilde{\tau}-1}(s,a)+1+h)^2}\right], \tag{19}$$

where the last equality follows from the tower property of conditional expectations and the Markov property.

Under Assumption 3.1, the Markov chain $\{(S_k, A_k)\}$ also enjoys geometric mixing. As a result, we have

$$
\begin{aligned}
&\mathbb{P}(S_k = s, A_k = a | S_{k-\tilde{\tau}-1}, A_{k-\tilde{\tau}-1}) \\
&\leq |\mathbb{P}(S_k = s, A_k = a | S_{k-\tilde{\tau}-1}, A_{k-\tilde{\tau}-1}) - D(s,a)| + D(s,a) \\
&\leq 2C\rho^{\tilde{\tau}} + D(s,a) \\
&\leq 2D(s,a),
\end{aligned}
\tag{20}
$$

where the last line follows from choosing $\tilde{\tau} = \min\{t : C\rho^t \leq D_{\min}\}$. See Appendix B.8 for more details.

Combining Eqs. (18), (19), and (20), we have

$$
T_4 \leq 2\left(1 + b_k\right)^2 \sum_{s,a} D(s,a)\mathbb{E}\left[\frac{(k+h)^2}{(N_{k-\tilde{\tau}-1}(s,a) + 1 + h)^2}\right].
\tag{21}
$$

**Step Three: Markov Chain Concentration.** To proceed from Eq. (21), the last challenge we face here is that the random variable $N_{k-\tilde{\tau}-1}(s,a)$ appears in the denominator of the fraction, which breaks the linearity. We overcome this challenge by using Markov chain concentration inequalities.

For simplicity of notation, denote $\bar{D}_{k-\tilde{\tau}-1}(s,a) = N_{k-\tilde{\tau}-1}(s,a)/(k-\tilde{\tau}-1)$. Given $\delta \in (0,1)$, for any $(s,a)$, let $E_\delta(s,a) = \{|\bar{D}_{k-\tilde{\tau}-1}(s,a) - D(s,a)| \leq \delta D(s,a)\}$ and let $E_\delta^c(s,a)$ be the complement of event $E_\delta(s,a)$. Note that on the event $E_\delta(s,a)$, we have $|\bar{D}_{k-\tilde{\tau}-1}(s,a) - D(s,a)| \leq \delta D(s,a)$, which implies $\bar{D}_{k-\tilde{\tau}-1}(s,a) \geq (1-\delta)D(s,a)$, while on the event $E_\delta^c(s,a)$, we have the trivial bound $\bar{D}_{k-\tilde{\tau}-1}(s,a) \geq 0$. Therefore, we obtain

$$
\begin{aligned}
\mathbb{E}\left[\frac{(k+h)^2}{(N_{k-\tilde{\tau}-1}(s,a) + 1 + h)^2}\right] &= \mathbb{E}\left[\frac{(k+h)^2}{((k-\tilde{\tau}-1)\bar{D}_{k-\tilde{\tau}-1}(s,a) + 1 + h)^2}\right] \\
&= (k+h)^2\mathbb{E}\left[\frac{\mathbb{1}_{\{E_\delta(s,a)\}} + \mathbb{1}_{\{E_\delta^c(s,a)\}}}{((k-\tilde{\tau}-1)\bar{D}_{k-\tilde{\tau}-1}(s,a) + 1 + h)^2}\right] \\
&\leq \frac{(k+h)^2}{(k-\tilde{\tau}-1)^2}\frac{1}{(1-\delta)^2 D(s,a)^2} + \frac{(k+h)^2}{(h+1)^2}\mathbb{P}(E_\delta^c(s,a)).
\end{aligned}
\tag{22}
$$

To bound $\mathbb{P}(E_\delta^c(s,a))$, we use the following Markov chain concentration inequality, which is a consequence of [56, Corollary 2.11]. The proof of Lemma A.5 is presented in Appendix B.5.

**Lemma A.5.** *Let $\{X_k\}_{k\geq 1}$ be a finite, irreducible, and aperiodic Markov chain taking values in $\mathcal{X} = \{1,2,3,\cdots,n\}$. Denote its unique stationary distribution by $\nu \in \Delta(\mathcal{X})$. Let $\tilde{C} \geq 1$ and $\tilde{\rho} \in (0,1)$ be such that $\max_{x \in \mathcal{X}} \|p(X_k = \cdot \mid X_1 = x) - \nu(\cdot)\|_{TV} \leq \tilde{C}\tilde{\rho}^{k-1}$ for all $k \geq 1$. Then, there exists $c > 0$ (which depends on the mixing time of the Markov chain) such that the following inequality holds for all $\epsilon \geq \frac{4\tilde{C}}{(1-\tilde{\rho})k}$:*

$$
\mathbb{P}\left(|\hat{\nu}_k(x) - \nu(x)| \geq \epsilon\right) \leq 2\exp(-ck\epsilon^2), \quad \forall x \in \mathcal{X}.
$$

*where $\hat{\nu}_k(x) = \sum_{j=1}^k \mathbb{1}_{\{X_j=x\}}/k$.*

Apply Lemma A.5 and we have $\mathbb{P}(E_\delta^c(s,a)) \leq 2\exp(-c_{mc}(k-\tilde{\tau}-1)\delta^2 D(s,a)^2)$ for any $\delta \geq 4C/[D_{\min}(1-\rho)(k-\tilde{\tau}-1)]$, where $c_{mc} > 0$ is a constant depending on the mixing time of the Markov chain $\{(S_k, A_k)\}$. Combining the previous inequality with Eq. (22) yields

$$
\begin{aligned}
\mathbb{E}\left[\frac{(k+h)^2}{(N_{k-\tilde{\tau}-1}(s,a) + 1 + h)^2}\right] &\leq \frac{(k+h)^2}{(k-\tilde{\tau}-1)^2}\frac{1}{(1-\delta)^2 D(s,a)^2} \\
&\quad + \frac{2(k+h)^2}{(h+1)^2}\exp(-c_{mc}(k-\tilde{\tau}-1)\delta^2 D(s,a)^2) \\
&\leq \frac{3}{D(s,a)^2},
\end{aligned}
$$

where the last line follows from (1) choosing $\delta$ properly based on $k$, and (2) when $k$ is large enough. See Appendix B.8 for more details.

Finally, using the previous inequality in Eq. (21), we have the following lemma, which shows that $T_4 = \tilde{\mathcal{O}}(1)$. A more detailed proof of Lemma A.6 (following the road map described above) is presented in Appendix B.8.

**Lemma A.6.** *The following inequality holds for all $k \geq K$:*

$$T_4 \leq \frac{18|\mathcal{S}||\mathcal{A}|(b_k + p_{span}(Q^*) + 1)^2}{D_{\min}}.$$

Following similar ideas, specifically, combining the time-varying almost-sure bounds (cf. Proposition A.1) with conditioning arguments and Markov chain concentration results (cf. Lemma A.5), we are able to bound the terms $T_2$ and $T_3$ on the right-hand side of Eq. (16). The results are presented in the following two propositions.

**Proposition A.2.** *The following inequality holds for all $k \geq K$:*

$$T_2 \leq \frac{28\tau_k L|\mathcal{S}||\mathcal{A}|(b_k + p_{span}(Q^*) + 1)^2}{D_{\min}^2}\alpha_k.$$

**Proposition A.3.** *The following inequality holds for all $k \geq K$:*

$$T_3 \leq \phi_1 \mathbb{E}\left[M(Q_k - Q^*)\right] + \frac{32C|\mathcal{S}||\mathcal{A}|(b_k + p_{span}(Q^*) + 1)^2}{\ell_m^2 \phi_1 (1 - \rho)\alpha D_{\min}^2}\alpha_k.$$

The proofs of Propositions A.2 and A.3 are presented in Appendices B.6 and B.7, respectively. Using the bounds we obtained for the terms $T_1$, $T_2$, $T_3$, and $T_4$ altogether in Eq. (16), we obtain the following result.

**Proposition A.4.** *The following inequality holds for all $k \geq K$:*

$$\mathbb{E}[M(Q_{k+1} - Q^*)] \leq (1 - \phi_1\alpha_k)\mathbb{E}[M(Q_k - Q^*)] + 35\phi_2\tau_k(b_k + p_{span}(Q^*) + 1)^2\alpha_k^2. \quad (23)$$

*where*

$$\phi_2 = \frac{C|\mathcal{S}||\mathcal{A}|}{(1 - \rho)\ell_m^2 \phi_1 D_{\min}^2 \alpha} + \frac{L|\mathcal{S}||\mathcal{A}|}{D_{\min}^2}.$$

Note that Eq. (23) is a one-step Lyapunov drift inequality of the desired form: it exhibits a negative drift with an additive error that is order-wise smaller than the magnitude of the drift.

## A.4 Solving the Recursion

This part is standard in the non-asymptotic analysis of SA algorithms. Repeatedly applying Proposition A.4, we have for any $k \geq K$ that

$$\mathbb{E}[M(Q_k - Q^*)] \leq \prod_{j=K}^{k-1}(1 - \phi_1\alpha_j)\mathbb{E}[M(Q_K - Q^*)]$$

$$+ 35\phi_2\sum_{i=K}^{k-1}\tau_i(B_i + p_{span}(Q^*) + 1)^2\alpha_i^2\prod_{j=i+1}^{k-1}(1 - \phi_1\alpha_j).$$

To translate the result to a bound on $\mathbb{E}[p_{span}(Q_k - Q^*)^2]$, using Lemma A.3 (3), we have

$$\frac{1}{2u_m^2}p_{span}(Q)^2 \leq M(Q) = \frac{1}{2}p_m(Q)^2 \leq \frac{1}{2\ell_m^2}p_{span}(Q)^2.$$

It follows that

$$\mathbb{E}[p_{span}(Q_k - Q^*)^2] \leq 2u_m^2\mathbb{E}[M(Q_k - Q^*)]$$

$$\leq 2u_m^2\prod_{j=K}^{k-1}(1 - \phi_1\alpha_j)\mathbb{E}[M(Q_K - Q^*)]$$

$$+ 70u_m^2\phi_2 \sum_{i=K}^{k-1} \tau_i(B_i + p_{\text{span}}(Q^*) + 1)^2\alpha_i^2 \prod_{j=i+1}^{k-1} (1 - \phi_1\alpha_j)$$

$$\leq \frac{u_m^2}{\ell_m^2} \prod_{j=K}^{k-1} (1 - \phi_1\alpha_j) \mathbb{E}[p_{\text{span}}(Q_K - Q^*)^2]$$

$$+ 70u_m^2\phi_2 \sum_{i=K}^{k-1} \tau_i(B_i + p_{\text{span}}(Q^*) + 1)^2\alpha_i^2 \prod_{j=i+1}^{k-1} (1 - \phi_1\alpha_j)$$

$$\leq \frac{u_m^2}{\ell_m^2}(b_K + p_{\text{span}}(Q^*))^2 \prod_{j=K}^{k-1} (1 - \phi_1\alpha_j)$$

$$+ 70u_m^2\phi_2\tau_k(b_k + p_{\text{span}}(Q^*) + 1)^2 \sum_{i=K}^{k-1} \alpha_i^2 \prod_{j=i+1}^{k-1} (1 - \phi_1\alpha_j),$$

where the last inequality follows from

$$p_{\text{span}}(Q_K - Q^*) \leq p_{\text{span}}(Q_K) + p_{\text{span}}(Q^*) \leq b_K + p_{\text{span}}(Q^*).$$

Before we proceed, we finalize the choices of the tunable parameters $q$ and $\theta$ in the definition of $M(\cdot)$ to make all constants on the right-hand side of the previous inequality explicit. Specifically, by choosing

$$\theta = \left(\frac{1+\beta}{2\beta}\right)^2 - 1, \; q = 2\log(|\mathcal{S}||\mathcal{A}|),$$

we have $u_q = 1$ and $\ell_q = d^{-1/q} = 1/\sqrt{e}$, which further implies

$$u_m^2 = 1 + \theta u_q^2 = 1 + \theta \leq \frac{1}{\beta^2} \leq 4. \qquad \text{(Assume without loss of generality that } \beta > 1/2.)$$

$$\frac{u_m^2}{\ell_m^2} = \frac{1 + \theta u_q^2}{1 + \theta \ell_q^2} = \frac{1 + \theta}{1 + \theta/\sqrt{e}} \leq \sqrt{e} \leq 3,$$

$$\phi_1 = 1 - \frac{\beta u_m}{\ell_m} \geq 1 - \beta\frac{1+\beta}{2\beta} = \frac{1-\beta}{2},$$

$$L = \frac{q-1}{\theta\ell_q^2} \leq \frac{6\log(|\mathcal{S}||\mathcal{A}|)}{1-\beta},$$

$$70u_m^2\phi_2 = \frac{70u_m^2|\mathcal{S}||\mathcal{A}|}{D_{\min}^2}\left(\frac{C_{mc}}{\ell_m^2(1 - \beta u_m/\ell_m)\alpha} + L\right)$$

$$\leq \frac{280|\mathcal{S}||\mathcal{A}|}{D_{\min}^2}\left(\frac{2C}{(1-\rho)(1-\beta)\alpha} + \frac{6\log(|\mathcal{S}||\mathcal{A}|)}{1-\beta}\right)$$

$$\leq \frac{2240C|\mathcal{S}||\mathcal{A}|\log(|\mathcal{S}||\mathcal{A}|)}{(1-\beta)(1-\rho)D_{\min}^2\min(1,\alpha)}.$$

Therefore, we have for all $k \geq K$ that

$$\mathbb{E}[p_{\text{span}}(Q_k - Q^*)^2] \leq 3(b_K + p_{\text{span}}(Q^*))^2 \underbrace{\prod_{j=K}^{k-1}\left(1 - \frac{(1-\beta)\alpha_j}{2}\right)}_{:=E_1}$$

$$+ \frac{2240C|\mathcal{S}||\mathcal{A}|\log(|\mathcal{S}||\mathcal{A}|)}{(1-\beta)(1-\rho)D_{\min}^2\min(1,\alpha)}\tau_k(b_k + p_{\text{span}}(Q^*) + 1)^2 \underbrace{\sum_{i=K}^{k-1}\alpha_i^2\prod_{j=i+1}^{k-1}\left(1 - \frac{(1-\beta)\alpha_j}{2}\right)}.$$

$$(24)$$

It remains to bound the terms $E_1$ and $E_2$. For simplicity of notation, denote $c = (1 - \beta)/2$. Then, since $\alpha_k = \alpha/(k + h)$, we have

$$E_1 = \prod_{j=K}^{k-1} (1 - c\alpha_j)$$

$$\leq \exp\left(-\sum_{j=K}^{k-1} c\alpha_j\right)$$

$$= \exp\left(-\sum_{j=K}^{k-1} \frac{c\alpha}{j + h}\right)$$

$$\leq \exp\left(-\int_K^k \frac{c\alpha}{x + h} dx\right)$$

$$= \exp\left(-c\alpha \log\left(\frac{k + h}{K + h}\right)\right)$$

$$= \left(\frac{K + h}{k + h}\right)^{c\alpha}.$$

Similarly, we also have

$$E_2 = \sum_{i=K}^{k-1} \alpha_i^2 \prod_{j=i+1}^{k-1} (1 - c\alpha_j)$$

$$\leq \sum_{i=K}^{k-1} \alpha_i^2 \prod_{j=i+1}^{k-1} (1 - c\alpha_j)$$

$$\leq \sum_{i=K}^{k-1} \frac{\alpha^2}{(i + h)^2} \left(\frac{i + 1 + h}{k + h}\right)^{c\alpha}$$

$$\leq \frac{4\alpha^2}{(k + h)^{c\alpha}} \sum_{i=K}^{k-1} \frac{1}{(i + 1 + h)^{2-c\alpha}}$$

$$\leq \frac{4\alpha^2}{(k + h)^{c\alpha}} \begin{cases} \dfrac{1}{1 - c\alpha} & c\alpha \in (0, 1), \\ \log(k + h) & c\alpha = 1, \\ \dfrac{(k + 1 + h)^{c\alpha - 1}}{c\alpha - 1} & c\alpha \in (1, \infty). \end{cases}$$

$$\leq \begin{cases} \dfrac{1}{(k + h)^{c\alpha}} \dfrac{4\alpha^2}{1 - c\alpha} & c\alpha \in (0, 1), \\ \dfrac{4\log(k + h)}{c^2(k + h)} & c\alpha = 1, \\ \dfrac{1}{(k + h)} \dfrac{12\alpha^2}{c\alpha - 1} & c\alpha \in (1, \infty). \end{cases}$$

Using the previous two inequalities together in Eq. (24) and recalling that $c = (1 - \beta)/2$, we finally obtain the desired finite-time bound:

(1) When $\alpha(1 - \beta) < 2$, we have

$$\mathbb{E}[p_{\text{span}}(Q_k - Q^*)^2] \leq 3(b_K + p_{\text{span}}(Q^*))^2 \left(\frac{K + h}{k + h}\right)^{\frac{\alpha(1-\beta)}{2}}$$

$$+ \frac{17920\alpha^2 C|\mathcal{S}||\mathcal{A}| \log(|\mathcal{S}||\mathcal{A}|)}{(1 - \beta)(1 - \rho)D_{\min}^2 \min(1, \alpha)(2 - (1 - \beta)\alpha)} \frac{\tau_k(b_k + p_{\text{span}}(Q^*) + 1)^2}{(k + h)^{\alpha(1-\beta)/2}}.$$

(2) When $\alpha(1 - \beta) = 2$, we have

$$\mathbb{E}[p_{\mathrm{span}}(Q_k - Q^*)^2] \leq 3(b_K + p_{\mathrm{span}}(Q^*))^2 \left( \frac{K + h}{k + h} \right)$$

$$+ \frac{35840 C |\mathcal{S}||\mathcal{A}| \log(|\mathcal{S}||\mathcal{A}|)}{(1 - \rho)(1 - \beta)^3 D_{\mathrm{min}}^2} \frac{\tau_k (b_k + p_{\mathrm{span}}(Q^*) + 1)^2 \log(k + h)}{(k + h)}.$$

(3) When $\alpha(1 - \beta) > 2$, we have

$$\mathbb{E}[p_{\mathrm{span}}(Q_k - Q^*)^2] \leq 3(b_K + p_{\mathrm{span}}(Q^*))^2 \left( \frac{K + h}{k + h} \right)^{\frac{\alpha(1 - \beta)}{2}}$$

$$+ \frac{53760 \alpha^2 C |\mathcal{S}||\mathcal{A}| \log(|\mathcal{S}||\mathcal{A}|)}{(1 - \rho)(1 - \beta) D_{\mathrm{min}}^2 ((1 - \beta)\alpha - 2)} \frac{\tau_k (b_k + p_{\mathrm{span}}(Q^*) + 1)^2}{(k + h)}.$$

The proof of Theorem 3.1 is complete.

## A.5 Proof of Corollary 3.1.1

By Jensen's inequality, we have

$$\mathbb{E}[p_{\mathrm{span}}(Q_k - Q^*)^2] \leq \epsilon^2 \quad \Longrightarrow \quad \mathbb{E}[p_{\mathrm{span}}(Q_k - Q^*)] \leq \epsilon.$$

To achieve $\mathbb{E}[p_{\mathrm{span}}(Q_k - Q^*)^2] \leq \epsilon^2$, using Theorem 3.1 with the case $\alpha(1 - \beta) = 2$, we have

$$k = \tilde{\mathcal{O}} \left( \frac{|\mathcal{S}|^3 |\mathcal{A}|^3 p_{\mathrm{span}}(Q^*)^2}{D_{\mathrm{min}}^2 (1 - \beta)^3 (1 - \rho)\epsilon^2} \right).$$

To bound $p_{\mathrm{span}}(Q^*)$ in terms of $\beta$, using the Bellman equation, we have

$$p_{\mathrm{span}}(Q^*) = p_{\mathrm{span}}(\mathcal{H}(Q^*) - r^* e)$$
$$= p_{\mathrm{span}}(\mathcal{H}(Q^*) - \mathcal{H}(0)) + p_{\mathrm{span}}(\mathcal{H}(0))$$
$$\leq \beta p_{\mathrm{span}}(Q^*) + 1,$$

where the last inequality follows from Lemma A.2. Rearranging terms, we obtain $p_{\mathrm{span}}(Q^*) \leq 1/(1 - \beta)$. Therefore, the overall sample complexity is

$$\tilde{\mathcal{O}} \left( \frac{|\mathcal{S}|^3 |\mathcal{A}|^3}{D_{\mathrm{min}}^2 (1 - \beta)^5 (1 - \rho)\epsilon^2} \right).$$

# B Proofs of Technical Lemmas and Propositions in Support of Theorem 3.1

We begin with a summary of notation.

For any $k \geq 1$, let $D_k$, $\bar{D}_k$, and $D$ be $|\mathcal{S}||\mathcal{A}| \times |\mathcal{S}||\mathcal{A}|$ diagonal matrices with diagonal entries $\{(N_k(s,a) + h)/(k + h)\}_{(s,a) \in \mathcal{S} \times \mathcal{A}}$, $\{N_k(s,a)/k\}_{(s,a) \in \mathcal{S} \times \mathcal{A}}$, and $\{\mu(s)\pi(a|s)\}_{(s,a) \in \mathcal{S} \times \mathcal{A}}$, respectively, where we recall that $N_k(s,a) = \sum_{i=1}^{k} \mathbb{1}_{\{(S_i, A_i) = (s,a)\}}$ counts the number of times the state-action pair $(s,a)$ has been visited up to iteration $k$. For simplicity, we will write $D_k(s,a)$ to denote the $(s,a)$-th diagonal entry of $D_k$; similarly for $\bar{D}_k(s,a)$ and $D(s,a)$. Let $D_{\mathrm{min}} = \min_{s,a} \mu(s)\pi(a|s)$, which is strictly positive under Assumption 3.1.

Let $Z_k = (D_k, S_k, A_k, S_{k+1})$ for all $k \geq 1$. Note that $\{Z_k\}$ is a time-inhomogeneous Markov chain with state space denoted by $\mathcal{Z}$. Define $Y_k = (S_k, A_k, S_{k+1})$ for all $k \geq 1$. It is clear that $\{Y_k\}$ is also a Markov chain, with state space denoted by $\mathcal{Y}$. Moreover, under Assumption 3.1, the Markov chain $\{Y_k\}$ admits a unique stationary distribution $\nu \in \Delta(\mathcal{Y})$, which satisfies $\nu(s, a, s') = \mu(s)\pi(a|s)p(s'|s,a)$ for all $y = (s, a, s') \in \mathcal{Y}$.

## B.1 Proof of Lemma A.1

(1) Let $x \in \mathbb{R}^{|\mathcal{S}||\mathcal{A}|}$ be defined as

$$x = F(Q_1, \tilde{D}, y) - F(Q_2, \tilde{D}, y) - (Q_1 - Q_2).$$

It is clear by the definition of $F(\cdot)$ that

$$x(s, a) = \frac{1}{\tilde{D}(s_0, a_0)} \left( \max_{a' \in \mathcal{A}} Q_1(s_1, a') - \max_{a' \in \mathcal{A}} Q_2(s_1, a') - Q_1(s_0, a_0) + Q_2(s_0, a_0) \right)$$

if $(s, a) = (s_0, a_0)$ and $x(s, a) = 0$ otherwise. By the triangle inequality, we have

$$p_{\text{span}}(F(Q_1, \tilde{D}, y) - F(Q_2, \tilde{D}, y)) \le p_{\text{span}}(Q_1 - Q_2) + p_{\text{span}}(x). \tag{25}$$

To bound $p_{\text{span}}(x)$, since $x$ has only one non-zero entry, we have

$$p_{\text{span}}(x) = \frac{1}{2\tilde{D}(s_0, a_0)} \left| \max_{a' \in \mathcal{A}} Q_1(s_1, a') - \max_{a' \in \mathcal{A}} Q_2(s_1, a') - (Q_1(s_0, a_0) - Q_2(s_0, a_0)) \right|. \tag{26}$$

To further bound the absolute value, observe that on the one hand, we have

$$\max_{a' \in \mathcal{A}} Q_1(s_1, a') - \max_{a' \in \mathcal{A}} Q_2(s_1, a') - (Q_1(s_0, a_0) - Q_2(s_0, a_0))$$
$$\le \max_{s', a'} (Q_1(s', a') - Q_2(s', a')) - \min_{s', a'} (Q_1(s', a') - Q_2(s', a'))$$
$$= 2p_{\text{span}}(Q_1 - Q_2).$$

On the other hand, since

$$\max_{a'} Q_1(s_1, a') - \max_{a'} Q_2(s_1, a') = \max_{a'} Q_1(s_1, a') - Q_2(s_1, \underline{a})$$
$$\text{(Denote } \underline{a} \in \arg\max_{a'} Q_2(s_1, a'))$$
$$\ge Q_1(s_1, \underline{a}) - Q_2(s_1, \underline{a})$$
$$\ge \min_{s', a'} (Q_1(s', a') - Q_2(s', a')),$$

we have

$$\max_{a' \in \mathcal{A}} Q_1(s_1, a') - \max_{a' \in \mathcal{A}} Q_2(s_1, a') - (Q_1(s_0, a_0) - Q_2(s_0, a_0))$$
$$\ge \min_{s', a'} (Q_1(s', a') - Q_2(s', a')) - (Q_1(s_0, a_0) - Q_2(s_0, a_0)),$$
$$\ge \min_{s', a'} (Q_1(s', a') - Q_2(s', a')) - \max_{s', a'} (Q_1(s', a') - Q_2(s', a'))$$
$$= -2p_{\text{span}}(Q_1 - Q_2).$$

It follows that

$$\left| \max_{a' \in \mathcal{A}} Q_1(s_1, a') - \max_{a' \in \mathcal{A}} Q_2(s_1, a') - (Q_1(s_0, a_0) - Q_2(s_0, a_0)) \right| \le 2p_{\text{span}}(Q_1 - Q_2).$$

Using the previous inequality in Eq. (26), we obtain

$$p_{\text{span}}(x) \le \frac{1}{\tilde{D}(s_0, a_0)} p_{\text{span}}(Q_1 - Q_2).$$

Combining the previous inequality with Eq. (25), we have

$$p_{\text{span}}(F(Q_1, \tilde{D}, y) - F(Q_2, \tilde{D}, y)) \le p_{\text{span}}(Q_1 - Q_2) + p_{\text{span}}(x)$$
$$\le \left( 1 + \frac{1}{\tilde{D}(s_0, a_0)} \right) p_{\text{span}}(Q_1 - Q_2)$$
$$\le \frac{2}{\tilde{D}(s_0, a_0)} p_{\text{span}}(Q_1 - Q_2),$$

where the last inequality follows from $\tilde{D}(s, a) \in (0, 1)$ for all $(s, a)$.

(2) By Part (1) of this lemma, we have

$$p_{\text{span}}(F(Q, \tilde{D}, y)) \le p_{\text{span}}(F(Q, \tilde{D}, y) - F(0, \tilde{D}, y)) + p_{\text{span}}(F(0, \tilde{D}, y))$$

$$\leq \frac{2}{\tilde{D}(s_0, a_0)} p_{\text{span}}(Q) + p_{\text{span}}(F(0, \tilde{D}, y)).$$

Since the vector $F(0, \tilde{D}, y)$ has only one non-zero entry, i.e., the $(s_0, a_0)$-th entry, which is equal to $\mathcal{R}(s_0, a_0)/\tilde{D}(s_0, a_0)$, we have

$$p_{\text{span}}(F(0, \tilde{D}, y)) = \frac{1}{2\tilde{D}(s_0, a_0)} |\mathcal{R}(s_0, a_0)| \leq \frac{1}{2\tilde{D}(s_0, a_0)}.$$

Combining the previous two inequalities, we have

$$p_{\text{span}}(F(Q, \tilde{D}, y)) \leq \frac{2}{\tilde{D}(s_0, a_0)} p_{\text{span}}(Q) + \frac{1}{2\tilde{D}(s_0, a_0)} \leq \frac{2}{\tilde{D}(s_0, a_0)}(p_{\text{span}}(Q) + 1).$$

(3) For any $(s, a)$, we have

$$\begin{aligned}
&\mathbb{E}_{Y \sim \nu}[[F(Q, D, Y)](s, a)] \\
&= \sum_{s_0, a_0, s_1} \frac{\mu(s_0)\pi(a_0|s_0)p(s_1|s_0, a_0)\mathbb{1}_{\{(s_0, a_0)=(s,a)\}}}{D(s, a)} \\
&\quad \times \left( \mathcal{R}(s_0, a_0) + \max_{a' \in \mathcal{A}} Q(s_1, a') - Q(s_0, a_0) \right) + Q(s, a) \\
&= \sum_{s_1} \frac{D(s, a)p(s_1|s, a)}{D(s, a)} \left( \mathcal{R}(s, a) + \max_{a' \in \mathcal{A}} Q(s_1, a') - Q(s, a) \right) + Q(s, a) \\
&= \sum_{s_1} p(s_1|s, a) \left( \mathcal{R}(s, a) + \max_{a' \in \mathcal{A}} Q(s_1, a') \right) \\
&= [\mathcal{H}(Q)](s, a).
\end{aligned}$$

## B.2 Proof of Lemma A.2

(1) For any $Q_1, Q_2 \in \mathbb{R}^{|\mathcal{S}||\mathcal{A}|}$ and $c \in \mathbb{R}$, we have

$$\begin{aligned}
&|[\mathcal{H}(Q_1)](s, a) - [\mathcal{H}(Q_2)](s, a) - c| \\
&= \left| \mathbb{E}\left[ \max_{a' \in \mathcal{A}}(Q_1(S_2, a') - c) - \max_{a' \in \mathcal{A}} Q_2(S_2, a') \,|dle|\, S_1 = s, A_1 = a \right] \right| \\
&\leq \mathbb{E}\left[ \left| \max_{a' \in \mathcal{A}}(Q_1(S_2, a') - c) - \max_{a' \in \mathcal{A}} Q_2(S_2, a') \right| \,|dle|\, S_1 = s, A_1 = a \right] \\
&\leq \mathbb{E}\left[ \max_{a' \in \mathcal{A}} |Q_1(S_2, a') - Q_2(S_2, a') - c| \,|dle|\, S_1 = s, A_1 = a \right] \\
&\leq \|Q_1 - Q_2 - ce\|_\infty.
\end{aligned}$$

The above inequality implies

$$\|\mathcal{H}(Q_1) - \mathcal{H}(Q_2) - ce\|_\infty \leq \|Q_1 - Q_2 - ce\|_\infty, \quad \forall Q_1, Q_2 \in \mathbb{R}^{|\mathcal{S}||\mathcal{A}|}, \ c \in \mathbb{R}.$$

As a side note, by setting $c = 0$ in the above inequality, we have

$$\|\mathcal{H}(Q_1) - \mathcal{H}(Q_2)\|_\infty \leq \|Q_1 - Q_2\|_\infty, \quad \forall Q_1, Q_2 \in \mathbb{R}^{|\mathcal{S}||\mathcal{A}|}.$$

To show the non-expansiveness property with respect to $p_{\text{span}}(\cdot)$, let $\bar{c} = \arg\min_{c \in \mathbb{R}} \|Q_1 - Q_2 - ce\|_\infty$. Then, we have

$$\begin{aligned}
p_{\text{span}}(\mathcal{H}(Q_1) - \mathcal{H}(Q_2)) &= \min_{c \in \mathbb{R}} \|\mathcal{H}(Q_1) - \mathcal{H}(Q_2) - ce\|_\infty \\
&\leq \|\mathcal{H}(Q_1) - \mathcal{H}(Q_2) - \bar{c}e\|_\infty \\
&\leq \|Q_1 - Q_2 - \bar{c}e\|_\infty \\
&= p_{\text{span}}(Q_1 - Q_2).
\end{aligned}$$

(2) Using Part (1) of this lemma, we have

$$p_{\text{span}}(\mathcal{H}(Q)) \leq p_{\text{span}}(\mathcal{H}(Q) - \mathcal{H}(0)) + p_{\text{span}}(\mathcal{H}(0)) \leq p_{\text{span}}(Q) + p_{\text{span}}(\mathcal{H}(0)).$$

Since

$$p_{\text{span}}(\mathcal{H}(0)) \leq \|\mathcal{H}(0)\|_\infty = \max_{s,a} |\mathcal{R}(s,a)| \leq 1,$$

we have

$$p_{\text{span}}(\mathcal{H}(Q)) \leq p_{\text{span}}(Q) + 1.$$

## B.3 Proof of Lemma A.4

Since $\mathcal{H}(Q^*) - Q^* = r^* e \in \ker(p_{\text{span}})$, we have by Lemma A.3 (4) that

$$\langle \nabla M(Q_k - Q^*), \mathcal{H}(Q_k) - Q_k \rangle = \langle \nabla M(Q_k - Q^*), \mathcal{H}(Q_k) - \mathcal{H}(Q^*) + Q^* - Q_k \rangle$$
$$= \underbrace{\langle \nabla M(Q_k - Q^*), \mathcal{H}(Q_k) - \mathcal{H}(Q^*) \rangle}_{T_{1,1}}$$
$$+ \underbrace{\langle \nabla M(Q_k - Q^*), Q^* - Q_k \rangle}_{T_{1,2}}. \tag{27}$$

Next, we bound the terms $T_{1,1}$ and $T_{1,2}$ on the right-hand side of the previous inequality.

Since $M(Q) = p_m(Q)^2/2$ differentiable, we have

$$\nabla M(Q) = p_m(Q) \nabla p_m(Q), \quad \forall Q \in \mathbb{R}^{|\mathcal{S}||\mathcal{A}|}.$$

It follows that for any $c \in \mathbb{R}$, we have

$$\begin{aligned} T_{1,1} &= \langle \nabla M(Q_k - Q^*), \mathcal{H}(Q_k) - \mathcal{H}(Q^*) + ce \rangle && \text{(Lemma A.3 (4))} \\ &= p_m(Q_k - Q^*) \langle \nabla p_m(Q_k - Q^*), \mathcal{H}(Q_k) - \mathcal{H}(Q^*) - ce \rangle \\ &\leq p_m(Q_k - Q^*) \|\nabla p_m(Q_k - Q^*)\|_{m,*} \|\mathcal{H}(Q_k) - \mathcal{H}(Q^*) - ce\|_m, \end{aligned}$$

where $\| \cdot \|_{m,*}$ is the dual norm of $\| \cdot \|_m$. Since

$$|p_m(Q_1) - p_m(Q_2)| \leq p_m(Q_1 - Q_2) = \min_{c' \in \mathbb{R}} \|Q_1 - Q_2 - c'e\|_m \leq \|Q_1 - Q_2\|_m,$$

the function $p_m(\cdot)$ is 1-Lipschitz continous with respect to $\| \cdot \|_m$. It then follows from [62, Lemma 2.6] that $\|\nabla p_m(Q_k - Q^*)\|_{m,*} \leq 1$. Therefore, we have

$$\begin{aligned} T_{1,1} &\leq p_m(Q_k - Q^*) \|\nabla p_m(Q_k - Q^*)\|_{m,*} \|\mathcal{H}(Q_k) - \mathcal{H}(Q^*) - ce\|_m \\ &\leq p_m(Q_k - Q^*) \|\mathcal{H}(Q_k) - \mathcal{H}(Q^*) - ce\|_m \\ &= p_m(Q_k - Q^*) p_m(\mathcal{H}(Q_k) - \mathcal{H}(Q^*)) \\ &\qquad\qquad \text{(Choosing } c = \arg\min_{c' \in \mathbb{R}} \|\mathcal{H}(Q_k) - \mathcal{H}(Q^*) - c'e\|_m) \\ &\leq \frac{1}{\ell_m} p_m(Q_k - Q^*) p_{\text{span}}(\mathcal{H}(Q_k) - \mathcal{H}(Q^*)) && \text{(Lemma A.3 (3))} \\ &\leq \frac{\beta}{\ell_m} p_m(Q_k - Q^*) p_{\text{span}}(Q_k - Q^*) && \text{(Assumption 2.1)} \\ &\leq \frac{\beta u_m}{\ell_m} p_m(Q_k - Q^*)^2 && \text{(Lemma A.3 (3))} \\ &= \frac{2\beta u_m}{\ell_m} M(Q_k - Q^*), && (28) \end{aligned}$$

where the last inequality follows from Lemma A.3 (2).

Next, we consider the term $T_{1,2}$ on the right-hand side of Eq. (27). Since $p_m(\cdot)$ is a convex function, which follows from

$$p_m(\alpha Q_1 + (1-\alpha)Q_2) \leq p_m(\alpha Q_1) + p_m((1-\alpha)Q_2) = \alpha p_m(Q_1) + (1-\alpha)p_m(Q_2)$$

for any $\alpha \in [0, 1]$ and $Q_1, Q_2 \in \mathbb{R}^{|\mathcal{S}||\mathcal{A}|}$, we have

$$p_m(0) - p_m(Q_k - Q^*) \geq \langle \nabla p_m(Q_k - Q^*), Q^* - Q_k \rangle.$$

It follows that

$$\begin{aligned}
T_{1,2} &= \langle \nabla M(Q_k - Q^*), Q^* - Q_k \rangle \\
&= p_m(Q_k - Q^*) \langle \nabla p_m(Q_k - Q^*), Q^* - Q_k \rangle \\
&\leq - p_m(Q_k - Q^*)^2 \\
&= - 2M(Q_k - Q^*).
\end{aligned} \tag{29}$$

Finally, using Eqs. (28) and (29) together in Eq. (27), we have

$$\begin{aligned}
\langle \nabla M(Q_k - Q^*), \mathcal{H}(Q_k) - Q_k \rangle &\leq T_{1,1} + T_{1,2} \\
&\leq - 2\left(1 - \frac{\beta u_m}{\ell_m}\right) M(Q_k - Q^*) \\
&= - 2\phi_1 M(Q_k - Q^*).
\end{aligned}$$

Taking expectations on both sides of the previous inequality finishes the proof.

## B.4 Proof of Proposition A.1

The following lemma is needed for the proof.

**Lemma B.1.** *The following inequality holds for all $k \geq 1$:*

$$p_{span}(Q_{k+1}) \leq p_{span}(Q_k) + \alpha_k(S_k, A_k).$$

The proof of Lemma B.1 is presented in Appendix B.10.1.

Recursively applying Lemma B.1 and then using the definition of $\alpha_k(S_k, A_k)$, we obtain

$$\begin{aligned}
p_{\text{span}}(Q_k) &\leq p_{\text{span}}(Q_1) + \sum_{i=1}^{k-1} \frac{\alpha}{N_i(S_i, A_i) + h} \\
&= \sum_{i=1}^{k-1} \frac{\alpha}{N_i(S_i, A_i) + h}, , \quad \forall k \geq 1,
\end{aligned}$$

where the last line follows from $Q_1 = 0$. It remains to bound the quantity $\sum_{i=1}^{k-1} \alpha/(N_i(S_i, A_i) + h)$. For any $k \geq 1$, define the set $\mathcal{M}_{k-1}$ as

$$\mathcal{M}_{k-1} = (\mathcal{S} \times \mathcal{A})^{k-1} = \{(s_1, a_1, \ldots, s_{k-1}, a_{k-1}) | s_i \in \mathcal{S}, \, a_i \in \mathcal{A}, \, \forall \, i = 1, 2, \ldots, k-1\}.$$

Then, the following inequality holds with probability one:

$$\sum_{i=1}^{k-1} \frac{\alpha}{N_i(S_i, A_i) + h} \leq \max_{(s_1, a_1, \ldots, s_{k-1}, a_{k-1}) \in \mathcal{M}_{k-1}} \sum_{i=1}^{k-1} \frac{\alpha}{N_i(s_i, a_i) + h}.$$

Next, we identify a scheduling $(s_1, a_1, \ldots, s_{k-1}, a_{k-1})$ that achieves the maximum on the right-hand side of the previous inequality. Let

$$N_{\max} = \max_{(s,a)} \sum_{i=1}^{k-1} \mathbb{1}_{\{(s_i, a_i) = (s,a)\}}, \qquad (\bar{s}, \bar{a}) \in \arg\max_{(s,a)} \sum_{i=1}^{k-1} \mathbb{1}_{\{(s_i, a_i) = (s,a)\}},$$

$$N_{\min} = \min_{(s,a)} \sum_{i=1}^{k-1} \mathbb{1}_{\{(s_i, a_i) = (s,a)\}}, \qquad (\underline{s}, \underline{a}) \in \arg\min_{(s,a)} \sum_{i=1}^{k-1} \mathbb{1}_{\{(s_i, a_i) = (s,a)\}}.$$

We now show that if a scheduling is optimal, it must hold that $N_{\max} - N_{\min} \leq 1$. Suppose, for contradiction, that $N_{\max} - N_{\min} \geq 2$. Consider a modified scheduling identical to the original one

except that, instead of choosing $(\bar{s}, \bar{a})$ at its last occurrence, we select $(\underline{s}, \underline{a})$ instead. Under this modification, the objective increases by exactly

$$\frac{\alpha}{N_{\min} + 1} - \frac{\alpha}{N_{\max}},$$

which is strictly positive since $N_{\max} - N_{\min} \geq 2$, contradicting the optimality of the original scheduling. Therefore, an optimal scheduling must satisfy $N_{\max} - N_{\min} \leq 1$.

Next, we show that all schedulings satisfying $N_{\max} - N_{\min} \leq 1$ must yield the same value. Let $k - 1 = m|\mathcal{S}||\mathcal{A}| + n$, where $m$ is the quotient and $n < |\mathcal{S}||\mathcal{A}|$ is the remainder. The only way to satisfy $N_{\max} - N_{\min} \leq 1$ is for exactly $n$ state-action pairs to be visited $m + 1$ times and the remaining $|\mathcal{S}||\mathcal{A}| - n$ pairs to be visited $m$ times. For any such scheduling $(s_1, a_1, \cdots, s_{k-1}, a_{k-1})$, the corresponding value is

$$\sum_{i=1}^{k-1} \frac{\alpha}{N_i(s_i, a_i) + h} = \left( |\mathcal{S}||\mathcal{A}| \sum_{i=1}^{m} \frac{\alpha}{i + h} + \frac{n\alpha}{m + 1 + h} \right).$$

Therefore, we obtain

$$\sum_{i=1}^{k-1} \frac{\alpha}{N_i(S_i, A_i) + h} \leq \max_{(s_1, a_1, \ldots, s_{k-1}, a_{k-1}) \in \mathcal{M}_k} \sum_{i=1}^{k-1} \frac{\alpha}{N_i(s_i, a_i) + h}$$

$$= \alpha \left( |\mathcal{S}||\mathcal{A}| \sum_{i=1}^{m} \frac{1}{i + h} + \frac{n}{m + 1 + h} \right)$$

$$\leq \alpha |\mathcal{S}||\mathcal{A}| \sum_{i=1}^{m+1} \frac{1}{i + h}$$

$$= \alpha |\mathcal{S}||\mathcal{A}| \sum_{i=1}^{t} \frac{1}{i + h},$$

where $t = \lceil (k - 1)/(|\mathcal{S}||\mathcal{A}|) \rceil$. Since

$$\sum_{i=1}^{t} \frac{1}{i + h} \leq \int_0^t \frac{1}{x + h} \, dx = \log \left( \frac{t + h}{h} \right),$$

we conclude that

$$p_{\text{span}}(Q_k) \leq \alpha |\mathcal{S}||\mathcal{A}| \log \left( \frac{\lceil (k - 1)/(|\mathcal{S}||\mathcal{A}|) \rceil + h}{h} \right).$$

## B.5 Proof of Lemma A.5

Fixing $x \in \mathcal{X}$, let $W_x : \mathcal{X}^k \to \mathbb{R}$ be a function defined as

$$W_x(x_1, x_2, \cdots, x_k) = \frac{\sum_{i=1}^{k} \mathbb{1}_{\{x_i = x\}}}{k}, \quad \forall x_{1,k} \in \mathcal{X}^k,$$

where we use $x_{1,k}$ to denote $(x_1, x_2, \cdots, x_k)$ for simplicity of notation.

It is clear that $W(\cdot)$ satisfies

$$W_x(x_1, x_2, \cdots, x_k) - W_x(y_1, y_2, \cdots, y_k) \leq \sum_{i=1}^{k} \frac{1}{k} \mathbb{1}_{\{x_i \neq y_i\}}, \quad \forall x_{1,k}, y_{1,k} \in \mathcal{X}^k.$$

With the above notation, our goal is to bound $\mathbb{P}\left( |W_x(X_{1,k}) - \nu(x)| \geq \epsilon \right)$. Observe that

$$|W_x(X_{1,k}) - \nu(x)| \leq |W_x(X_{1,k}) - \mathbb{E}[W_x(X_{1,k})]| + |\mathbb{E}[W_x(X_{1,k})] - \nu(x)|.$$

Since

$$|\mathbb{E}[W_x(X_{1,k})] - \nu(x)| = \frac{1}{k} \left| \sum_{i=1}^{k} [p(X_i = x \mid X_1) - \mu(x)] \right|$$

$$\leq \frac{1}{k} \sum_{i=1}^{k} |p(X_i = x \mid X_1) - \mu(x)|$$

$$\leq \frac{1}{k} \sum_{i=1}^{k} 2\tilde{C}\tilde{\rho}^{i-1}$$

$$\leq \frac{2\tilde{C}}{(1-\tilde{\rho})k},$$

we have

$$|W_x(X_{1,k}) - \nu(x)| \leq |W_x(X_{1,k}) - \mathbb{E}[W_x(X_{1,k})]| + \frac{2\tilde{C}}{(1-\tilde{\rho})k}.$$

Therefore, for any $\epsilon \geq \frac{4\tilde{C}}{(1-\tilde{\rho})k}$, applying Corollary 2.11 from [56], we have

$$\mathbb{P}\left(|W_x(X_{1,k}) - \nu(x)| \geq \epsilon\right) \leq \mathbb{P}\left(|W_x(X_{1,k}) - \mathbb{E}[W_x(X_{1,k})]| \geq \epsilon - \frac{2\tilde{C}}{(1-\tilde{\rho})k}\right)$$

$$\leq 2\exp\left(-c'k\left(\epsilon - \frac{2\tilde{C}}{(1-\tilde{\rho})k}\right)^2\right)$$

$$\leq 2\exp\left(-\frac{c'k\epsilon^2}{4}\right),$$

where $c'$ is a constant depending on the mixing time of the Markov chain $\{X_k\}$. The result follows by redefining $c = c'/4$.

## B.6 Proof of Proposition A.2

We begin with the following decomposition:

$$\begin{aligned}
T_2 &= \mathbb{E}[\langle \nabla M(Q_k - Q^*), F(Q_k, D, Y_k) - \mathcal{H}(Q_k)\rangle] \\
&= \underbrace{\mathbb{E}[\langle \nabla M(Q_{k-\tau_k} - Q^*), F(Q_{k-\tau_k}, D, Y_k) - \mathcal{H}(Q_{k-\tau_k})\rangle]}_{:=T_{2,1}} \\
&\quad + \underbrace{\mathbb{E}[\langle \nabla M(Q_{k-\tau_k} - Q^*), F(Q_k, D, Y_k) - F(Q_{k-\tau_k}, D, Y_k) - \mathcal{H}(Q_k) + \mathcal{H}(Q_{k-\tau_k})\rangle]}_{:=T_{2,2}} \\
&\quad + \underbrace{\mathbb{E}[\langle \nabla M(Q_k - Q^*) - \nabla M(Q_{k-\tau_k} - Q^*), F(Q_k, D, Y_k) - \mathcal{H}(Q_k)\rangle]}_{:=T_{2,3}}, \quad (30)
\end{aligned}$$

where we recall that $\tau_k = \min\{t : C\rho^{t-1} \leq \alpha/(k+h)\}$.

### B.6.1 Bounding the Term $T_{2,1}$

We begin with some notation. For any $k \geq 1$, let $\mathcal{F}_k$ be the $\sigma$-algebra generated by the set of random variables $\{Q_1, S_1, A_1, S_2, A_2, \cdots, S_{k-1}, A_{k-1}, S_k\}$. Note that $Q_k$ is measurable with respect to $\mathcal{F}_k$. By the tower property of conditional expectations, we have for any $k \geq \tau_k + 1$ that

$$\begin{aligned}
T_{2,1} &= \mathbb{E}[\langle \nabla M(Q_{k-\tau_k} - Q^*), F(Q_{k-\tau_k}, D, Y_k) - \mathcal{H}(Q_{k-\tau_k})\rangle] \\
&= \mathbb{E}[\langle \nabla M(Q_{k-\tau_k} - Q^*), \mathbb{E}[F(Q_{k-\tau_k}, D, Y_k)|\mathcal{F}_{k-\tau_k}] - \mathcal{H}(Q_{k-\tau_k})\rangle] \\
&\leq L\mathbb{E}[p_{\text{span}}(Q_{k-\tau_k} - Q^*)p_{\text{span}}(\mathbb{E}[F(Q_{k-\tau_k}, D, Y_k)|\mathcal{F}_{k-\tau_k}] - \mathcal{H}(Q_{k-\tau_k}))],
\end{aligned}$$

where the last inequality follows from Lemma A.3 (5).

According to Proposition A.1, we have

$$p_{\text{span}}(Q_{k-\tau_k} - Q^*) \leq p_{\text{span}}(Q_{k-\tau_k}) + p_{\text{span}}(Q^*) \leq b_{k-\tau_k} + p_{\text{span}}(Q^*) \leq b_k + p_{\text{span}}(Q^*).$$

which further implies

$$T_{2,1} \leq L(b_k + p_{\text{span}}(Q^*))\mathbb{E}[p_{\text{span}}(\mathbb{E}[F(Q_{k-\tau_k}, D, Y_k)|\mathcal{F}_{k-\tau_k}] - \mathcal{H}(Q_{k-\tau_k}))]$$

$$\leq L(b_k + p_{\mathrm{span}}(Q^*))\mathbb{E}\left[\sum_{y \in \mathcal{Y}} |p(Y_k = y|Y_{k-\tau_k-1}) - \nu(y)|p_{\mathrm{span}}(F(Q_{k-\tau_k}, D, y))\right],$$

where the last line follows from Lemma A.1 (3) and the triangle inequality.

On the one hand, we have by Lemma A.1 (2) and Proposition A.1 that

$$p_{\mathrm{span}}(F(Q_{k-\tau_k}, D, y)) \leq \frac{2(p_{\mathrm{span}}(Q_{k-\tau_k}) + 1)}{D_{\min}} \leq \frac{2(b_{k-\tau_k} + 1)}{D_{\min}} \leq \frac{2(b_k + 1)}{D_{\min}}.$$

On the other hand, we have by Assumption 3.1 that

$$
\begin{aligned}
\sum_{y \in \mathcal{Y}} |p(Y_k = y|Y_{k-\tau_k-1}) - \nu(y)| &= \sum_{s_0, a_0, s_1} |p(Y_k = (s_0, a_0, s_1)|Y_{k-\tau_k-1}) - \nu(s_0, a_0, s_1)| \\
&= \sum_{s_0, a_0, s_1} |p_\pi(S_k = s_0|S_{k-\tau_k}) - \mu(s_0)|\pi(a_0|s_0)p(s_1|s_0, a_0) \\
&= \sum_{s_0} |p_\pi(S_k = s_0|S_{k-\tau_k}) - \mu(s_0)| \\
&= 2\|p_\pi(S_k = \cdot|S_{k-\tau_k}) - \mu(\cdot)\|_{\mathrm{TV}} \\
&\leq 2C\rho^{\tau_k}.
\end{aligned}
$$

Together, they imply

$$
\begin{aligned}
T_{2,1} &\leq L(b_k + p_{\mathrm{span}}(Q^*))\mathbb{E}\left[\sum_{y \in \mathcal{Y}} |p(Y_k = y|Y_{k-\tau_k-1}) - \nu(y)|p_{\mathrm{span}}(F(Q_{k-\tau_k}, D, y))\right] \\
&\leq \frac{4L(b_k + p_{\mathrm{span}}(Q^*))(b_k + 1)}{D_{\min}}C\rho^{\tau_k} \\
&\leq \frac{4L(b_k + p_{\mathrm{span}}(Q^*))(b_k + 1)}{D_{\min}}\alpha_k \qquad \text{(This follows from the definition of } \tau_k.) \\
&\leq \frac{4L(b_k + p_{\mathrm{span}}(Q^*) + 1)^2}{D_{\min}}\alpha_k. \qquad\qquad\qquad\qquad\qquad\qquad\qquad (31)
\end{aligned}
$$

for all $k \geq \tau_k + 1$.

### B.6.2  Bounding the Term $T_{2,2}$

The following lemma is needed to bound the terms $T_{2,2}$ and $T_{2,3}$ on the right-hand side of Eq. (30). See Appendix B.10.2 for its proof.

**Lemma B.2.** *Let $k_1$ be a positive integer. Then, we have for all $k \geq k_1$ that*

$$p_{span}(Q_k - Q_{k_1}) \leq f\left(\frac{\alpha(k - k_1)}{N_{k_1-1,\min} + 1 + h}\right)(p_{span}(Q_{k_1}) + 1),$$

*where $f(x) := xe^x$ for all $x > 0$ and $N_{k,\min} := \min_{s,a} N_k(s, a)$ for any $k \geq 1$.*

Next, we proceed to bound the term $T_{2,2}$ on the right-hand side of Eq. (30).

Using Lemma A.3 (5), we have

$$
\begin{aligned}
T_{2,2} &= \mathbb{E}[\langle \nabla M(Q_{k-\tau_k} - Q^*), F(Q_k, D, Y_k) - F(Q_{k-\tau_k}, D, Y_k)\rangle] \\
&\quad + \mathbb{E}[\langle \nabla M(Q_{k-\tau_k} - Q^*), \mathcal{H}(Q_{k-\tau_k}) - \mathcal{H}(Q_k)\rangle] \\
&\leq L\mathbb{E}[p_{\mathrm{span}}(Q_{k-\tau_k} - Q^*)p_{\mathrm{span}}(F(Q_k, D, Y_k) - F(Q_{k-\tau_k}, D, Y_k))] \\
&\quad + L\mathbb{E}[p_{\mathrm{span}}(Q_{k-\tau_k} - Q^*)p_{\mathrm{span}}(\mathcal{H}(Q_k) - \mathcal{H}(Q_{k-\tau_k}))] \\
&\leq \left(\frac{2L}{D_{\min}} + L\right)\mathbb{E}[p_{\mathrm{span}}(Q_{k-\tau_k} - Q^*)p_{\mathrm{span}}(Q_k - Q_{k-\tau_k})] \\
&\qquad\qquad\qquad\qquad\qquad\qquad \text{(Lemma A.1 (1) and Lemma A.2 (1))}
\end{aligned}
$$

$$\leq \frac{3L}{D_{\min}} \mathbb{E}[p_{\mathrm{span}}(Q_{k-\tau_k} - Q^*)p_{\mathrm{span}}(Q_k - Q_{k-\tau_k})],$$

where the last inequality follows from $D_{\min} \in (0, 1)$.

On the one hand, we have by Lemma B.2 and Proposition A.1 that

$$p_{\mathrm{span}}(Q_k - Q_{k-\tau_k}) \leq f\left(\frac{\alpha\tau_k}{N_{k-\tau_k-1,\min} + 1 + h}\right)(p_{\mathrm{span}}(Q_{k-\tau_k}) + 1)$$

$$\leq f\left(\frac{\alpha\tau_k}{N_{k-\tau_k-1,\min} + 1 + h}\right)(b_{k-\tau_k} + 1)$$

$$\leq f\left(\frac{\alpha\tau_k}{N_{k-\tau_k-1,\min} + 1 + h}\right)(b_k + 1).$$

On the other hand, we have

$$p_{\mathrm{span}}(Q_{k-\tau_k} - Q^*) \leq p_{\mathrm{span}}(Q_{k-\tau_k}) + p_{\mathrm{span}}(Q^*) \leq b_{k-\tau_k} + p_{\mathrm{span}}(Q^*) \leq b_k + p_{\mathrm{span}}(Q^*).$$

Together, they imply

$$T_{2,2} \leq \frac{3L}{D_{\min}} \mathbb{E}[p_{\mathrm{span}}(Q_{k-\tau_k} - Q^*)p_{\mathrm{span}}(Q_k - Q_{k-\tau_k})]$$

$$\leq \frac{3L(b_k + 1)(b_k + p_{\mathrm{span}}(Q^*))}{D_{\min}} \mathbb{E}\left[f\left(\frac{\alpha\tau_k}{N_{k-\tau_k-1,\min} + 1 + h}\right)\right]$$

$$\leq \frac{3L(b_k + p_{\mathrm{span}}(Q^*) + 1)^2}{D_{\min}} \mathbb{E}\left[f\left(\frac{\alpha\tau_k}{N_{k-\tau_k-1,\min} + 1 + h}\right)\right]. \tag{32}$$

It remains to bound the term $\mathbb{E}[f(\alpha\tau_k/(N_{k-\tau_k-1,\min} + 1 + h)]$.

Using the definition of $f(\cdot)$ (cf. Lemma B.2), we have

$$\mathbb{E}\left[f\left(\frac{\alpha\tau_k}{N_{k-\tau_k-1,\min} + 1 + h}\right)\right]$$

$$= \mathbb{E}\left[\frac{\alpha\tau_k}{N_{k-\tau_k-1,\min} + 1 + h} \exp\left(\frac{\alpha\tau_k}{N_{k-\tau_k-1,\min} + 1 + h}\right)\right]$$

$$= \mathbb{E}\left[\max_{s,a}\left\{\frac{\alpha\tau_k}{N_{k-\tau_k-1}(s,a) + 1 + h} \exp\left(\frac{\alpha\tau_k}{N_{k-\tau_k-1}(s,a) + 1 + h}\right)\right\}\right]$$
$$(f(x) = xe^x \text{ is an increasing function of } x \text{ on } [0, \infty).)$$

$$\leq \sum_{s,a} \mathbb{E}\left[\frac{\alpha\tau_k}{N_{k-\tau_k-1}(s,a) + 1 + h} \exp\left(\frac{\alpha\tau_k}{N_{k-\tau_k-1}(s,a) + 1 + h}\right)\right]$$

$$= \sum_{s,a} \mathbb{E}\left[\frac{\alpha\tau_k}{(k - \tau_k - 1)\bar{D}_{k-\tau_k-1}(s,a) + 1 + h} \exp\left(\frac{\alpha\tau_k}{(k - \tau_k - 1)\bar{D}_{k-\tau_k-1}(s,a) + 1 + h}\right)\right],$$

where the last line follows from our notation $\bar{D}_k(s,a) = N_k(s,a)/k$.

To proceed, for any $\delta > 0$ and $(s,a)$, let

$$E_\delta(s,a) = \{|\bar{D}_{k-\tau_k-1}(s,a) - D(s,a)| \leq \delta D(s,a)\}$$

and let $E_\delta^c(s,a)$ be the complement of the event $E_\delta(s,a)$. Note that on the event $E_\delta(s,a)$, we have

$$\bar{D}_{k-\tau_k-1}(s,a) \geq (1 - \delta)D(s,a),$$

while on the event $E_\delta^c(s,a)$, we have $\bar{D}_{k-\tau_k-1}(s,a) \geq 0$. Therefore, we obtain

$$\mathbb{E}\left[f\left(\frac{\alpha\tau_k}{N_{k-\tau_k-1,\min} + 1 + h}\right)\right]$$

$$\leq \sum_{s,a} \mathbb{E}\left[\frac{\alpha\tau_k}{(k - \tau_k - 1)\bar{D}_{k-\tau_k-1}(s,a) + 1 + h} \exp\left(\frac{\alpha\tau_k}{(k - \tau_k - 1)\bar{D}_{k-\tau_k-1}(s,a) + 1 + h}\right)\right]$$

$$= \sum_{s,a} \mathbb{E}\left[\frac{\alpha\tau_k(\mathbb{1}_{\{E_\delta(s,a)\}} + \mathbb{1}_{\{E_\delta^c(s,a)\}})}{(k-\tau_k-1)\bar{D}_{k-\tau_k-1}(s,a)+1+h} \exp\left(\frac{\alpha\tau_k}{(k-\tau_k-1)\bar{D}_{k-\tau_k-1}(s,a)+1+h}\right)\right]$$

$$\leq \sum_{s,a} \frac{\alpha\tau_k}{(k-\tau_k-1)(1-\delta)D(s,a)+1+h} \exp\left(\frac{\alpha\tau_k}{(k-\tau_k-1)(1-\delta)D(s,a)+1+h)}\right)$$

$$+ \frac{\alpha\tau_k}{h+1} \exp\left(\frac{\alpha\tau_k}{h+1}\right) \sum_{s,a} \mathbb{P}(E_\delta^c(s,a)).$$

To proceed, let $K'$ be such that $k \geq 2\tau_k$ for all $k \geq K'$, which is always possible since $\tau_k = \min\{t : C\rho^{t-1} \leq \alpha_k\} = \mathcal{O}(\log(k))$. Then, we have

$$\frac{\alpha\tau_k}{(k-\tau_k-1)(1-\delta)D(s,a)+1+h} \leq \frac{\alpha\tau_k}{(k-\tau_k-1)(1-\delta)D(s,a)+(1+h)(1-\delta)D(s,a)}$$

$$= \frac{\alpha}{(1-\delta)D(s,a)} \frac{\tau_k}{k+h-\tau_k}$$

$$\leq \frac{\alpha}{(1-\delta)D(s,a)} \frac{2\tau_k}{k+h}$$

$$\leq \frac{\alpha}{(1-\delta)D_{\min}} \frac{2\tau_k}{k+h},$$

which further implies

$$\mathbb{E}\left[f\left(\frac{\alpha\tau_k}{N_{k-\tau_k-1,\min}+1+h}\right)\right] \leq \frac{2\alpha\tau_k|\mathcal{S}||\mathcal{A}|}{(k+h)(1-\delta)D_{\min}} \exp\left(\frac{2\alpha\tau_k}{(k+h)(1-\delta)D_{\min}}\right)$$

$$+ \frac{\alpha\tau_k}{h+1} \exp\left(\frac{\alpha\tau_k}{h+1}\right) \sum_{s,a} \mathbb{P}(E_\delta^c(s,a))$$

because $f(x) = xe^x$ is an increasing function of $x$ on $[0, \infty)$.

To bound $\mathbb{P}(E_\delta^c(s,a))$, we use the Markov chain concentration inequality stated in Lemma A.5. As long as

$$\delta \geq \frac{4C}{D_{\min}(1-\rho)(k-\tau_k-1)}, \tag{33}$$

we have

$$\mathbb{P}(E_\delta^c(s,a)) = \mathbb{P}\left(|\bar{D}_{k-\tau_k-1}(s,a) - D(s,a)| > \delta D(s,a)\right)$$

$$\leq 2\exp(-c_{mc}(k-\tau_k-1)\delta^2 D(s,a)^2),$$

where $c_{mc}$ is a constant depending on the mixing time of the Markov chain $\{S_k\}$ induced by $\pi$. It follows that

$$\mathbb{E}\left[f\left(\frac{\alpha\tau_k}{N_{k-\tau_k-1,\min}+1+h}\right)\right]$$

$$\leq \frac{2\alpha\tau_k|\mathcal{S}||\mathcal{A}|}{(k+h)(1-\delta)D_{\min}} \exp\left(\frac{2\alpha\tau_k}{(k+h)(1-\delta)D_{\min}}\right)$$

$$+ \frac{2\alpha\tau_k}{h+1} \exp\left(\frac{\alpha\tau_k}{h+1}\right) \sum_{s,a} \exp(-c_{mc}(k-\tau_k-1)\delta^2 D(s,a)^2)$$

$$\leq \frac{2\alpha\tau_k|\mathcal{S}||\mathcal{A}|}{(k+h)(1-\delta)D_{\min}} \exp\left(\frac{2\alpha\tau_k}{(k+h)(1-\delta)D_{\min}}\right)$$

$$+ \frac{2\alpha\tau_k|\mathcal{S}||\mathcal{A}|}{h+1} \exp\left(\frac{\alpha\tau_k}{h+1}\right) \exp(-c_{mc}(k-\tau_k-1)\delta^2 D_{\min}^2).$$

The next step is to choose $\delta$ appropriately. Specifically, we want to choose $\delta$ such that

$$\frac{2\alpha\tau_k|\mathcal{S}||\mathcal{A}|}{h+1} \exp\left(\frac{\alpha\tau_k}{h+1}\right) \exp(-c_{mc}(k-\tau_k-1)\delta^2 D_{\min}^2) = \frac{\alpha\tau_k|\mathcal{S}||\mathcal{A}|}{(k+h)D_{\min}},$$

which is equivalent to

$$\delta = \sqrt{\frac{1}{c_{mc}(k - \tau_k - 1)\,D_{\min}^2}\left[\log\left(\frac{2(k+h)D_{\min}}{h+1}\right) + \frac{\alpha\tau_k}{h+1}\right]}. \tag{34}$$

In view of Eq. (33) on the lower bound of $\delta$, let $K''$ be such that the following inequality holds for all $k \geq K''$:

$$\sqrt{\frac{1}{c_{mc}(k - \tau_k - 1)\,D_{\min}^2}\left[\log\left(\frac{2(k+h)D_{\min}}{h+1}\right) + \frac{\alpha\tau_k}{h+1}\right]} \geq \frac{4C}{D_{\min}(1-\rho)(k - \tau_k - 1)},$$

Then, for all $k \geq K''$, choosing $\delta$ according to Eq. (34), we have

$$\mathbb{E}\left[f\left(\frac{\alpha\tau_k}{N_{k-\tau_k-1,\min} + 1 + h}\right)\right] \leq \frac{2\alpha\tau_k|\mathcal{S}||\mathcal{A}|}{(k+h)(1-\delta)D_{\min}}\exp\left(\frac{2\alpha\tau_k}{(k+h)(1-\delta)D_{\min}}\right) + \frac{\alpha\tau_k|\mathcal{S}||\mathcal{A}|}{(k+h)D_{\min}}.$$

Since $\tau_k = \min\{t : C\rho^{t-1} \leq \alpha_k\} = \mathcal{O}(\log(k))$, there exists $K''' > 0$ such that the following inequality holds for all $k \geq K'''$:

$$\frac{2\alpha\tau_k|\mathcal{S}||\mathcal{A}|}{(k+h)(1-\delta)D_{\min}}\exp\left(\frac{2\alpha\tau_k}{(k+h)(1-\delta)D_{\min}}\right) \leq \frac{3\alpha\tau_k|\mathcal{S}||\mathcal{A}|}{(k+h)D_{\min}}.$$

Therefore, when $k \geq K'''$, we have

$$\mathbb{E}\left[f\left(\frac{\alpha\tau_k}{N_{k-\tau_k-1,\min} + 1 + h}\right)\right] \leq \frac{4\alpha\tau_k|\mathcal{S}||\mathcal{A}|}{(k+h)D_{\min}} = \frac{4\tau_k|\mathcal{S}||\mathcal{A}|}{D_{\min}}\alpha_k. \tag{35}$$

Finally, using the previous inequality in Eq. (32), we have for any $k \geq K_2 := \max(K', K'', K''')$ that

$$\begin{aligned}
T_{2,2} &\leq \frac{3L(b_k + p_{\mathrm{span}}(Q^*) + 1)^2}{D_{\min}}\mathbb{E}\left[f\left(\frac{\alpha\tau_k}{N_{k-\tau_k-1,\min} + 1 + h}\right)\right] \\
&\leq \frac{3L(b_k + p_{\mathrm{span}}(Q^*) + 1)^2}{D_{\min}}\frac{4\tau_k|\mathcal{S}||\mathcal{A}|}{D_{\min}}\alpha_k \\
&= \frac{12\tau_k L|\mathcal{S}||\mathcal{A}|(b_k + p_{\mathrm{span}}(Q^*) + 1)^2}{D_{\min}^2}\alpha_k.
\end{aligned} \tag{36}$$

### B.6.3  Bounding the Term $T_{2,3}$

Using Lemma A.3 (5), we have

$$\begin{aligned}
T_{2,3} &= \mathbb{E}[\langle \nabla M(Q_k - Q^*) - \nabla M(Q_{k-\tau_k} - Q^*), F(Q_k, D, Y_k) - \mathcal{H}(Q_k)\rangle] \\
&\leq L\mathbb{E}[p_{\mathrm{span}}(Q_k - Q_{k-\tau_k})p_{\mathrm{span}}(F(Q_k, D, Y_k) - \mathcal{H}(Q_k))].
\end{aligned}$$

On the one hand, we have by Lemma B.2 and Proposition A.1 that

$$\begin{aligned}
p_{\mathrm{span}}(Q_k - Q_{k-\tau_k}) &\leq f\left(\frac{\alpha\tau_k}{N_{k-\tau_k-1,\min} + 1 + h}\right)(p_{\mathrm{span}}(Q_{k-\tau_k}) + 1) \\
&\leq f\left(\frac{\alpha\tau_k}{N_{k-\tau_k-1,\min} + 1 + h}\right)(b_{k-\tau_k} + 1) \\
&\leq f\left(\frac{\alpha\tau_k}{N_{k-\tau_k-1,\min} + 1 + h}\right)(b_k + 1).
\end{aligned}$$

On the other hand, we have by Lemma A.1 (2), Lemma A.2 (2), and Proposition A.1 that

$$\begin{aligned}
p_{\mathrm{span}}(F(Q_k, D, Y_k) - \mathcal{H}(Q_k)) &\leq p_{\mathrm{span}}(F(Q_k, D, Y_k)) + p_{\mathrm{span}}(\mathcal{H}(Q_k)) \\
&\leq \frac{2(p_{\mathrm{span}}(Q_k) + 1)}{D_{\min}} + p_{\mathrm{span}}(Q_k) + 1
\end{aligned}$$

$$\leq \frac{3(b_k + 1)}{D_{\min}}.$$

Together, they imply

$$T_{2,3} \leq L\mathbb{E}[p_{\text{span}}(Q_k - Q_{k-\tau_k})p_{\text{span}}(F(Q_k, D, Y_k) - \mathcal{H}(Q_k))]$$

$$\leq \frac{3L(b_k + 1)^2}{D_{\min}}\mathbb{E}\left[f\left(\frac{\alpha\tau_k}{N_{k-\tau_k-1,\min} + 1 + h}\right)\right]$$

$$\leq \frac{3L(b_k + p_{\text{span}}(Q^*) + 1)^2}{D_{\min}}\mathbb{E}\left[f\left(\frac{\alpha\tau_k}{N_{k-\tau_k-1,\min} + 1 + h}\right)\right].$$

Recall that we have already shown in the previous section (cf. Eq. (35)) that

$$\mathbb{E}\left[f\left(\frac{\alpha\tau_k}{N_{k-\tau_k-1,\min} + 1 + h}\right)\right] \leq \frac{4\tau_k|\mathcal{S}||\mathcal{A}|}{D_{\min}}\alpha_k, \quad \forall\, k \geq K_2.$$

Therefore, we have for any $k \geq K_2$ that

$$T_{2,3} \leq \frac{12\tau_k L|\mathcal{S}||\mathcal{A}|(b_k + p_{\text{span}}(Q^*) + 1)^2}{D_{\min}^2}\alpha_k. \tag{37}$$

### B.6.4 Combining Everything Together

Finally, using the bounds we obtained for the terms $T_{2,1}$ (cf. Eq. (31) from Appendix B.6.1), $T_{2,2}$ (cf. Eq. (36) from Appendix B.6.2), and $T_{2,3}$ (cf. Eq. (37) from Appendix B.6.3) altogether in Eq. (30), we have for all $k \geq K_2$ that

$$T_2 \leq \frac{4L(b_k + p_{\text{span}}(Q^*) + 1)^2}{D_{\min}}\alpha_k + \frac{12\tau_k L|\mathcal{S}||\mathcal{A}|(b_k + p_{\text{span}}(Q^*) + 1)^2}{D_{\min}^2}\alpha_k$$

$$+ \frac{12\tau_k L|\mathcal{S}||\mathcal{A}|(b_k + p_{\text{span}}(Q^*) + 1)^2}{D_{\min}^2}\alpha_k$$

$$\leq \frac{28\tau_k L|\mathcal{S}||\mathcal{A}|(b_k + p_{\text{span}}(Q^*) + 1)^2}{D_{\min}^2}\alpha_k.$$

The proof of Proposition A.2 is complete.

### B.7 Proof of Proposition A.3

Using Lemma A.3 (3) and (4), we have for any $c \in \mathbb{R}$ that

$$\langle \nabla M(Q_k - Q^*), F(Q_k, D_k, Y_k) - F(Q_k, D, Y_k)\rangle$$

$$= \langle \nabla M(Q_k - Q^*), F(Q_k, D_k, Y_k) - F(Q_k, D, Y_k) - ce\rangle$$

$$= p_m(Q_k - Q^*)\langle \nabla p_m(Q_k - Q^*), F(Q_k, D_k, Y_k) - F(Q_k, D, Y_k) - ce\rangle$$

$$\leq p_m(Q_k - Q^*)\|\nabla p_m(Q_k - Q^*)\|_{m,*}\|F(Q_k, D_k, Y_k) - F(Q_k, D, Y_k) - ce\|_m.$$

By choosing

$$c = \arg\min_{c' \in \mathbb{R}} \|F(Q_k, D_k, Y_k) - F(Q_k, D, Y_k) - c'e\|_m,$$

we have by Lemma A.3 (2) that

$$\langle \nabla M(Q_k - Q^*), F(Q_k, D_k, Y_k) - F(Q_k, D, Y_k)\rangle$$

$$\leq p_m(Q_k - Q^*)\|\nabla p_m(Q_k - Q^*)\|_{m,*}p_m(F(Q_k, D_k, Y_k) - F(Q_k, D, Y_k)).$$

Recall that we have shown in the proof of Lemma A.4 that $\|\nabla p_m(Q_k - Q^*)\|_{m,*} \leq 1$. Therefore, we have

$$\langle \nabla M(Q_k - Q^*), F(Q_k, D_k, Y_k) - F(Q_k, D, Y_k)\rangle$$

$$\leq p_m(Q_k - Q^*)p_m(F(Q_k, D_k, Y_k) - F(Q_k, D, Y_k))$$

$$\leq \frac{1}{\ell_m}p_m(Q_k - Q^*)p_{\text{span}}(F(Q_k, D_k, Y_k) - F(Q_k, D, Y_k)) \qquad \text{(Lemma A.3 (3))}$$

$$\leq \frac{1}{2c'\ell_m}p_m^2(Q_k - Q^*) + \frac{c'}{2\ell_m}p_{\text{span}}(F(Q_k, D_k, Y_k) - F(Q_k, D, Y_k))^2$$
$$\text{(This follows from Cauchy - Schwarz inequality, where } c' > 0 \text{ can be arbitrary.)}$$
$$= \frac{1}{c'\ell_m}M(Q_k - Q^*) + \frac{c'}{2\ell_m}p_{\text{span}}(F(Q_k, D_k, Y_k) - F(Q_k, D, Y_k))^2,$$

where the last line follows from $M(Q) = p_m(Q)^2/2$ (cf. Lemma A.3 (2)). By choosing

$$c' = \frac{1}{\ell_m(1 - \beta u_m/\ell_m)},$$

we have from the previous inequality that

$$\langle \nabla M(Q_k - Q^*), F(Q_k, D_k, Y_k) - F(Q_k, D, Y_k) \rangle$$
$$\leq \frac{1}{c'\ell_m}M(Q_k - Q^*) + \frac{c'}{2\ell_m}p_{\text{span}}(F(Q_k, D_k, Y_k) - F(Q_k, D, Y_k))^2$$
$$= \left(1 - \frac{\beta u_m}{\ell_m}\right)M(Q_k - Q^*) + \frac{p_{\text{span}}(F(Q_k, D_k, Y_k) - F(Q_k, D, Y_k))^2}{2\ell_m^2(1 - \beta u_m/\ell_m)}.$$

To proceed, observe that the vector $F(Q_k, D_k, Y_k) - F(Q_k, D, Y_k)$ has only one non-zero entry, i.e., the $(S_k, A_k)$-th one. Therefore, we have by the definition of $p_{\text{span}}(\cdot)$ that

$$p_{\text{span}}(F(Q_k, D_k, Y_k) - F(Q_k, D, Y_k))^2$$
$$= \frac{1}{4}\left(\frac{1}{D_k(S_k, A_k)} - \frac{1}{D(S_k, A_k)}\right)^2 \left(\mathcal{R}(S_k, A_k) + \max_{a' \in \mathcal{A}} Q_k(S_{k+1}, a') - Q_k(S_k, A_k)\right)^2$$
$$\leq \frac{1}{4}\left(\frac{1}{D_k(S_k, A_k)} - \frac{1}{D(S_k, A_k)}\right)^2 (1 + 2p_{\text{span}}(Q_k))^2$$
$$\text{(Definition of } p_{\text{span}}(\cdot) \text{ and } \max_{s,a}|\mathcal{R}(s, a)| \leq 1)$$
$$\leq (1 + b_k)^2\left(\frac{1}{D_k(S_k, A_k)} - \frac{1}{D(S_k, A_k)}\right)^2,$$

where the last line follows from Proposition A.1.

Combining the previous two inequalities together, we have

$$T_3 = \mathbb{E}\left[\langle \nabla M(Q_k - Q^*), F(Q_k, D_k, Y_k) - F(Q_k, D, Y_k) \rangle\right]$$
$$\leq \left(1 - \frac{\beta u_m}{\ell_m}\right)\mathbb{E}\left[M(Q_k - Q^*)\right] + \frac{\mathbb{E}\left[p_{\text{span}}(F(Q_k, D_k, Y_k) - F(Q_k, D, Y_k))^2\right]}{2\ell_m^2(1 - \beta u_m/\ell_m)}$$
$$\leq \left(1 - \frac{\beta u_m}{\ell_m}\right)\mathbb{E}\left[M(Q_k - Q^*)\right] + \frac{(1 + b_k)^2\mathbb{E}\left[\left(\frac{1}{D_k(S_k, A_k)} - \frac{1}{D(S_k, A_k)}\right)^2\right]}{2\ell_m^2(1 - \beta u_m/\ell_m)}. \qquad (38)$$

It remains to bound $\mathbb{E}[(1/D_k(S_k, A_k) - 1/D(S_k, A_k))^2]$, which is highly nontrivial due to the following reasons: (1) for any fixed $(s, a)$, $D_k(s, a)$ depends on the entire history of the Markov chain induced by the behavior policy; (2) $D_k(s, a)$ is correlated with $(S_k, A_k)$; and (3) the appearance of random variables in the denominators of fractions destroys linearity.

For any $\tilde{\tau} \leq k - 1$, we have

$$\mathbb{E}\left[\left(\frac{1}{D_k(S_k, A_k)} - \frac{1}{D(S_k, A_k)}\right)^2\right]$$
$$= \mathbb{E}\left[\frac{(D_k(S_k, A_k) - D(S_k, A_k))^2}{D_k(S_k, A_k)^2 D(S_k, A_k)^2}\right]$$
$$= \mathbb{E}\left[\frac{(N_k(S_k, A_k) + h - (k + h)D(S_k, A_k))^2}{(N_k(S_k, A_k) + h)^2 D(S_k, A_k)^2}\right] \quad (D_k(S_k, A_k) = (N_k(S_k, A_k) + h)/(k + h))$$
$$= \mathbb{E}\left[\frac{(N_{k-1}(S_k, A_k) + 1 + h - (k + h)D(S_k, A_k))^2}{(N_{k-1}(S_k, A_k) + 1 + h)^2 D(S_k, A_k)^2}\right] \quad ((S_k, A_k) \text{ is visited at time step } k)$$

$$= \sum_{s,a} \mathbb{E}\left[\mathbb{1}_{\{(S_k, A_k)=(s,a)\}} \frac{(N_{k-1}(s,a)+1+h-(k+h)D(s,a))^2}{(N_{k-1}(s,a)+1+h)^2 D(s,a)^2}\right]$$

$$\leq \sum_{s,a} \mathbb{E}\left[\mathbb{1}_{\{(S_k, A_k)=(s,a)\}} \frac{(N_{k-1}(s,a)+1+h-(k+h)D(s,a))^2}{(N_{k-\tilde{\tau}-1}(s,a)+1+h)^2 D(s,a)^2}\right]$$

$$(N_{k_1}(s,a) \leq N_{k_2}(s,a) \text{ for any } k_1 \leq k_2.)$$

$$= \sum_{s,a} \mathbb{E}\left[\mathbb{1}_{\{(S_k, A_k)=(s,a)\}} \frac{(A_{k,\tilde{\tau}} + B_{k,\tilde{\tau}} + C_{k,\tilde{\tau}})^2}{(N_{k-\tilde{\tau}-1}(s,a)+1+h)^2 D(s,a)^2}\right],$$

where

$$A_{k,\tilde{\tau}} := N_{k-\tilde{\tau}-1}(s,a) - (k-\tilde{\tau}-1)D(s,a) = (k-\tilde{\tau}-1)(\bar{D}_{k-\tilde{\tau}-1}(s,a) - D(s,a)),$$
$$B_{k,\tilde{\tau}} := N_{k-1}(s,a) - N_{k-\tilde{\tau}-1}(s,a) - \tilde{\tau}D(s,a), \text{ which satisfies } |B_{k,\tilde{\tau}}| \leq \tilde{\tau},$$
$$C_{k,\tilde{\tau}} := (1+h)(1-D(s,a)), \text{ which satisfies } |C_{k,\tilde{\tau}}| \leq (1+h).$$

It follows that

$$\mathbb{E}\left[\left(\frac{1}{D_k(S_k, A_k)} - \frac{1}{D(S_k, A_k)}\right)^2\right]$$

$$\leq \sum_{s,a} \mathbb{E}\left[\mathbb{1}_{\{(S_k, A_k)=(s,a)\}} \frac{(A_{k,\tilde{\tau}} + B_{k,\tilde{\tau}} + C_{k,\tilde{\tau}})^2}{(N_{k-\tilde{\tau}-1}(s,a)+1+h)^2 D(s,a)^2}\right]$$

$$\leq \sum_{s,a} \mathbb{E}\left[\mathbb{1}_{\{(S_k, A_k)=(s,a)\}} \frac{3(A_{k,\tilde{\tau}}^2 + \tilde{\tau}^2 + (h+1)^2)}{(N_{k-\tilde{\tau}-1}(s,a)+1+h)^2 D(s,a)^2}\right]$$

$$((a+b+c)^2 \leq 3(a^2+b^2+c^2) \text{ for any } a,b,c \in \mathbb{R}.)$$

$$= \sum_{s,a} \mathbb{E}\left[\mathbb{P}(S_k = s, A_k = a|S_{k-\tilde{\tau}-1}, A_{k-\tilde{\tau}-1}) \frac{3(A_{k,\tilde{\tau}}^2 + \tilde{\tau}^2 + (h+1)^2)}{(N_{k-\tilde{\tau}-1}(s,a)+1+h)^2 D(s,a)^2}\right],$$

where the last line follows from the tower property of conditional expectations and the Markov property.

Observe that

$$\mathbb{P}(S_k = s, A_k = a|S_{k-\tilde{\tau}-1}, A_{k-\tilde{\tau}-1})$$
$$\leq |\mathbb{P}(S_k = s, A_k = a|S_{k-\tilde{\tau}-1}, A_{k-\tilde{\tau}-1}) - D(s,a)| + D(s,a)$$
$$= \sum_{s'} |\mathbb{P}(S_{k-\tilde{\tau}} = s'|S_{k-\tilde{\tau}-1}, A_{k-\tilde{\tau}-1})p_\pi(S_k = s|S_{k-\tilde{\tau}} = s')\pi(a|s) - \mu(s)\pi(a|s)| + D(s,a)$$
$$\leq \max_{s'} |p_\pi(S_k = s|S_{k-\tilde{\tau}} = s') - \mu(s)| + D(s,a)$$
$$\leq \max_{s'} \sum_s |p_\pi(S_k = s|S_{k-\tilde{\tau}} = s') - \mu(s)| + D(s,a)$$
$$\leq 2C\rho^{\tilde{\tau}} + D(s,a),$$

where the last inequality follows from Assumption 3.1. By choosing

$$\tilde{\tau} = \min\{t : 2C\rho^t \leq D_{\min}\},$$

we have

$$\mathbb{P}(S_k = s, A_k = a|S_{k-\tilde{\tau}-1}, A_{k-\tilde{\tau}-1}) \leq 2D(s,a). \tag{39}$$

It follows that

$$\mathbb{E}\left[\left(\frac{1}{D_k(S_k, A_k)} - \frac{1}{D(S_k, A_k)}\right)^2\right]$$

$$\leq 2 \sum_{s,a} D(s,a)\mathbb{E}\left[\frac{3(A_{k,\tilde{\tau}}^2 + \tilde{\tau}^2 + (h+1)^2)}{(N_{k-\tilde{\tau}-1}(s,a)+1+h)^2 D(s,a)^2}\right]$$

$$= 2\sum_{s,a} D(s,a)\mathbb{E}\left[\frac{3(k-\tilde{\tau}-1)^2(\bar{D}_{k-\tilde{\tau}-1}(s,a) - D(s,a))^2}{((k-\tilde{\tau}-1)\bar{D}_{k-\tilde{\tau}-1}(s,a) + 1 + h)^2 D(s,a)^2}\right]$$

$$+ 2\sum_{s,a} D(s,a)\mathbb{E}\left[\frac{3(\tilde{\tau}^2 + (h+1)^2)}{((k-\tilde{\tau}-1)\bar{D}_{k-\tilde{\tau}-1}(s,a) + 1 + h)^2 D(s,a)^2}\right].$$

To proceed, for any $\delta \in (0,1)$ and $(s,a)$, let $E_\delta(s,a) = \{|\bar{D}_{k-\tilde{\tau}-1}(s,a) - D(s,a)| \le \delta D(s,a)\}$ and let $E_\delta^c(s,a)$ be the complement of event $E_\delta(s,a)$. Note that on the event $E_\delta(s,a)$, we have

$$(1-\delta)D(s,a) \le \bar{D}_{k-\tilde{\tau}-1}(s,a) \le (1+\delta)D(s,a),$$

while on the event $E_\delta^c(s,a)$, we trivially have $\bar{D}_{k-1}(s,a) \ge 0$. It follows that

$$\mathbb{E}\left[\left(\frac{1}{D_k(S_k,A_k)} - \frac{1}{D(S_k,A_k)}\right)^2\right]$$

$$\le 2\sum_{s,a} D(s,a)\mathbb{E}\left[\frac{3(k-\tilde{\tau}-1)^2(\bar{D}_{k-\tilde{\tau}-1}(s,a) - D(s,a))^2(\mathbb{1}_{E_\delta(s,a)} + \mathbb{1}_{E_\delta(s,a)^c})}{((k-\tilde{\tau}-1)\bar{D}_{k-\tilde{\tau}-1}(s,a) + 1 + h)^2 D(s,a)^2}\right]$$

$$+ 2\sum_{s,a} D(s,a)\mathbb{E}\left[\frac{3(\tilde{\tau}^2 + (h+1)^2)(\mathbb{1}_{E_\delta(s,a)} + \mathbb{1}_{E_\delta(s,a)^c})}{((k-\tilde{\tau}-1)\bar{D}_{k-\tilde{\tau}-1}(s,a) + 1 + h)^2 D(s,a)^2}\right]$$

$$\le 2\sum_{s,a} \frac{3(k-\tilde{\tau}-1)^2}{((k-\tilde{\tau}-1)(1-\delta)D(s,a) + 1 + h)^2 D(s,a)}\mathbb{E}\left[(\bar{D}_{k-\tilde{\tau}-1}(s,a) - D(s,a))^2\right]$$

$$+ 2\sum_{s,a} \frac{3(k-\tilde{\tau}-1)^2}{(1+h)^2 D(s,a)}\mathbb{P}(E_\delta^c(s,a))$$

$$+ 2\sum_{s,a} \frac{3(\tilde{\tau}^2 + (h+1)^2)}{((k-\tilde{\tau}-1)(1-\delta)D(s,a) + 1 + h)^2 D(s,a)}$$

$$+ 2\sum_{s,a} \frac{3(\tilde{\tau}^2 + (h+1)^2)}{(1+h)^2 D(s,a)}\mathbb{P}(E_\delta^c(s,a))$$

$$= 2\sum_{s,a} \frac{3(k-\tilde{\tau}-1)^2}{((k-\tilde{\tau}-1)(1-\delta)D(s,a) + 1 + h)^2 D(s,a)}\mathbb{E}\left[(\bar{D}_{k-\tilde{\tau}-1}(s,a) - D(s,a))^2\right]$$

$$+ 2\sum_{s,a} \frac{3(\tilde{\tau}^2 + (h+1)^2)}{((k-\tilde{\tau}-1)(1-\delta)D(s,a) + 1 + h)^2 D(s,a)}$$

$$+ 2\sum_{s,a} \frac{3((k-\tilde{\tau}-1)^2 + \tilde{\tau}^2 + (1+h)^2)}{(1+h)^2 D(s,a)}\mathbb{P}(E_\delta^c(s,a)).$$

It remains to bound the terms $\mathbb{E}[(\bar{D}_{k-\tilde{\tau}-1}(s,a) - D(s,a))^2]$ and $\mathbb{P}(E_\delta^c(s,a))$. To bound $\mathbb{E}[(\bar{D}_{k-\tilde{\tau}-1}(s,a) - D(s,a))^2]$, we use the following lemma, which is a mean-square bound for functions of finite, irreducible, and aperiodic Markov chains. The proof of Lemma B.3 is provided in Appendix B.10.3.

**Lemma B.3.** *Under Assumption 3.1, the following inequality holds for all $k \ge 14C/[(1-\rho)D_{\min}]$:*

$$\mathbb{E}\left[(\bar{D}_k(s,a) - D(s,a))^2\right] \le \frac{10CD(s,a)}{(1-\rho)k}.$$

For simplicity of notation, denote $C_{mc} = C/(1-\rho)$. Apply Lemma B.3, we have when $k \ge 14C_{mc}/D_{\min}$ that

$$\mathbb{E}\left[\left(\frac{1}{D_k(S_k,A_k)} - \frac{1}{D(S_k,A_k)}\right)^2\right] \le \sum_{s,a} \frac{60C_{mc}(k-\tilde{\tau}-1)}{((k-\tilde{\tau}-1)(1-\delta)D(s,a) + 1 + h)^2}$$

$$+ \sum_{s,a} \frac{6(\tilde{\tau}^2 + (h+1)^2)}{((k-\tilde{\tau}-1)(1-\delta)D(s,a) + 1 + h)^2 D(s,a)}$$

$$+\sum_{s,a}\frac{6((k-\tilde{\tau}-1)^2+\tilde{\tau}^2+(1+h)^2)}{(1+h)^2D(s,a)}\mathbb{P}(E_\delta^c(s,a)).$$

To proceed, observe that

$$\frac{60C_{mc}(k-\tilde{\tau}-1)}{((k-\tilde{\tau}-1)(1-\delta)D(s,a)+1+h)^2}\leq\frac{60C_{mc}(k-\tilde{\tau}-1)}{((k-\tilde{\tau}-1)(1-\delta)D(s,a)+(1+h)(1-\delta)D(s,a))^2}$$

$$\leq\frac{60C_{mc}(k-\tilde{\tau}-1)}{(k+h-\tilde{\tau})^2(1-\delta)^2D(s,a)^2}$$

and

$$\frac{6(\tilde{\tau}^2+(h+1)^2)}{((k-\tilde{\tau}-1)(1-\delta)D(s,a)+1+h)^2D(s,a)}$$

$$\leq\frac{6(\tilde{\tau}^2+(h+1)^2)}{((k-\tilde{\tau}-1)(1-\delta)D(s,a)+(1+h)(1-\delta)D(s,a))^2D(s,a)}$$

$$=\frac{6(\tilde{\tau}^2+(h+1)^2)}{(k+h-\tilde{\tau})^2(1-\delta)^2D(s,a)^3}.$$

Therefore, we have

$$\mathbb{E}\left[\left(\frac{1}{D_k(S_k,A_k)}-\frac{1}{D(S_k,A_k)}\right)^2\right]\leq\frac{60C_{mc}|\mathcal{S}||\mathcal{A}|(k-\tilde{\tau}-1)}{(k+h-\tilde{\tau})^2(1-\delta)^2D_{\min}^2}+\frac{6|\mathcal{S}||\mathcal{A}|(\tilde{\tau}^2+(h+1)^2)}{(k+h-\tilde{\tau})^2(1-\delta)^2D_{\min}^3}$$

$$+\frac{6|\mathcal{S}||\mathcal{A}|((k-\tilde{\tau}-1)^2+\tilde{\tau}^2+(1+h)^2)}{(1+h)^2D_{\min}}\mathbb{P}(E_\delta^c(s,a)).$$

To bound $\mathbb{P}(E_\delta^c(s,a))$, we use the Markov chain concentration inequality stated in Lemma A.5, which implies

$$\mathbb{P}(E_\delta^c(s,a))=\mathbb{P}\left(|\bar{D}_{k-\tilde{\tau}-1}(s,a)-D(s,a)|>\delta D(s,a)\right)$$

$$\leq 2\exp(-c_{mc}(k-\tilde{\tau}-1)\delta^2D(s,a)^2)$$

for all $\delta\geq 4C/[D_{\min}(1-\rho)(k-\tilde{\tau}-1)]$. It follows that

$$\mathbb{E}\left[\left(\frac{1}{D_k(S_k,A_k)}-\frac{1}{D(S_k,A_k)}\right)^2\right]$$

$$\leq\frac{60C_{mc}|\mathcal{S}||\mathcal{A}|(k-\tilde{\tau}-1)}{(k+h-\tilde{\tau})^2(1-\delta)^2D_{\min}^2}+\frac{6|\mathcal{S}||\mathcal{A}|(\tilde{\tau}^2+(h+1)^2)}{(k+h-\tilde{\tau})^2(1-\delta)^2D_{\min}^3}$$

$$+\frac{12|\mathcal{S}||\mathcal{A}|((k-\tilde{\tau}-1)^2+\tilde{\tau}^2+(1+h)^2)}{(1+h)^2D_{\min}}\exp(-c_{mc}(k-\tilde{\tau}-1)\delta^2D_{\min}^2).$$

Let $\tilde{K}'$ be such that the following inequality holds for all $k\geq\tilde{K}'$:

$$\frac{4C}{D_{\min}(1-\rho)(k-\tilde{\tau}-1)}\leq\sqrt{\frac{\log\left(\frac{12(k+h)((k-\tilde{\tau}-1)^2+\tilde{\tau}^2+(1+h)^2)}{(1+h)^2}\right)}{c_{mc}(k-\tilde{\tau}-1)D_{\min}^2}}.$$

Then, for any $k\geq\tilde{K}'$, choose

$$\delta=\sqrt{\frac{1}{c_{mc}(k-\tilde{\tau}-1)D_{\min}^2}\log\left(\frac{12(k+h)((k-\tilde{\tau}-1)^2+\tilde{\tau}^2+(1+h)^2)}{(1+h)^2}\right)},$$

we have

$$\frac{12|\mathcal{S}||\mathcal{A}|((k-\tilde{\tau}-1)^2+\tilde{\tau}^2+(1+h)^2)}{(1+h)^2D_{\min}}\exp(-c_{mc}(k-\tilde{\tau}-1)\delta^2D_{\min}^2)=\frac{|\mathcal{S}||\mathcal{A}|}{(k+h)D_{\min}}.$$

It follows that

$$\mathbb{E}\left[\left(\frac{1}{D_k(S_k,A_k)}-\frac{1}{D(S_k,A_k)}\right)^2\right]\leq\frac{60C_{mc}|\mathcal{S}||\mathcal{A}|(k-\tilde{\tau}-1)}{(k+h-\tilde{\tau})^2(1-\delta)^2D_{\min}^2}+\frac{6|\mathcal{S}||\mathcal{A}|(\tilde{\tau}^2+(h+1)^2)}{(k+h-\tilde{\tau})^2(1-\delta)^2D_{\min}^3}$$

$$+ \frac{|\mathcal{S}||\mathcal{A}|}{(k+h)D_{\min}}.$$

Let $\tilde{K}''$ be such that the following inequalities hold for all $k \geq \tilde{K}''$:

$$\frac{(k-\tilde{\tau}-1)}{(k+h-\tilde{\tau})^2(1-\delta)^2} \leq \frac{61}{60(k+h)}, \qquad \frac{(\tilde{\tau}^2+(h+1)^2)}{(k+h-\tilde{\tau})^2(1-\delta)^2 D_{\min}} \leq \frac{1}{6(k+h)}.$$

Then, we have for all $k \geq \tilde{K}''$ that

$$\mathbb{E}\left[\left(\frac{1}{D_k(S_k,A_k)} - \frac{1}{D(S_k,A_k)}\right)^2\right] \leq \frac{61 C_{mc}|\mathcal{S}||\mathcal{A}|}{(k+h)D_{\min}^2} + \frac{|\mathcal{S}||\mathcal{A}|}{(k+h)D_{\min}^2} + \frac{|\mathcal{S}||\mathcal{A}|}{(k+h)D_{\min}}$$
$$\leq \frac{63 C_{mc}|\mathcal{S}||\mathcal{A}|}{(k+h)D_{\min}^2},$$

where the last line follows from $C_{mc} = C/(1-\rho) \geq 1$. Finally, combining the previous inequality with Eq. (38), we have for all $k \geq K_3 := \max(14 C_{mc}/D_{\min}, \tilde{K}', \tilde{K}'')$ that

$$
\begin{aligned}
T_3 &\leq \left(1 - \frac{\beta u_m}{\ell_m}\right)\mathbb{E}\left[M(Q_k - Q^*)\right] + \frac{(1+b_k)^2 \mathbb{E}\left[\left(\frac{1}{D_k(S_k,A_k)} - \frac{1}{D(S_k,A_k)}\right)^2\right]}{2\ell_m^2(1-\beta u_m/\ell_m)} \\
&\leq \left(1 - \frac{\beta u_m}{\ell_m}\right)\mathbb{E}\left[M(Q_k - Q^*)\right] + \frac{32 C_{mc}|\mathcal{S}||\mathcal{A}|(1+b_k)^2}{\ell_m^2(1-\beta u_m/\ell_m)\alpha D_{\min}^2}\alpha_k \\
&\leq \left(1 - \frac{\beta u_m}{\ell_m}\right)\mathbb{E}\left[M(Q_k - Q^*)\right] + \frac{32 C_{mc}|\mathcal{S}||\mathcal{A}|(b_k + p_{\mathrm{span}}(Q^*) + 1)^2}{\ell_m^2(1-\beta u_m/\ell_m)\alpha D_{\min}^2}\alpha_k \\
&= \phi_1 \mathbb{E}\left[M(Q_k - Q^*)\right] + \frac{32 C|\mathcal{S}||\mathcal{A}|(b_k + p_{\mathrm{span}}(Q^*) + 1)^2}{\ell_m^2\phi_1(1-\rho)\alpha D_{\min}^2}\alpha_k,
\end{aligned}
\tag{40}
$$

where the last line follows from our definition $\phi_1 = 1 - \beta u_m/\ell_m$.

## B.8 Proof of Lemma A.6

Since the vector $F(Q_k, D_k, Y_k) - Q_k$ has only one non-zero entry, i.e., the $(S_k, A_k)$-th one, we have by the definition of $p_{\mathrm{span}}(\cdot)$ that

$$
\begin{aligned}
&p_{\mathrm{span}}(F(Q_k, D_k, Y_k) - Q_k)^2 \\
&= \frac{1}{4}\left(\frac{1}{D_k(S_k,A_k)}\right)^2 \left(\mathcal{R}(S_k,A_k) + \max_{a' \in \mathcal{A}} Q_k(S_{k+1},a') - Q_k(S_k,A_k)\right)^2 \\
&\leq \frac{1}{4}\left(\frac{1}{D_k(S_k,A_k)}\right)^2 (1 + 2p_{\mathrm{span}}(Q_k))^2 \\
&\leq \frac{1}{4}\left(\frac{1}{D_k(S_k,A_k)}\right)^2 (1 + 2b_k)^2 \qquad\qquad \text{(Proposition A.1)} \\
&\leq (1+b_k)^2 \left(\frac{1}{D_k(S_k,A_k)}\right)^2.
\end{aligned}
$$

Taking expectations on both sides, we obtain

$$T_4 \leq (1+b_k)^2 \mathbb{E}\left[\frac{1}{D_k(S_k,A_k)^2}\right].
\tag{41}$$

It remains to bound $\mathbb{E}[1/D_k(S_k,A_k)^2]$. Observe that for any $\tilde{\tau} \leq k - 1$, we have

$$
\begin{aligned}
\mathbb{E}\left[\frac{1}{D_k(S_k,A_k)^2}\right] &= \mathbb{E}\left[\frac{(k+h)^2}{(N_k(S_k,A_k)+h)^2}\right] \qquad\qquad (D_k(s,a) = (N_k(s,a)+h)/(k+h)) \\
&= \mathbb{E}\left[\frac{(k+h)^2}{(N_{k-1}(S_k,A_k)+1+h)^2}\right] \\
&\qquad\qquad\qquad \text{(The pair } (S_k, A_k) \text{ is visited at time step } k.)
\end{aligned}
$$

$$= \sum_{s,a} \mathbb{E}\left[\frac{\mathbb{1}_{\{(s,a)=(S_k,A_k)\}}(k+h)^2}{(N_{k-1}(s,a)+1+h)^2}\right]$$

$$\leq \sum_{s,a} \mathbb{E}\left[\frac{\mathbb{1}_{\{(s,a)=(S_k,A_k)\}}(k+h)^2}{(N_{k-\tilde{\tau}-1}(s,a)+1+h)^2}\right]$$

$$(N_{k_1}(s,a) \leq N_{k_2}(s,a) \text{ for any } k_1 \leq k_2)$$

$$= \sum_{s,a} \mathbb{E}\left[\mathbb{P}(S_k=s, A_k=a|S_{k-\tilde{\tau}-1}, A_{k-\tilde{\tau}-1})\frac{(k+h)^2}{(N_{k-\tilde{\tau}-1}(s,a)+1+h)^2}\right],$$

where the last line follows from the tower property of conditional expectations and the Markov property.

Recall that we have shown in Eq. (39) that, when choosing $\tilde{\tau} = \min\{t : C\rho^t \leq D_{\min}\}$, we have

$$\mathbb{P}(S_k=s, A_k=a|S_{k-\tilde{\tau}-1}, A_{k-\tilde{\tau}-1}) \leq 2D(s,a).$$

It follows that

$$\mathbb{E}\left[\frac{1}{D_k(S_k, A_k)^2}\right] \leq \sum_{s,a} 2D(s,a)\mathbb{E}\left[\frac{(k+h)^2}{(N_{k-\tilde{\tau}-1}(s,a)+1+h)^2}\right]. \tag{42}$$

To proceed and bound $\mathbb{E}\left[(k+h)^2/(N_{k-\tilde{\tau}-1}(s,a)+1+h)^2\right]$, given $\delta \in (0,1)$, for any $(s,a)$, let

$$E_\delta(s,a) = \{|\bar{D}_{k-\tilde{\tau}-1}(s,a) - D(s,a)| \leq \delta D(s,a)\},$$

and let $E_\delta^c(s,a)$ be the complement of event $E_\delta(s,a)$. Note that on the event $E_\delta(s,a)$, we have $|\bar{D}_{k-\tilde{\tau}-1}(s,a) - D(s,a)| \leq \delta D(s,a)$, which implies

$$\bar{D}_{k-\tilde{\tau}-1}(s,a) \geq (1-\delta)D(s,a),$$

while on the event $E_\delta^c(s,a)$, we have the trivial bound $\bar{D}_{k-\tilde{\tau}-1}(s,a) \geq 0$. Therefore, we obtain

$$\mathbb{E}\left[\frac{(k+h)^2}{(N_{k-\tilde{\tau}-1}(s,a)+1+h)^2}\right] = \frac{(k+h)^2}{(k-\tilde{\tau}-1)^2}\mathbb{E}\left[\frac{(k-\tilde{\tau}-1)^2}{(N_{k-\tilde{\tau}-1}(s,a)+1+h)^2}\right]$$

$$= \frac{(k+h)^2}{(k-\tilde{\tau}-1)^2}\mathbb{E}\left[\frac{1}{(\bar{D}_{k-\tilde{\tau}-1}(s,a)+(1+h)/(k-\tilde{\tau}-1))^2}\right]$$

$$= \frac{(k+h)^2}{(k-\tilde{\tau}-1)^2}\mathbb{E}\left[\frac{\mathbb{1}_{\{E_\delta(s,a)\}} + \mathbb{1}_{\{E_\delta^c(s,a)\}}}{(\bar{D}_{k-\tilde{\tau}-1}(s,a)+(1+h)/(k-\tilde{\tau}-1))^2}\right]$$

$$\leq \frac{(k+h)^2}{(k-\tilde{\tau}-1)^2}\frac{1}{((1-\delta)D(s,a)+(1+h)/(k-\tilde{\tau}-1))^2}$$

$$+ \frac{(k+h)^2}{(h+1)^2}\mathbb{P}(E_\delta^c(s,a))$$

$$\leq \frac{(k+h)^2}{(k-\tilde{\tau}-1)^2}\frac{1}{(1-\delta)^2 D(s,a)^2} + \frac{(k+h)^2}{(h+1)^2}\mathbb{P}(E_\delta^c(s,a)).$$

To bound $\mathbb{P}(E_\delta^c(s,a))$, we use the Markov chain concentration inequality stated in Lemma A.5, which implies

$$\mathbb{P}(E_\delta^c(s,a)) = \mathbb{P}\left(|\bar{D}_{k-\tilde{\tau}-1}(s,a) - D(s,a)| > \delta D(s,a)\right)$$

$$\leq 2\exp(-c_{mc}(k-\tilde{\tau}-1)\delta^2 D(s,a)^2)$$

for any $\delta \geq 4C/[D_{\min}(1-\rho)(k-\tilde{\tau}-1)]$. Combining the previous two inequalities, we have

$$\mathbb{E}\left[\frac{(k+h)^2}{(N_{k-\tilde{\tau}-1}(s,a)+1+h)^2}\right] \leq \frac{(k+h)^2}{(k-\tilde{\tau}-1)^2}\frac{1}{(1-\delta)^2 D(s,a)^2}$$

$$+ \frac{2(k+h)^2}{(h+1)^2}\exp(-c_{mc}(k-\tilde{\tau}-1)\delta^2 D(s,a)^2).$$

Let $\bar{K}'$ be such that the following inequality holds for all $k \geq \bar{K}'$:

$$\frac{4C}{D_{\min}(1-\rho)(k-\tilde{\tau}-1)} \leq \max_{s,a} \sqrt{\frac{\log[2(k+h)^2 D(s,a)^2/(h+1)^2]}{c_{mc}(k-\tilde{\tau}-1)D(s,a)^2}}.$$

Then, for any $k \geq \bar{K}'$, choosing

$$\delta = \max_{s,a} \sqrt{\frac{\log[2(k+h)^2 D(s,a)^2/(h+1)^2]}{c_{mc}(k-\tilde{\tau}-1)D(s,a)^2}},$$

we have

$$\frac{2(k+h)^2}{(h+1)^2} \exp(-c_{mc}(k-\tilde{\tau}-1)\delta^2 D(s,a)^2) \leq \frac{1}{D(s,a)^2}.$$

It follows that

$$\begin{aligned}
\mathbb{E}\left[\frac{(k+h)^2}{(N_{k-\tilde{\tau}-1}(s,a)+1+h)^2}\right] &\leq \frac{(k+h)^2}{(k-\tilde{\tau}-1)^2} \frac{1}{(1-\delta)^2 D(s,a)^2} \\
&\quad + \frac{2(k+h)^2}{(h+1)^2} \exp(-c_{mc}(k-\tilde{\tau}-1)\delta^2 D(s,a)^2) \\
&\leq \frac{(k+h)^2}{(k-\tilde{\tau}-1)^2} \frac{1}{(1-\delta)^2 D(s,a)^2} + \frac{1}{D(s,a)^2}.
\end{aligned}$$

Let $\bar{K}''$ be such that the following inequality holds for all $k \geq \bar{K}''$:

$$\frac{(k+h)^2}{(k-\tilde{\tau}-1)^2} \frac{1}{(1-\delta)^2} \leq 2.$$

Then, for any $k \geq \bar{K}''$, we have

$$\mathbb{E}\left[\frac{(k+h)^2}{(N_{k-\tilde{\tau}-1}(s,a)+1+h)^2}\right] \leq \frac{2}{D(s,a)^2} + \frac{1}{D(s,a)^2} \leq \frac{3}{D(s,a)^2}.$$

Using the previous inequality in Eq. (42), we have

$$\mathbb{E}\left[\frac{1}{D_k(S_k,A_k)^2}\right] \leq \sum_{s,a} \frac{6}{D(s,a)^2} \leq \frac{6|\mathcal{S}||\mathcal{A}|}{D_{\min}}.$$

Finally, using the previous inequality in Eq. (41), we obtain for any $k \geq K_4 := \max(\bar{K}', \bar{K}'')$ that

$$T_4 \leq \frac{6|\mathcal{S}||\mathcal{A}|(1+b_k)^2}{D_{\min}} \leq \frac{6|\mathcal{S}||\mathcal{A}|(b_k + p_{\mathrm{span}}(Q^*)+1)^2}{D_{\min}}.$$

## B.9  Proof of Proposition A.4

Using the upper bounds we obtained for the terms $T_1$ (cf. Lemma A.4), $T_2$ (cf. Proposition A.2), $T_3$ (cf. Proposition A.3), and $T_4$ (cf. Lemma A.6), in Eq. (16), we obtain for any $k \geq K = \max(K_2, K_3, K_4)$ that

$$\begin{aligned}
\mathbb{E}[M(Q_{k+1}-Q^*)] &\leq \mathbb{E}[M(Q_k-Q^*)] + \alpha_k(T_1+T_2+T_3) + \frac{L\alpha_k^2}{2}T_3 \\
&\leq \left(1 - \left(1 - \frac{\beta u_m}{\ell_m}\right)\alpha_k\right) \mathbb{E}\left[M(Q_k-Q^*)\right] \\
&\quad + \frac{35|\mathcal{S}||\mathcal{A}|}{D_{\min}^2}\left(\frac{C}{\ell_m^2(1-\beta u_m/\ell_m)(1-\rho)\alpha} + L\right)\tau_k(b_k + p_{\mathrm{span}}(Q^*)+1)^2\alpha_k \\
&= (1 - \phi_1\alpha_k)\mathbb{E}\left[M(Q_k-Q^*)\right] + 35\phi_2\tau_k(b_k + p_{\mathrm{span}}(Q^*)+1)^2\alpha_k^2,
\end{aligned}$$

where

$$\phi_2 = \frac{|\mathcal{S}||\mathcal{A}|}{D_{\min}^2}\left(\frac{C}{\ell_m^2(1-\rho)\phi_1\alpha} + L\right).$$

### B.10  Proofs of Auxiliary Lemmas

#### B.10.1  Proof of Lemma B.1

To show the desired inequality, it suffices to prove that

$$\max_{s,a} Q_{k+1}(s,a) \leq \max_{s,a} Q_k(s,a) + \alpha_k(S_k, A_k), \tag{43}$$

$$\min_{s,a} Q_{k+1}(s,a) \geq \min_{s,a} Q_k(s,a) - \alpha_k(S_k, A_k). \tag{44}$$

The result then follows from the definition of the span seminorm:

$$p_{\mathrm{span}}(Q_{k+1}) = \frac{1}{2}\left(\max_{s,a} Q_{k+1}(s,a) - \min_{s,a} Q_{k+1}(s,a)\right)$$

$$\leq \frac{1}{2}\left(\max_{s,a} Q_k(s,a) - \min_{s,a} Q_k(s,a)\right) + \alpha_k(S_k, A_k)$$

$$= p_{\mathrm{span}}(Q_k) + \alpha_k(S_k, A_k).$$

In the following, we will prove Eq. (43); the proof of Eq. (44) follows from an identical argument. In view of Algorithm 1, when $(s,a) \neq (S_k, A_k)$, we have

$$Q_{k+1}(s,a) = Q_k(s,a) + c_k \leq \max_{s,a} Q_k(s,a) + \alpha_k(S_k, A_k).$$

When $(s,a) = (S_k, A_k)$, since $\max_{s,a}|\mathcal{R}(s,a)| \leq 1$, we have

$$Q_{k+1}(s,a) = Q_k(s,a) + \alpha_k(S_k, A_k)\left(\mathcal{R}(S_k, A_k) + \max_{a'\in\mathcal{A}} Q_k(S_{k+1}, a') - Q_k(S_k, A_k)\right)$$

$$= (1 - \alpha_k(S_k, A_k))\, Q_k(S_k, A_k) + \alpha_k(S_k, A_k)\left(\mathcal{R}(s,a) + \max_{a'\in\mathcal{A}} Q_{k+1}(S_{k+1}, a')\right)$$

$$\leq (1 - \alpha_k(S_k, A_k))\max_{s',a'} Q_k(s',a') + \alpha_k(S_k, A_k)\left(1 + \max_{s',a'} Q_k(s',a')\right) + c_k$$

$$= \max_{s',a'} Q_k(s',a') + \alpha_k(S_k, A_k).$$

Combining both cases, we obtain

$$\max_{s,a} Q_{k+1}(s,a) \leq \max_{s,a} Q_k(s,a) + \alpha_k(S_k, A_k),$$

which establishes Eq. (43).

#### B.10.2  Proof of Lemma B.2

Using the definition of $p_{\mathrm{span}}(\cdot)$, we have by Algorithm 1 that

$$p_{\mathrm{span}}(Q_{k+1}) - p_{\mathrm{span}}(Q_k) \leq p_{\mathrm{span}}(Q_{k+1} - Q_k)$$

$$= \frac{\alpha_k(S_k, A_k)}{2}\left|\mathcal{R}(S_k, A_k) + \max_{a'\in\mathcal{A}} Q_k(S_{k+1}, a') - Q_k(S_k, A_k)\right|$$

$$\leq \frac{\alpha_k(S_k, A_k)}{2}\left(|\mathcal{R}(S_k, A_k)| + \left|\max_{a'\in\mathcal{A}} Q_k(S_{k+1}, a') - Q_k(S_k, A_k)\right|\right)$$

$$\leq \frac{\alpha_k(S_k, A_k)}{2}\left(|\mathcal{R}(S_k, A_k)| + \max_{s',a'} Q_k(s',a') - \min_{s',a'} Q_k(s',a')\right)$$

$$\leq \frac{\alpha_k(S_k, A_k)}{2}(1 + 2p_{\mathrm{span}}(Q_k))$$

$$\leq \alpha_k(S_k, A_k)\,(p_{\mathrm{span}}(Q_k) + 1)\,. \tag{45}$$

Rearranging terms, we obtain

$$p_{\mathrm{span}}(Q_{k+1}) + 1 \leq (1 + \alpha_k(S_k, A_k))\,(p_{\mathrm{span}}(Q_k) + 1)\,.$$

Therefore, we have for all $k \geq k_1$ that

$$p_{\text{span}}(Q_k) \leq \prod_{j=k_1}^{k-1} (1 + \alpha_j(S_j, A_j)) \, (p_{\text{span}}(Q_{k_1}) + 1) - 1$$

$$= \prod_{j=k_1}^{k-1} \left(1 + \frac{\alpha}{N_j(S_j, A_j) + h}\right) (p_{\text{span}}(Q_{k_1}) + 1) - 1$$

$$= \prod_{j=k_1}^{k-1} \left(1 + \frac{\alpha}{N_{j-1}(S_j, A_j) + 1 + h}\right) (p_{\text{span}}(Q_{k_1}) + 1) - 1$$

$$((S_j, A_j) \text{ is visited in the } j\text{-th iteration})$$

$$\leq \prod_{j=k_1}^{k-1} \left(1 + \frac{\alpha}{N_{k_1-1,\min} + 1 + h}\right) (p_{\text{span}}(Q_{k_1}) + 1) - 1$$

$$\leq \exp\left(\frac{\alpha(k - k_1)}{N_{k_1-1,\min} + 1 + h}\right) (p_{\text{span}}(Q_{k_1}) + 1) - 1,$$

where the last line follows from $(1 + x) \leq e^x$ for all $x \in \mathbb{R}$. Rearranging terms, we have

$$p_{\text{span}}(Q_k) + 1 \leq \exp\left(\frac{\alpha(k - k_1)}{N_{k_1-1,\min} + 1 + h}\right) (p_{\text{span}}(Q_{k_1}) + 1), \quad \forall k \geq k_1.$$

Using the previous inequality in Eq. (45), we have

$$p_{\text{span}}(Q_{k+1} - Q_k) \leq \alpha_k(S_k, A_k)(p_{\text{span}}(Q_k) + 1)$$

$$\leq \frac{\alpha}{(N_{k_1-1,\min} + 1 + h)} \exp\left(\frac{\alpha(k - k_1)}{N_{k_1-1,\min} + 1 + h}\right) (p_{\text{span}}(Q_{k_1}) + 1).$$

By telescoping, we have for any $k \geq k_1$ that

$$p_{\text{span}}(Q_k - Q_{k_1}) \leq \sum_{i=k_1}^{k-1} p_{\text{span}}(Q_{i+1} - Q_i)$$

$$\leq \sum_{i=k_1}^{k-1} \frac{\alpha}{N_{k_1-1,\min} + 1 + h} \exp\left(\frac{\alpha(i - k_1)}{N_{k_1-1,\min} + 1 + h}\right) (p_{\text{span}}(Q_{k_1}) + 1)$$

$$\leq \frac{\alpha(k - k_1)}{N_{k_1-1,\min} + 1 + h} \exp\left(\frac{\alpha(k - k_1)}{N_{k_1-1,\min} + 1 + h}\right) (p_{\text{span}}(Q_{k_1}) + 1)$$

$$= f\left(\frac{\alpha(k - k_1)}{N_{k_1-1,\min} + 1 + h}\right) (p_{\text{span}}(Q_{k_1}) + 1),$$

where the last line follows from the definition of $f(\cdot)$.

### B.10.3   Proof of Lemma B.3

For simplicity of notation, let $X_k(s, a) = \mathbb{1}_{\{(S_k, A_k) = (s,a)\}}$ for all $k \geq 1$. By definition of $\bar{D}_k(s, a)$, we have

$$\mathbb{E}\left[(\bar{D}_k(s, a) - D(s, a))^2\right] = \mathbb{E}\left[(\bar{D}_k(s, a) - \mathbb{E}[\bar{D}_k(s, a)] + \mathbb{E}[\bar{D}_k(s, a)] - D(s, a))^2\right]$$

$$= \underbrace{\mathbb{E}\left[(\bar{D}_k(s, a) - \mathbb{E}[\bar{D}_k(s, a)])^2\right]}_{\text{Variance}} + \underbrace{(\mathbb{E}[\bar{D}_k(s, a)] - D(s, a))^2}_{\text{bias}^2}. \quad (46)$$

We first bound the bias term. Observe that

$$\text{bias}^2 = \left(\frac{\sum_{i=1}^{k}(\mathbb{E}[X_i(s, a)] - D(s, a))}{k}\right)^2 \leq \frac{1}{k^2}\left(\sum_{i=1}^{k} |\mathbb{E}[X_i(s, a)] - D(s, a)|\right)^2.$$

Moreover, under Assumption 3.1, we have

$$
\begin{aligned}
|\mathbb{E}[X_i(s,a)] - D(s,a)| &= |\mathbb{P}(S_i = s, A_i = a|S_1) - \mu(s)\pi(a|s)| \\
&= |p_\pi(S_i = s|S_1) - \mu(s)|\pi(a|s) \\
&\leq \max_{s'} \sum_s |p_\pi(S_i = s|S_1 = s') - \mu(s)| \\
&\leq 2C\rho^{i-1}.
\end{aligned}
\tag{47}
$$

It follows that

$$
\text{bias}^2 \leq \frac{1}{k^2}\left(\sum_{i=1}^{k} 2C\rho^{i-1}\right)^2 \leq \frac{4C^2}{k^2(1-\rho)^2}.
\tag{48}
$$

Next, we turn to the variance term on the right-hand side of Eq (46). Observe that

$$
\begin{aligned}
\text{Variance} &= \frac{1}{k^2}\text{Var}\left(\sum_{i=1}^{k} X_i(s,a)\right) \\
&= \frac{1}{k^2}\sum_{i=1}^{k}\text{Var}(X_i(s,a)) + \frac{2}{k^2}\sum_{1\leq i<j\leq k}\text{Cov}(X_i(s,a), X_j(s,a))
\end{aligned}
\tag{49}
$$

On the one hand, we have

$$
\begin{aligned}
\text{Var}(X_i(s,a)) &= \mathbb{E}[X_i(s,a)^2] - (\mathbb{E}[X_i(s,a)])^2 \\
&= \mathbb{E}[X_i(s,a)] - (\mathbb{E}[X_i(s,a)])^2 \\
&\leq \mathbb{E}[X_i(s,a)] \\
&\leq |\mathbb{E}[X_i(s,a)] - D(s,a)| + D(s,a) \\
&\leq 2C\rho^{i-1} + D(s,a),
\end{aligned}
\tag{50}
$$

where the last line follows from Eq. (47). On the other hand, we have

$$
\begin{aligned}
\text{Cov}(X_i(s,a), X_j(s,a)) &= \mathbb{E}[X_i(s,a)X_j(s,a)] - \mathbb{E}[X_i(s,a)]\mathbb{E}[X_j(s,a)] \\
&= \mathbb{P}(X_i(s,a)=1, X_j(s,a)=1) - \mathbb{P}(X_i(s,a)=1)\mathbb{P}(X_j(s,a)=1) \\
&= \mathbb{P}(X_i(s,a)=1)(\mathbb{P}(X_j(s,a)=1|X_i(s,a)=1) - \mathbb{P}(X_j(s,a)=1)) \\
&\leq \mathbb{P}(X_i(s,a)=1)|\mathbb{P}(X_j(s,a)=1|X_i(s,a)=1) - \mathbb{P}(X_j(s,a)=1)|.
\end{aligned}
$$

Using the same analysis as in Eq. (47), we obtain

$$
\mathbb{P}(X_i(s,a)=1) \leq |\mathbb{P}(X_i(s,a)=1) - D(s,a)| + D(s,a) \leq 2C\rho^{i-1} + D(s,a)
$$

and

$$
\begin{aligned}
&|\mathbb{P}(X_j(s,a)=1|X_i(s,a)=1) - \mathbb{P}(X_j(s,a)=1)| \\
&\leq |\mathbb{P}(X_j(s,a)=1|X_i(s,a)=1) - D(s,a)| + |D(s,a) - \mathbb{P}(X_j(s,a)=1)| \\
&\leq 2C\rho^{j-i-1} + 2C\rho^{j-1} \\
&\leq 4C\rho^{j-i-1}.
\end{aligned}
$$

Combining the previous three inequalities together, we have

$$
\begin{aligned}
\text{Cov}(X_i(s,a), X_j(s,a)) &\leq (C\rho^{i-1} + D(s,a))4C\rho^{j-i-1} \\
&= 4C^2\rho^{j-2} + D(s,a)4C\rho^{j-i-1}.
\end{aligned}
$$

Using the previous inequality and Eq. (50) together in Eq. (49), we have

$$
\begin{aligned}
\text{Variance} &= \frac{1}{k^2}\sum_{i=1}^{k}\text{Var}(X_i(s,a)) + \frac{2}{k^2}\sum_{1\leq i<j\leq k}\text{Cov}(X_i(s,a), X_j(s,a)) \\
&\leq \frac{1}{k^2}\sum_{i=1}^{k}2C\rho^{i-1} + \frac{D(s,a)}{k} + \frac{2}{k^2}\sum_{i=1}^{k}\sum_{j=i+1}^{k}(4C^2\rho^{j-2} + D(s,a)4C\rho^{j-i-1})
\end{aligned}
$$

$$\leq \frac{2C}{k^2(1-\rho)} + \frac{D(s,a)}{k} + \frac{2}{k^2}\frac{4C^2}{(1-\rho)^2} + \frac{8C}{k(1-\rho)}D(s,a)$$

$$\leq \frac{10C^2}{k^2(1-\rho)^2} + \frac{9CD(s,a)}{k(1-\rho)},$$

where the last line follows from $C \geq 1$ and $\rho \in (0,1)$.

Finally, using the previous inequality and Eq. (48) together in Eq. (46), we obtain

$$\mathbb{E}\left[(\bar{D}_k(s,a) - D(s,a))^2\right] = \text{variance} + \text{bias}^2$$

$$\leq \frac{14C^2}{k^2(1-\rho)^2} + \frac{9CD(s,a)}{k(1-\rho)}$$

$$\leq \frac{10CD(s,a)}{(1-\rho)k},$$

where the last inequality follows from $k \geq 14C/[(1-\rho)D_{\min}]$.

## C  Proofs of All Technical Results in Sections 2 and 4

### C.1  Proof of Lemma 2.1

We first verify that when $c = (\max_i x_i + \min_j x_j)/2$, we have $\|x - ce\|_\infty = (\max_i x_i - \min_j x_j)/2$. On the one hand, for any $i \in \{1, 2, \cdots, d\}$, we have

$$|x_i - c| \leq \max\left(\left|\max_i x_i - c\right|, \left|\min_j x_j - c\right|\right) = \frac{\max_i x_i - \min_j x_j}{2},$$

which implies

$$\|x - ce\|_\infty \leq \frac{\max_i x_i - \min_j x_j}{2}.$$

On the other hand, letting $i_1 = \arg\max_{i \in \{1,2,\cdots,d\}} x_i$, we have

$$\|x - ce\|_\infty \geq |x_{i_1} - c| = \frac{\max_i x_i - \min_j x_j}{2}.$$

Together, they imply

$$\|x - ce\|_\infty = \frac{\max_i x_i - \min_j x_j}{2} = p_{\text{span}}(x).$$

To complete the proof, it remains to show that for any $c' \neq c$, we have $\|x - c'e\|_\infty > \|x - ce\|_\infty$. Assume without loss of generality that $c' < c = (\max_i x_i + \min_j x_j)/2$. Then, we have

$$\|x - c'e\|_\infty \geq |x_{i_1} - c'e|$$

$$= \max_i x_i - c'$$

$$> \max_i x_i - \frac{\max_i x_i + \min_j x_j}{2},$$

$$= \frac{\max_i x_i - \min_j x_j}{2}.$$

### C.2  Proof of Lemma 2.2

For simplicity of notation, denote

$$\mathcal{Q}_1 = \{Q \mid \mathcal{H}(Q) - Q = r^*e\}, \quad \text{and} \quad \mathcal{Q}_2 = \{Q \mid p_{\text{span}}(\mathcal{H}(Q) - Q) = 0\}.$$

We will show that $\mathcal{Q} \subseteq \mathcal{Q}_1 \subseteq \mathcal{Q}_2 \subseteq \mathcal{Q}$, which would imply $\mathcal{Q} = \mathcal{Q}_1 = \mathcal{Q}_2$.

For any $Q \in \mathcal{Q}$, there exists $c \in \mathbb{R}$ such that $Q = Q^* + ce$. Since

$$\mathcal{H}(Q) - Q = \mathcal{H}(Q^* + ce) - Q^* - ce = \mathcal{H}(Q^*) - Q^* = r^*e,$$

we have $Q \in \mathcal{Q}_1$, implying $\mathcal{Q} \subseteq \mathcal{Q}_1$.

For any $Q \in \mathcal{Q}_1$, since $\mathcal{H}(Q) - Q = r^* e$, we have $p_{\mathrm{span}}(\mathcal{H}(Q) - Q) = p_{\mathrm{span}}(r^* e) = 0$. As a result, we have $Q \in \mathcal{Q}_2$, implying $\mathcal{Q}_1 \subseteq \mathcal{Q}_2$.

Finally, to show that $\mathcal{Q}_2 \subseteq \mathcal{Q}$, it is enough to show that for any $Q \in \mathcal{Q}_2$, we must have $Q - Q^* \in \ker(p_{\mathrm{span}})$. To see this, since $Q \in \mathcal{Q}_2$, there exists $c \in \mathbb{R}$ such that $\mathcal{H}(Q) - Q = ce$. It follows that

$$
\begin{aligned}
p_{\mathrm{span}}(Q - Q^*) &= p_{\mathrm{span}}(\mathcal{H}(Q) - ce - \mathcal{H}(Q^*) + r^* e) \\
&= p_{\mathrm{span}}(\mathcal{H}(Q) - \mathcal{H}(Q^*)) \\
&\leq \beta \, p_{\mathrm{span}}(Q - Q^*),
\end{aligned}
$$

where the last line follows from Assumption 2.1. Therefore, we must have $p_{\mathrm{span}}(Q - Q^*) = 0$, implying $Q - Q^* \in \ker(p_{\mathrm{span}})$.

### C.3 Proof of Lemma 4.1

Given any $Q_1, Q_2 \in \mathbb{R}^{|\mathcal{S}||\mathcal{A}|}$, for simplicity of notation, denote

$$
M = \max_{s,a}([\mathcal{H}(Q_1)](s,a) - [\mathcal{H}(Q_2)](s,a)), \qquad m = \min_{s,a}([\mathcal{H}(Q_1)](s,a) - [\mathcal{H}(Q_2)](s,a)),
$$

$$
L = \max_{s,a}(Q_1(s,a) - Q_1(s,a)), \qquad\qquad \ell = \min_{s,a}(Q_1(s,a) - Q_1(s,a)).
$$

To show the seminorm contraction property of the asynchronous Bellman operator $\bar{\mathcal{H}}(\cdot)$, we first show that $M \leq L$ and $m \geq \ell$. By definition of the Bellman operator $\mathcal{H}(\cdot)$, we have

$$
\begin{aligned}
M &= \max_{s,a} \sum_{s' \in S} p(s'|s,a) \left( \max_{a'} Q_1(s',a') - \max_{a''} Q_2(s',a'') \right) \\
&\leq \max_{s,a} \sum_{s' \in S} p(s'|s,a) \max_{a'}(Q_1(s',a') - Q_2(s',a')) \\
&\leq \max_{s,a}(Q_1(s,a) - Q_2(s,a)) \\
&= L.
\end{aligned}
$$

Similarly, we also have

$$
\begin{aligned}
m &= \min_{s,a} \sum_{s' \in S} p(s'|s,a) \left( \max_{a'} Q_1(s',a') - \max_{a''} Q_2(s',a'') \right) \\
&\geq \min_{s,a} \sum_{s' \in S} p(s'|s,a) \min_{a'}(Q_1(s',a') - Q_2(s',a')) \\
&\geq \min_{s,a}(Q_1(s,a) - Q_2(s,a)) \\
&= \ell.
\end{aligned}
$$

Now that we have established $M \leq L$ and $m \geq \ell$, to proceed with the proof, using the definition of $\bar{\mathcal{H}}(\cdot)$, we have

$$
\begin{aligned}
&p_{\mathrm{span}}(\bar{\mathcal{H}}(Q_1) - \bar{\mathcal{H}}(Q_2)) \\
&= p_{\mathrm{span}}((I - D)(Q_1 - Q_2) + D(\mathcal{H}(Q_1) - \mathcal{H}(Q_2))) \\
&= \frac{1}{2} \max_{s,a} \left[ (1 - D(s,a))(Q_1(s,a) - Q_2(s,a)) + D(s,a)([\mathcal{H}(Q_1)](s,a) - [\mathcal{H}(Q_2)](s,a)) \right] \\
&\quad - \frac{1}{2} \min_{s,a} \left[ (1 - D(s,a))(Q_1(s,a) - Q_2(s,a)) + D(s,a)([\mathcal{H}(Q_1)](s,a) - [\mathcal{H}(Q_2)](s,a)) \right].
\end{aligned}
$$

On the one hand, we have

$$
\begin{aligned}
&\max_{s,a} \left[ (1 - D(s,a))(Q_1(s,a) - Q_2(s,a)) + D(s,a)([\mathcal{H}(Q_1)](s,a) - [\mathcal{H}(Q_2)](s,a)) \right] \\
&\leq \max_{s,a} \left[ (1 - D(s,a))L + D(s,a)M \right] \\
&= \max_{s,a} \left[ L - D(s,a)(L - M) \right]
\end{aligned}
$$

$$= L - D_{\min}(L - M),$$

where the last inequality follows from $M \leq L$.

On the other hand, we have

$$\min_{s,a} \left[ (1 - D(s,a))(Q_1(s,a) - Q_2(s,a)) + D(s,a)([\mathcal{H}(Q_1)](s,a) - [\mathcal{H}(Q_2)](s,a)) \right]$$

$$\geq \min_{s,a} \left[ (1 - D(s,a))\ell + D(s,a)m \right]$$

$$= \min_{s,a} \left[ \ell - D(s,a)(\ell - m) \right]$$

$$= \ell - D_{\min}(\ell - m),$$

where the last inequality follows from $m \geq \ell$.

Together, we obtain

$$p_{\mathrm{span}}(\bar{\mathcal{H}}(Q_1) - \bar{\mathcal{H}}(Q_2)) \leq \frac{1}{2}(L - D_{\min}(L - M)) - \frac{1}{2}(\ell - D_{\min}(\ell - m))$$

$$= \frac{1}{2}(1 - D_{\min})(L - \ell) + \frac{1}{2}D_{\min}(M - m).$$

Since $\mathcal{H}(\cdot)$ is a contraction mapping with respect to $p_{\mathrm{span}}(\cdot)$ (cf. Assumption 2.1), we have

$$p_{\mathrm{span}}(\mathcal{H}(Q_1) - \mathcal{H}(Q_2)) = \frac{M - m}{2} \leq \beta p_{\mathrm{span}}(Q_1 - Q_2) = \frac{\beta(L - \ell)}{2}.$$

It follows that

$$p_{\mathrm{span}}(\bar{\mathcal{H}}(Q_1) - \bar{\mathcal{H}}(Q_2)) \leq \frac{1}{2}(1 - D_{\min})(L - \ell) + \frac{1}{2}D_{\min}(M - m)$$

$$\leq \frac{1}{2}(1 - D_{\min})(L - \ell) + \frac{1}{2}D_{\min}\beta(L - \ell)$$

$$= \frac{1}{2}(1 - D_{\min}(1 - \beta))(L - \ell)$$

$$= (1 - D_{\min}(1 - \beta))p_{\mathrm{span}}(Q_1 - Q_2).$$

Finally, since $D_{\min} \in (0, 1)$ under Assumption 3.1, which implies $\bar{\beta} := (1 - D_{\min}(1 - \beta)) \in (0, 1)$, the operator $\bar{\mathcal{H}}(\cdot)$ is a contraction mapping with respect to $p_{\mathrm{span}}(\cdot)$, with contraction factor $\bar{\beta}$.

### C.4   Proof of Proposition 4.1

Using the definition of $\bar{\mathcal{H}}(\cdot)$, if $D = I/(|\mathcal{S}||\mathcal{A}|)$, we have

$$p_{\mathrm{span}}(\bar{\mathcal{H}}(Q) - Q) = 0 \quad \Longleftrightarrow \quad p_{\mathrm{span}}(D(\mathcal{H}(Q) - Q)) = 0,$$

$$\Longleftrightarrow \quad \frac{1}{|\mathcal{S}||\mathcal{A}|}p_{\mathrm{span}}(\mathcal{H}(Q) - Q) = 0,$$

$$\Longleftrightarrow \quad p_{\mathrm{span}}(\mathcal{H}(Q) - Q) = 0,$$

which implies

$$\{Q|p_{\mathrm{span}}(\bar{\mathcal{H}}(Q) - Q) = 0\} = \{Q|p_{\mathrm{span}}(\mathcal{H}(Q) - Q) = 0\}.$$

If $D \neq I/(|\mathcal{S}||\mathcal{A}|)$, let $\tilde{Q} \in \mathcal{Q} = \{Q^* + ce|c \in \mathbb{R}\} = \{Q|p_{\mathrm{span}}(\mathcal{H}(Q) - Q) = 0\}$ be arbitrary. Then, we have

$$p_{\mathrm{span}}(\bar{\mathcal{H}}(\tilde{Q}) - \tilde{Q}) = p_{\mathrm{span}}(D(\mathcal{H}(\tilde{Q}) - \tilde{Q}))$$

$$= p_{\mathrm{span}}(Dr^*e)$$

$$= |r^*| \left( \max_{s,a} D(s,a) - \min_{s,a} D(s,a) \right)$$

$$> 0.$$

Since $p_{\text{span}}(\bar{\mathcal{H}}(\tilde{Q}) - \tilde{Q}) \neq 0$ for any $\tilde{Q}$ in the set of solutions $\mathcal{Q}$ of the original Bellman equation (2), we must have

$$\{Q | p_{\text{span}}(\bar{\mathcal{H}}(Q) - Q) = 0\} \cap \{Q | p_{\text{span}}(\mathcal{H}(Q) - Q) = 0\} = \emptyset.$$

To establish the convergence rate of $\mathbb{E}[p_{\text{span}}(Q_k - \bar{Q}^*)^2]$, we will apply [24, Theorem 4.1]. For the completeness of this work, we first restate [24, Theorem 4.1] for the special case of span seminorm contractive SA in the following.

**Theorem C.1** (Theorem 4.1 from [24]). *Consider a Markovian SA of the form*

$$x_{k+1} = x_k + \alpha_k(\mathcal{T}(x_k, Y_k) - x_k), \quad \forall k \geq 1,$$

*where $\{Y_k\}$ is a stochastic process taking values in a finite set $\mathcal{Y}$ and $\mathcal{T} : \mathbb{R}^d \times \mathcal{Y} \to \mathbb{R}^d$ is an operator. Suppose that the following assumptions are satisfied:*

*(1) The stochastic process $\{Y_k\}$ is a uniformly ergodic Markov chain with stationary distribution $\nu$.*

*(2) The operator $\bar{\mathcal{T}} : \mathbb{R}^d \to \mathbb{R}^d$ defined as $\bar{\mathcal{T}}(x) = \mathbb{E}_{Y \sim \nu}[\mathcal{T}(x, Y)]$ is a contraction mapping with respect to $p_{\text{span}}(\cdot)$, with contraction factor $\beta' \in (0, 1)$.*

*(3) The operator $\mathcal{T}(\cdot)$ satisfies $p_{\text{span}}(\mathcal{T}(x_1, y) - \mathcal{T}(x_2, y)) \leq A_1 p_{\text{span}}(x_1 - x_2)$ and $p_{\text{span}}(\mathcal{T}(0, y) \leq B_1$ for any $x_1, x_2 \in \mathbb{R}^d$ and $y \in \mathcal{Y}$, where $A_1, B_1 > 0$ are constants.*

*Then, when using $\alpha_k = \alpha/(k + h)$ with appropriately chosen $\alpha$ and $h$, we have $\mathbb{E}[p_{\text{span}}(x_k - x^*)^2] \leq \tilde{\mathcal{O}}(1/k)$, where $x^*$ is a particular solution to $p_{\text{span}}(\bar{\mathcal{T}}(x) - x) = 0$, which is guaranteed to exist due to the seminorm fixed-point theorem [24, Theorem 2.1].*

To apply Theorem C.1 to $Q$-learning with universal stepsizes as described in Eq. (3), we only need to verify Assumption (3) because Assumption (1) holds under Assumption 3.1 and Assumption (2) has been verified in Lemma 4.1.

Using the definition of $G(\cdot)$, for any $Q_1, Q_2 \in \mathbb{R}^{|\mathcal{S}||\mathcal{A}|}$ and $y = (s_0, a_0, s_1) \in \mathcal{Y}$, observe that the vector $G(Q_1, y) - G(Q_2, y)$ and the vector $Q_1 - Q_2$ differ by exactly one entry, namely the $(s_0, a_0)$-th one. Therefore, let $Q_{\text{diff}} \in \mathbb{R}^{|\mathcal{S}||\mathcal{A}|}$ be defined as

$$Q_{\text{diff}}(s_0, a_0) = [G(Q_1, y)](s_0, a_0) - [G(Q_2, y)](s_0, a_0) - (Q_1(s_0, a_0) - Q_2(s_0, a_0))$$
$$= \max_{a'} Q_1(s_1, a') - \max_{a'} Q_2(s_1, a') - (Q_1(s_0, a_0) - Q_2(s_0, a_0)),$$

and $Q_{\text{diff}}(s, a) = 0$ for any $(s, a) \neq (s_0, a_0)$. Then, we have

$$G(Q_1, y) - G(Q_2, y) = Q_1 - Q_2 + Q_{\text{diff}}.$$

By the triangle inequality, we have

$$p_{\text{span}}(G(Q_1, y) - G(Q_2, y)) \leq p_{\text{span}}(Q_1 - Q_2) + p_{\text{span}}(Q_{\text{diff}}). \tag{51}$$

To bound $p_{\text{span}}(Q_{\text{diff}})$, since $Q_{\text{diff}}$ has only one non-zero entry, we have by the definition of $p_{\text{span}}(\cdot)$ that

$$p_{\text{span}}(Q_{\text{diff}}) = \frac{1}{2} \left| \max_{a'} Q_1(s_1, a') - \max_{a'} Q_2(s_1, a') - (Q_1(s_0, a_0) - Q_2(s_0, a_0)) \right|.$$

To further bound the absolute value on the right-hand side of the previous inequality, observe that on the one hand, we have

$$\max_{a'} Q_1(s_1, a') - \max_{a'} Q_2(s_1, a') - (Q_1(s_0, a_0) - Q_2(s_0, a_0))$$
$$\leq \max_{s', a'}(Q_1(s', a') - Q_2(s', a')) - (Q_1(s_0, a_0) - Q_2(s_0, a_0))$$
$$\leq \max_{s', a'}(Q_1(s', a') - Q_2(s', a')) - \min_{s', a'}(Q_1(s', a') - Q_2(s', a'))$$
$$= 2p_{\text{span}}(Q_1 - Q_2).$$

On the other hand, since

$$\max_{a'} Q_1(s_1, a') - \max_{a'} Q_2(s_1, a') = \max_{a'} Q_1(s_1, a') - Q_2(s_1, \underline{a})$$

$$\text{(Denote } \underline{a} \in \arg\max_{a'} Q_2(s_1, a'))$$

$$\geq Q_1(s_1, \underline{a}) - Q_2(s_1, \underline{a})$$
$$\geq \min_{s', a'} (Q_1(s', a') - Q_2(s', a')),$$

we have

$$\max_{a'} Q_1(s_1, a') - \max_{a'} Q_2(s_1, a') - (Q_1(s_0, a_0) - Q_2(s_0, a_0))$$
$$\geq \min_{s', a'} (Q_1(s', a') - Q_2(s', a')) - (Q_1(s_0, a_0) - Q_2(s_0, a_0))$$
$$\geq \min_{s', a'} (Q_1(s', a') - Q_2(s', a')) - \max_{s', a'} (Q_1(s', a') - Q_2(s', a'))$$
$$= -2 p_{\text{span}}(Q_1 - Q_2)$$

It follows that

$$p_{\text{span}}(Q_{\text{diff}}) = \frac{1}{2} \left| \max_{a'} Q_1(s_1, a') - \max_{a'} Q_2(s_1, a') - (Q_1(s_0, a_0) - Q_2(s_0, a_0)) \right|$$
$$\leq p_{\text{span}}(Q_1 - Q_2).$$

Combining the previous inequality with Eq. (51), we obtain

$$p_{\text{span}}(G(Q_1, y) - G(Q_2, y)) \leq 2 p_{\text{span}}(Q_1 - Q_2).$$

To show that $p_{\text{span}}(G(0, y))$ is uniformly bounded for all $y \in \mathcal{Y}$, observe that $G(0, y)$ has only one non-zero entry, i.e., the $(s_0, a_0)$-th one, which is equal to $\mathcal{R}(s_0, a_0)$. Therefore, using the definition of $G(\cdot)$, we have

$$\max_{y=(s_0, a_0, s_1) \in \mathcal{Y}} p_{\text{span}}(G(0, y)) = \frac{1}{2} \max_{s_0, a_0} |\mathcal{R}(s_0, a_0)| \leq \frac{1}{2}.$$

### C.5 Proof of Lemma 4.2

Using the definition of $\bar{\mathcal{H}}(\cdot)$ (cf. Eq. (5)), since $p_{\text{span}}(\bar{\mathcal{H}}(\bar{Q}^*) - \bar{Q}^*) = 0$, there must exist some $c_1 \in \mathbb{R}$ such that $D(\mathcal{H}(\bar{Q}^*) - \bar{Q}^*) = c_1 e$, which implies

$$\mathcal{H}(\bar{Q}^*) - \bar{Q}^* = c_1 D^{-1} e.$$

On the other hand, we know that

$$\mathcal{H}(Q^*) - Q^* = r^* e.$$

Subtracting the two equations yields

$$\bar{Q}^* - Q^* = \mathcal{H}(\bar{Q}^*) - \mathcal{H}(Q^*) + r^* e - c_1 D^{-1} e.$$

Taking the span seminorm on both sides and using Assumption 2.1, we have

$$p_{\text{span}}(\bar{Q}^* - Q^*) \leq \frac{|c_1| \, p_{\text{span}}(D^{-1} e)}{1 - \beta}.$$

