# OpenReview forum: "Non-Asymptotic Guarantees for Average-Reward Q-Learning with Adaptive Stepsizes"
_NeurIPS.cc/2025/Conference — NeurIPS 2025 poster_

### Official Review · Reviewer_pG1e · 2025-06-22

**Clarity:** 4
**Significance:** 3
**Originality:** 3
**Rating:** 5
**Confidence:** 3

**Summary:**

This paper delivers the first non-asymptotic analysis of average-reward asynchronous Q-learning. Due to the asynchronous sampling, the authors show the universal stepsizes cannot guarantee convergence. Motivated by the importance-sampling, the authors design an adaptive stepsize and establish an $O(1/k)$ convergence rate. To establish the convergence, the authors reformulate the non-Markovian update as a time-inhomogeneous Markovian SA and breaks the correlation between iterates and data through two main steps: 1. demonstrating that the span of $Q_k$ grows at most $O(\log k)$ and 2. controlling the data sequence via conditioning arguments and appropriate Markov concentration bounds.

**Questions:**

It may be due to my lack of expertise in averaged-reward RL, I have some questions about the related work and the main theorem. I hope the following questions would help the authors improve the paper.

Q1. Theorem 3.1 makes no assumptions on the initial point $Q_1$. Should the constant in the upper bound therefore depend on  $||Q_1||_{\text{span}}$ (or a comparable measure of its magnitude)?

Q2. This work focuses on the adaptive stepsize: $\alpha_k(s,a) = \alpha/(N_k(s,a)+h)$ and achieves a $O(1/k)$ convergence rate with some condition on $\alpha$. In the Markovian-SA literature (e.g. [21]), one can obtain $O(1/k^\xi)$ for $0<\xi<1$ with a broader rule of stepsize $\alpha_k = \alpha/(k+h)^\xi$ without imposing extra conditions on $\alpha$. Would it be straightforward, or are there hidden technical obstacles, to extend your proof to the more robust choice, with no extra conditions on $\alpha$?

Q3. I understand that convergence in a semi-norm guarantees only convergence to a set, the paper would benefit from a discussion of how your techniques could be adapted—either through algorithm design or deeper analysis—to produce a consistent estimate of the true $Q^{*}$.

**Ethical Concerns:**

["NO or VERY MINOR ethics concerns only"]

**Limitations:**

Yes.

**Paper Formatting Concerns:**

I didn’t notice any major formatting issues in this paper.

**Quality:**

4

**Strengths And Weaknesses:**

Strengths:

S1. The paper delivers the first finite-time performance bound for average-reward asynchronous Q-learning.

S2. It extends classical Lyapunov-based analyses for Markovian SA to settings with time-inhomogeneous Markovian noise—a contribution that, to my knowledge, is new to the literature. The technique appears broadly applicable to a wider class of SA algorithms with adaptive stepsizes.

S3. The use of adaptive stepsizes derived through an importance-sampling argument is particularly interesting.

S4. This is a theoretically solid work. The paper is relatively well-written. The proof seems correct, although I did not check line by line.

Weaknesses:

Please see questions below.

---

> ### Author Rebuttal · Authors · 2025-07-27
>
> We thank the reviewer for the encouraging comments regarding the contributions of this work. Please find our point-by-point responses to all the questions below.
>
> >Question: Theorem 3.1 makes no assumptions on the initial point $Q_1$. Should the constant in the upper bound therefore depend on $p_{span}(Q_1)$ (or a comparable measure of its magnitude)?
>
> **Response:** The reviewer is absolutely correct. To clarify, our finite-time bound is different for $k \leq K$ and $k \geq K$, where $K$ is a threshold that roughly captures the mixing of the Markov chain {$(S_k, A_k)$} induced by the behavior policy $\pi$. For $k \leq K$, we bound $p_{span}(Q_k - Q^\star)$ as
>
> $p_{span}(Q_k - Q^\star)\leq p_{span}(Q_k) + p_{span}( Q^\star )\leq b_k+p_{span}(Q^\star)=m_k$,
>
> where $b_k$ is the time-varying almost-sure bound of $p_{span}(Q_k)$ established in Proposition B.1, and $m_k$ is defined right before Theorem 3.1. Note that in the proof of Proposition B.1, we assume without loss of generality that $p_{span}(Q_1) = 0$, which will be made explicit in the next version of this work. This can be easily achieved by initializing $Q_1$ as a constant vector.
>
> As for $k \geq K$ (which is the more interesting case), the bound naturally depends on the "initial error" $p_{span}(Q_K - Q^\star)$. However, because $Q_K$ is inherently random, to maintain the mathematical rigor of our result, we apply the following inequality to ensure that the bound does not depend on random quantities:
>
> $p_{span}(Q_K - Q^\star) \leq p_{span}(Q_K) + p_{span}(Q^\star) \leq b_K + p_{span}(Q^\star) = m_K$,
>
> which is the term that multiplies the leading factor in all cases of Theorem 3.1.
>
> To summarize, if one accepts having $p_{span}(Q_K - Q^\star)$ in the finite-time bound, one can simply replace $m_K$ by $\mathbb{E}[p_{span}(Q_K - Q^\star)^2]$ in Theorem 3.1 so that the theorem captures the dependence on the "initial error" $Q_K - Q^*$.
>
>
>
> >Question: This work focuses on the adaptive stepsize: $\alpha_k(s, a) = \alpha / (N_k(s, a) + h)$ and achieves a $O(1/k)$ convergence rate with some condition on $\alpha$. In the Markovian-SA literature (e.g. [21]), one can obtain $O(1/k^\xi)$ for $0 < \xi < 1$ with a broader rule of stepsize $\alpha_k = \alpha / (k + h)^\xi$ without imposing extra conditions on $\alpha$. Would it be straightforward, or are there hidden technical obstacles, to extend your proof to the more robust choice, with no extra conditions on $\alpha$?
>
> **Response:**  This is a very good question. We next elaborate on the algorithmic design of Q-learning under alternative choices of adaptive stepsizes, which is not as straightforward as directly setting $\alpha_k(s,a) = \alpha / (N_k(s,a) + h)^\xi$. We then discuss the potential challenges that arise from the analysis.
>
> We start with Eq. (9) from the submission:
>
> $Q_{k+1}(s,a) = Q_k(s,a) + \tilde{\alpha}_k \cdot \frac{\mathbb{1}{(S_k, A_k) = (s,a)}}{D_k(s,a)} \cdot \delta_k$
>
> where $D_k$ is the empirical frequency matrix and $\delta_k$ denotes the temporal difference. As illustrated in Section 4.2, the above equation is the correct way of viewing average-reward Q-learning, where $D_k$ serves as implicit importance sampling. When choosing $\tilde{\alpha}_k = \alpha / (k + h)$, we arrive at the Q-learning algorithm presented in this work. However, we can also set $\tilde{\alpha}_k$ to be either a polynomial stepsize or a constant stepsize, as illustrated below.
>
> *Polynomial stepsizes:*
> Analogous to using $\alpha_k = \alpha / (k+h)^\xi$ in Markovian stochastic approximation, by setting $\tilde{\alpha}_k = \alpha / (k+h)^\xi$, we arrive at the following Q-learning algorithm:
>
> $Q_{k+1}(s,a) = Q_k(s,a) + \frac{\alpha}{(k + h)^\xi} \cdot \frac{\mathbb{1}_{{(S_k, A_k) = (s,a)}}}{D_k(s,a)} \cdot \delta_k = Q_k(s,a) + \frac{\alpha \cdot (k + h)^{1 - \xi}}{N_k(s,a) + h} \cdot \mathbb{1} _{(S_k, A_k) = (s,a)} \cdot \delta_k$,
>
> where we recall that $D_k(s,a) = (N_k(s,a) + h)/(k + h)$. In this case, the algorithm can be viewed as using adaptive stepsizes of the form $\alpha_k(s,a) =\alpha (k + h)^{1 - \xi}/(N_k(s,a) + h)$. When using this stepsize, we expect similar convergence behavior as in Markovian stochastic approximation with $\alpha_k = \alpha / (k + h)^\xi$, robustly achieving an $\mathcal{O}(1 / k^\xi)$ rate of convergence independent of the choice of $\alpha$.
>
> *Constant stepsizes:*
> Similarly, by setting $\tilde{\alpha}_k \equiv \alpha$ in Eq. (9), we obtain:
>
> $Q_{k+1}(s,a) = Q_k(s,a) + \alpha \cdot \frac{\mathbb{1}{{(S_k, A_k) = (s,a)}}}{D_k(s,a)} \cdot \delta_k = Q_k(s,a) + \frac{\alpha \cdot (k + h)}{N_k(s,a) + h} \cdot \mathbb{1}_ {(S_k, A_k) = (s,a)} \cdot \delta_k$
>
> In this case, the adaptive stepsize can be understood as $\alpha_k(s,a) = \alpha (k + h)/(N_k(s,a) + h)$. When using this stepsize, we expect similar convergence behavior as in Markovian stochastic approximation with constant stepsize, achieving geometric convergence to a ball with radius proportional to $\alpha$. Moreover, we expect that existing studies on constant stepsize Markovian stochastic approximation [32,81]—such as using Polyak averaging for variance reduction, RR-extrapolation for asymptotic bias reduction, and weak convergence behavior—are extendable to our setting.
>
> *Challenges in the Analysis:*  To extend our result to the above variants of adaptive stepsizes, one must address the challenge of non-Markovian noise introduced by the use of adaptive stepsizes, as in this work. Another challenge we anticipate is that $\alpha_k(s,a)$ may not lie in the interval $(0,1)$. Specifically, when using $\alpha_k(s,a) = \alpha / (N_k(s,a) + h)$, as long as $h \geq \alpha$, the stepsize is always in the interval $(0,1)$ with probability one. However, when using the above variants—for example, $\alpha_k(s,a) = \alpha (k + h)/(N_k(s,a) + h)$—no matter how small $\alpha$ is, $\alpha_k(s,a)$ lies in $(0,1)$ only with high probability for any fixed large enough $k$. Nevertheless, we do not expect this is a fundamental challenge and anticipate that it can be overcome with some careful Markov chain concentration analysis.
>
> >Question: I understand that convergence in a semi-norm guarantees only convergence to a set, the paper would benefit from a discussion of how your techniques could be adapted—either through algorithm design or deeper analysis—to produce a consistent estimate of the true $Q^\star$.
>
> **Response:** This is an excellent question. For the current algorithm design, we do not expect pointwise convergence. To achieve this, recall the algorithm design for classical RVI Q-learning [1] (see also the last paragraph in Section 3.1 of our submission). By subtracting an offset $f(Q)$ within the temporal difference, RVI Q-learning achieves pointwise convergence to a solution $Q' \in \mathcal{Q}:=Q^\star +$ {$ce \mid c \in \mathbb{R}$} satisfying $f(Q') = r^*$. We believe similar ideas could be incorporated into our algorithm to achieve pointwise convergence to some point in $\mathcal{Q}$, though convergence to the exact $Q^\star$ may not be guaranteed. This is an interesting future direction.
>
> That being said, as the goal of Q-learning is to find an optimal policy by choosing actions greedily based on the output Q-function estimate, an error in the direction of the all-ones vector does not matter, and achieving set convergence to $\mathcal{Q}$ (or equivalently, convergence in $p_{span}(\cdot)$ to $Q^*$) is sufficient.

---

> > ### Author Response · Authors · 2025-08-04
> > **Response to Reviewer Comments**
> >
> > We thank the reviewer for the valuable comments. We hope our response has successfully addressed the reviewer’s concerns and answered their questions. If there are any remaining issues, we would be happy to continue the discussion. Otherwise, we would greatly appreciate the reviewer’s consideration in updating their score in light of our clarifications.

---

> > ### Comment · Reviewer_pG1e · 2025-08-04
> >
> > The authors have done a good job in responding to my previous comments. I find their responses clear and detailed. I have no further questions and will maintain my score.

---

### Official Review · Reviewer_pbot · 2025-06-28

**Clarity:** 3
**Significance:** 2
**Originality:** 2
**Rating:** 4
**Confidence:** 3

**Summary:**

In this paper, the authors propose a variant of the Average-Reward Q-learning algorithm with an adaptive step-size based on the counters of each state-action pair. They provide a finite-time analysis for the last-iterate convergence in the span seminorm, based on a single trajectory of Markovian samples. Additionally, they demonstrate that standard Q-learning with a diminishing step-size might fail to converge to the same optimum $Q^*$. The derived convergence rate is optimal in terms of the accuracy level $\varepsilon$ for $\varepsilon$ small enough, resulting in sample complexity $O(\varepsilon^{-2})$.

**Questions:**

1. Can Assumption 2.1 be relaxed, at least to cover the setting of uniformly geometrically ergodic chain (s_k,a_k,s_{k+1})?
2. Assumption 3.1 requires a fixed behavioral policy with full support. In practice, behaviour policies are often learned, sometimes in a particular parametric family. How the derived bounds scale with the minimum visitation probability of a state-action pair $(s,a)$, that is, with $\min_{s,a}\pi(a|s)$?

**Ethical Concerns:**

["NO or VERY MINOR ethics concerns only"]

**Final Justification:**

I would like to thank the authors for their clarifications. I am not an expert in the average-reward RL setting, and appreciate the detailed response. I think the paper will greatly benefit from including the outlined discussion (especially about assumption 2.1 and challenges related with the geometric ergodicity and multi-step contraction) to the paper.

Regarding the numerical experiments, I do not insist on including them, as this is a purely theoretical paper. I think that including a toy experiment will not add much to the value of the paper. The main purpose of my comment was to question the practical applicability of adaptive step-size versions of Q-learning.

That being said, I am happy to increase my score and wish the authors good luck.

**Limitations:**

Yes

**Paper Formatting Concerns:**

No concerns.

**Quality:**

3

**Strengths And Weaknesses:**

This paper is well-structured and rather easy to follow. The paper suggests an MSE bounds for the last-iterate for average-reward Q-learning with a single Markovian data trajectory. These bounds are, to the best of my understanding, new.

***Weaknesses***

Main weaknesses, to my opinion, are limited theoretical novelty and overly strong theoretical assumptions.

First, regarding the theoretical assumptions - theoretical guarantees, developed by the authors, rely on the Assumption 2.1 that the Bellman operator is a \(\beta\)-contraction in the span seminorm. The authors suggest a sufficient condition (which is a total-variation contraction of the transition kernel in $1$ step), which is overly strong. This assumption, in fact, is even stronger compared to a unifrom geometric ergodicity of transition kernel.

Regarding the theoretical novelty, it seems that significant parts of the analysis adapts recent papers on seminorm-contractive SA [21,22] in the current submission. The contribution would be much clearer if the authors highlight the new elements of the current submission from those which readily follows from the past stream of works.

Finally, although the work is primarily theoretical, it lacks experimental validation of the results. Moreover, the literature review shows that while adaptive step-sizes have been explored in prior works on discounted Q-learning, no works have attempted to apply them in practice.

***Minor points***
Line 132: I do not understand where "(to be discussed shortly)" refers to

---

> ### Author Rebuttal · Authors · 2025-07-28
>
> >Comment: The main weaknesses, in my opinion, are limited theoretical novelty and overly strong theoretical assumptions.
>
> **Response:** We thank the reviewer for the comments. While we agree that Assumption 2.1 is indeed a strong assumption, we believe the reviewer may have misunderstood the nature of our technical contributions. Below, we provide a detailed point-by-point response to the reviewer's concerns.
>
> >Comment: Assumption 2.1 being strong, which is also related to the question of relaxing Assumption 2.1.
>
> **Response:** We agree with the reviewer that Assumption 2.1 is stronger than assuming uniform ergodicity of the transition kernel, in the sense that uniform ergodicity only implies a multi-step contraction of the Bellman operator, not a single-step contraction.
>
> Note that such a multi-step contraction property has been used in the existing study of synchronous Q-learning [80], where a stochastic estimator of the multi-step Bellman operator is obtainable precisely due to the availability of a generative model that enables free i.i.d. sampling. However, in the much more challenging asynchronous setting, to our knowledge, no techniques have been developed (and in our opinion, such techniques may be impossible) to obtain an estimator of the multi-step Bellman operator, due to the presence of only a single trajectory of samples and the nonlinearity of the Bellman operator. Therefore, having a multi-step contraction (implied by uniform ergodicity) does not benefit the analysis, and we view our assumption as somewhat necessary if one wishes to study Q-learning from a span-seminorm contractive stochastic approximation perspective.
>
> That being said, as mentioned in the conclusion of this work, relaxing such an assumption to more realistic ones—such as the MDP being weakly communicating—would require completely abandoning the contractive viewpoint of the Bellman operator and instead studying Q-learning as a stochastic approximation with a non-expansive operator (which always holds). To our knowledge, a finite-time analysis of asynchronous Q-learning from this perspective does not yet exist in the literature and is, in fact, the next step of this line of work.
>
> >Comment: The theoretical novelty of this work, especially relative to [21].
>
> **Response:** We thank the reviewer for raising this comment. To clarify, the technical contributions of this work are in fact significant and do not follow directly from, nor are they a minor extension of, [21, 22]. We have highlighted the unique technical challenges and our techniques in multiple places throughout the paper: in the second paragraph of the abstract, as a bullet point in Section 1 (the last paragraph), in the proof sketch in Section 5, and in the full proof in Appendix B, especially Appendix B.3. Next, we elaborate in more detail our technical contributions relative to [21]. Note that [22] is not particularly relevant, as it studies concentration bounds for norm-contractive stochastic approximation with i.i.d. noise, whereas this work studies mean-square bounds for average-reward Q-learning with adaptive stepsizes, which is inherently a seminorm-contractive stochastic approximation with non-Markov noise.
>
> In general, to perform a finite-time analysis of a stochastic approximation algorithm using a Lyapunov-drift argument, there are two main steps in the proof:
>
> (1) constructing a valid Lyapunov function and showing that the "deterministic" part of the algorithm induces a negative drift with respect to this Lyapunov function;
>
> (2) showing that the errors due to discretization and, more importantly, stochasticity are dominated by the negative drift.
>
> As for part (1), due to the seminorm-contractive property of the Bellman operator, a valid Lyapunov function based on the generalized Moreau envelope has already been constructed in the literature. Therefore, as clearly stated on page 9, line 353 of the submission—"inspired by [18, 21, 24, 81],..."—we do not claim any technical contribution in constructing a valid Lyapunov function; this credit goes to [21].
>
> Part (2)—i.e., controlling the error due to stochasticity—is where the unique challenges arise in Q-learning with adaptive stepsizes. In the existing literature, when the noise sequence is i.i.d., a martingale, or forms a uniformly ergodic Markov chain, the stochastic error can be handled either through a simple conditioning argument (in the i.i.d. and martingale cases) or through a more sophisticated conditioning argument based on mixing time (in the Markovian case), as used in [21]. However, in our case, due to the use of adaptive stepsizes (which, as illustrated in Section 4, are necessary) that depend on the entire history of the sample path, the Q-learning algorithm—viewed as a stochastic approximation—is not even Markovian. This leads to the unique technical contribution of this work: handling non-Markovian noise within Q-learning.
>
> The first step in our proof (illustrated in detail in Section 5, line 339) is to reformulate the algorithm as a Markovian stochastic approximation. Specifically, we reformulate the Q-learning algorithm as a Markovian stochastic approximation driven by the Markov chain {$Z_k = (D_k, S_k, A_k, S_{k+1})$}, where $D_k$ is the empirical frequency matrix and $(S_k, A_k, S_{k+1})$ captures the latest transition. However, while {$Z_k$} is a Markov chain, it is time-inhomogeneous and lacks geometric mixing—precisely due to incorporating $D_k$ into its definition. This is discussed in detail in Appendix B.1. Note that geometric mixing is typically required in the existing literature on Markovian stochastic approximation [65].
>
>
> To further handle the time-inhomogeneous Markovian noise {$Z_k$}, we developed an approach based on first showing that the stochastic iterate $Q_k$, while not uniformly bounded by a constant (in stark contrast to discounted Q-learning [29]), satisfies a time-varying almost-sure bound that grows at most logarithmically with $k$ (cf. Proposition B.1). These two properties (i.e., almost-sure bound + logarithmic growth) enable us to decouple the iterate $Q_k$ from the time-inhomogeneous Markovian noise $Z_k$. After decoupling, we handle the randomness in $Z_k = (D_k, S_k, A_k, S_{k+1})$ separately, where two challenges arise: (1) the correlation between $D_k$ and $(S_k, A_k, S_{k+1})$, and (2) the presence of the empirical frequency $D_k(s,a)$ in the denominator of the algorithm update, which breaks linearity. To address the first challenge, we use a conditioning argument based on the fast mixing of $(S_k, A_k, S_{k+1})$ (cf. Assumption 3.2). To tackle the second, we employ Markov chain concentration inequalities and the $\mathcal{L}^2$ weak law of large numbers for functions of Markov chains. This is illustrated at a high level in Section 5 and elaborated in detail with clear intuition in Appendix B.3.
>
> Overall, we believe that the technical novelties of this work—specifically, handling non-Markovian noise in Q-learning with adaptive stepsizes—represent a significant contribution. In addition to the technical contributions, the conceptual contributions of this work are also noteworthy. In particular, identifying the necessity of using adaptive stepsizes and leveraging them as a means of implicit importance sampling are important insights for the future study of the reinforcement learning community.
>
> >Comment: Numerical simulations and practical implementations of Q-learning
>
> **Response:** We will add numerical experiments to verify the performance of Q-learning with adaptive stepsizes in the next version of this work.
>
>
>
> >Minor points Line 132: I do not understand where "(to be discussed shortly)" refers to
>
> **Response:**  It refers to the paragraph after Assumption 2.1, where we discuss the seminorm contraction of the Bellman operator. We will make this clearer in the next version of this work.
>
> >Question: Assumption 3.1 requires a fixed behavioral policy with full support. In practice, behaviour policies are often learned, sometimes in a particular parametric family. How the derived bounds scale with the minimum visitation probability of a state-action pair $(s,a)$, that is, with $\min_{s,a}\pi(a|s)$?
>
> **Response:** The minimum visitation probability appears in our finite-time bound in the form of $D_{\min} := \min_{s,a} \mu(s)\pi(a|s)$ (see the definitions of the constants $C_1$ and $C_2$ in Theorem 3.1), where $\mu$ is the stationary distribution of the Markov chain {$S_k$} induced by the behavior policy $\pi$.  The parameter $D_{\min}$ appears inversely and quadratically in the sample complexity bound stated in Corollary 3.1.1.
>
> Assuming a fixed behavior policy that induces a uniformly ergodic Markov chain is common even in existing studies of discounted Q-learning [45]. In contrast, Q-learning with a learned (and potentially time-varying) policy is often studied in the context of regret; see [3] and the work of Jin et al., "Is Q-learning provably efficient?" Advances in Neural Information Processing Systems 31 (2018), which differs from the focus of this work—namely, establishing last-iterate convergence rates measured in the span seminorm.

---

> > ### Author Response · Authors · 2025-08-04
> > **Response to Reviewer Comments**
> >
> > We thank the reviewer for the valuable comments. We hope our response has successfully addressed the reviewer’s concerns and answered their questions. If there are any remaining issues, we would be happy to continue the discussion. Otherwise, we would greatly appreciate the reviewer’s consideration in updating their score in light of our clarifications.

---

> > ### Comment · Reviewer_pbot · 2025-08-05
> >
> > I would like to thank the authors for their clarifications. I am not an expert in the average-reward RL setting, and appreciate the detailed response. I think the paper will greatly benefit from including the outlined discussion (especially about assumption 2.1 and challenges related with the geometric ergodicity and multi-step contraction) to the paper.
> >
> > Regarding the numerical experiments, I do not insist on including them, as this is a purely theoretical paper. I think that including a toy experiment will not add much to the value of the paper. The main purpose of my comment was to question the practical applicability of adaptive step-size versions of Q-learning.
> >
> > That being said, I am happy to increase my score and wish the authors good luck.

---

> > > ### Author Response · Authors · 2025-08-05
> > > **Acknowledgment of Reviewer Feedback**
> > >
> > > We thank the reviewer for the thoughtful response and kind words. We’re glad the clarifications were helpful and appreciate the reviewer’s perspective on the experiments. We’ll incorporate the suggested discussion in the revision.

---

### Official Review · Reviewer_vKC5 · 2025-07-01

**Clarity:** 4
**Significance:** 4
**Originality:** 3
**Rating:** 6
**Confidence:** 3

**Summary:**

The main theorem is the first to establish nonasymptotic convergence guarantees for the classical asynchronous Q-learning algorithm for average-reward RL. The algorithm deviates from usual Q-learning by using adaptive stepsizes, and the paper also presents formal arguments for why they are necessary for the process to converge to the correct limit, as well as demonstrating that they can be understood via a connection to importance sampling. Their use introduces the main technical challenge of the paper (by causing the underlying stochastic process to be non-Markovian).

**Questions:**

What can be said about the distance between the two sets of fixed points described in Proposition 4.1? (For instance with a universal stepsize schedule such as $ \alpha_k(s,a) = c/k $)?

Could the authors provide any commentary on aspects of the proof of the main theorem which are suboptimal (in terms of leading to a suboptimal sample complexity)?

Could the theorem statement be refined to provide a guarantee which depends on the fixed-point residual of the initialization $ Q_1 $ or its distance from a fixed-point ? (Would such a term replace some factors of $ 1/(1-\beta)$ ?)

**Ethical Concerns:**

["NO or VERY MINOR ethics concerns only"]

**Final Justification:**

Overall the paper has many strengths (see above) and makes an important contribution to a fundamental problem. The rebuttal also satisfactorily addresses my main outstanding questions regarding Proposition 4.1.

**Limitations:**

yes

**Quality:**

4

**Strengths And Weaknesses:**

Strengths: The paper is very well-written and clear. The main result is a fundamental contribution with great significance, and also is the consequence of overcoming multiple conceptual and technical obstacles. At a high level, I think it is great that the paper attempts to study whether adaptive stepsizes are necessary, and I imagine that the related result and discussion may have even greater significance than the convergence analysis of Theorem 3.1. The connection to importance sampling is similarly an interesting and significant contribution. I also appreciate the comparison with discounted Q-learning.

Weaknesses:
Assumption 2.1 is a strong assumption which is not satisfied in many of the more general classes of MDPs and is crucial to the convergence guarantees of the algorithm.

While Proposition 4.1 states that the fixed points of the synchronous and asynchronous fixed-point equations are disjoint, to completely justify the necessity of adaptive stepsizes it seems like we would instead need to understand the distance between the sets of fixed points? In principle if the distance is small (relative to the number of iterations $ k $) then it seems this would not be a fatal issue.

(As discussed after Corollary 3.1.1) the sample complexity analysis appears suboptimal. Specifically, I do not view it as a weakness that the upper bound does not match the minimax lower bound (for a synchronous setting), since this would not be expected, both due to the more difficult setting and the lack of variance-reducing techniques. It is very worthwhile to understand the complexity of Q-learning despite its algorithmic suboptimality. However, it additionally seems that the analysis is very likely suboptimal. It might be interesting to have additional discussion of suboptimal aspects of the proof structure.

While the algorithm accepts an arbitrary initialization $ Q_1$, the guarantees do not seem to depend on the distance between $ Q_1$ and some fixed point.

---

> ### Author Rebuttal · Authors · 2025-07-28
>
> We thank the reviewer for the encouraging comments regarding the significance of the contributions of this work. In the following, we provide a point-by-point response to all of the reviewer's comments.
>
> >Comment: Assumption 2.1 is a strong assumption that is not satisfied in many of the more general classes of MDPs and is crucial to the convergence guarantees of the algorithm.
>
> **Response:** We agree with the reviewer that Assumption 2.1 is a strong assumption. The next step in this line of work is to remove this assumption by considering stochastic iterative algorithms under non-expansive operators, which would allow us to relax it to weakly communicating MDPs. That said, as acknowledged by the reviewer, this paper has already overcome several major conceptual and technical challenges—such as identifying the necessity of using adaptive stepsizes, leveraging adaptive stepsizes as a means of implicit importance sampling, and handling non-Markovian noise—to achieve the main result. Relaxing Assumption 2.1 is an interesting future direction, but it is beyond the scope of this work.
>
> > Question: What about the distance between the two sets of fixed points described in Proposition 4.1? (For instance, with a universal stepsize schedule such as $\alpha_k=c/k$)?
>
> **Response:** This is an excellent question. Before going into the details, we would like to clarify that, regardless of the number of iterations we run, Q-learning with universal (respectively, adaptive) stepsizes can only converge to the set of solutions to the asynchronous Bellman equation $p_{span}(\bar{\mathcal{H}}(Q) - Q) = 0$ (respectively, the original Bellman equation $p_{span}(\mathcal{H}(Q) - Q) = 0$). Recall that the sets of solutions to $p_{span}(\bar{\mathcal{H}}(Q) - Q) = 0$ and $p_{span}(\mathcal{H}(Q) - Q) = 0$ are of the forms $\bar{Q}^\star + ${$ce \mid c \in \mathbb{R}$} and $Q^\star + ${$ce \mid c \in \mathbb{R}$}, which are identical when $D = I / (|\mathcal{S}||\mathcal{A}|)$ (cf. Proposition 4.1). Therefore, to conduct the sensitivity analysis, it is enough to bound $p_{span}(\bar{Q}^\star - Q^\star)$ as a function of $D$.
>
> Since the asynchronous Bellman operator $\bar{\mathcal{H}}(\cdot)$ is defined as $\bar{\mathcal{H}}(Q) = (I-D)Q + D\mathcal{H}(Q)$ (see Eq. (5) of the submission), if $p_{span}(\bar{\mathcal{H}}(\bar{Q}^\star) - \bar{Q}^\star) = 0$, there must exist some $c \in \mathbb{R}$ such that
>
> $\bar{\mathcal{H}}(\bar{Q}^\star) - \bar{Q}^\star=D(\mathcal{H}(\bar{Q}^\star) - \bar{Q}^\star) = ce \quad \Leftrightarrow \quad \mathcal{H}(\bar{Q}^\star) - \bar{Q}^\star = c D^{-1} e$,
>
> where $e$ is the all-ones vector. On the other hand, we know that
>
> $\mathcal{H}(Q^\star) - Q^\star = r^\star e$.
>
> Taking the difference of the above two equations, we obtain
>
> $\bar{Q}^\star - Q^\star = \mathcal{H}(\bar{Q}^\star) - \mathcal{H}(Q^\star) + r^\star e - c D^{-1} e$.
>
> Taking the span seminorm on both sides and using the facts that $\mathcal{H}(\cdot)$ is a $\beta$-contraction with respect to $p_{span}(\cdot)$ and $\text{ker}(p_{span}) = ${$ce \mid c \in \mathbb{R}$}, we conclude that
>
> $p_{span}(\bar{Q}^\star - Q^\star) \leq \frac{|c| \cdot p_{span}(D^{-1} e)}{1 - \beta}$.
>
> To interpret the result, consider the special case where $D = I / (|\mathcal{S}||\mathcal{A}|)$, in which case the right-hand side vanishes and we conclude that $p_{span}(\bar{Q}^\star - Q^\star) = 0$, implying that the solution sets to $p_{span}(\bar{\mathcal{H}}(Q) - Q) = 0$ and $p_{span}(\mathcal{H}(Q) - Q) = 0$ are the same. This agrees with Proposition 4.1. More generally, when $D \neq I / (|\mathcal{S}||\mathcal{A}|)$, the above inequality provides a sensitivity analysis. We will add the above result to the next version of this work.
>
> As a follow-up to this comment, we note that the ultimate goal of reinforcement learning is not to find $Q^\star$ per se, but to identify an optimal policy $\pi^\star$. Therefore, even if $p_{span}(\bar{Q}^\star - Q^\star) \neq 0$, Q-learning with universal stepsizes could still be acceptable if the greedy policy induced by $\bar{Q}^\star$ is optimal (or close to optimal). Unfortunately, we have constructed a two-state, two-action MDP that demonstrates this is, in general, not the case. Due to space limitations, we are unable to include the example here, but we will add it in the next version of the paper (or during the discussion phase, if the reviewer is particularly interested).
>
> >Question: Could the authors provide any commentary on aspects of the proof of the main theorem that are suboptimal (in terms of leading to a suboptimal sample complexity)?
>
> **Response:** This is a very good question. We agree with the reviewer that the sub-optimality arises from both algorithmic design choices (e.g., without using variance reduction techniques) and artifacts of our proof. Regarding potential looseness in the analysis, we identify two main sources.
>
> *The almost sure bound on $p_{span}(Q_k):$*
> As highlighted in Sections 1, 3, and 5, a major technical challenge in the analysis is handling the non-Markovian noise introduced by the use of adaptive stepsizes. In Section 5.1, we reformulate the algorithm as a Markovian stochastic approximation by incorporating the empirical frequency matrix $D_k$ into the definition of the Markov chain. However, the resulting chain {$Z_k$} lacks geometric mixing, which is typically used in the literature to handle Markovian noise—i.e., to break the correlation between the stochastic iterates $Q_k$ and the Markov noise $Z_k = (D_k, S_k, A_k, S_{k+1})$.
>
> To address this issue, our approach—detailed in Section 5.3—first establishes that while the iterates $Q_k$ do not admit a uniform bound as in the discounted setting, they do admit a time-varying almost-sure bound that grows at most logarithmically in $k$ (cf. Proposition B.1). The proof of Proposition B.1 is nontrivial: it involves an induction argument to bound $p_{span}(Q_{k+1})$ in terms of $p_{span}(Q_k)$ and a combinatorial argument to identify the worst-case scheduling of adaptive stepsizes that leads to the maximal upper bound.
>
> Unfortunately, this time-varying bound (denoted $b_k$) stated in Proposition B.1 scales as $b_k = \mathcal{O}(|\mathcal{S}||\mathcal{A}|\log k)$. Since $b_k$ appears quadratically in the finite-time analysis, this introduces an additional $\mathcal{O}(|\mathcal{S}|^2|\mathcal{A}|^2)$ dependence in the sample complexity. Intuitively, the almost-sure time-varying bound on $Q_k$ should not exhibit any $|\mathcal{S}||\mathcal{A}|$ dependence, as we use $p_{span}(\cdot)$ as the measure. Note that $p_{span}(\cdot)$ can be viewed as a projection with respect to the $||\cdot||_\infty$ norm (cf. Lemma 2.1), which does not depend on the dimension of the vector. However, rigorously proving that $b_k$ is independent of $|\mathcal{S}||\mathcal{A}|$—that is, refining the analysis in Proposition B.1—appears to be significantly more challenging due to the use of adaptive (stochastic) stepsizes.
>
> *The variance analysis:* To further mitigate the dependence on the size of the state-action space and $1/(1-\beta)$, we note that the existing literature has employed advanced concentration inequalities—such as Freedman’s inequality [45] and Bernstein's inequality [46]—to refine the analysis of the indicator function of the Markov chain {$(S_k, A_k)$} in asynchronous Q-learning. In contrast, our Lyapunov-based technique for mean-square analysis does not exploit such techniques.
>
> >Question: Could the theorem statement be refined to provide a guarantee that depends on the fixed-point residual of the initialization $Q_1$ or its distance from a fixed-point? (Would such a term replace some factors of $1/(1-\beta)$?)
>
> **Response:** The reviewer is absolutely correct. To clarify, our finite-time bound is different for $k \leq K$ and $k \geq K$, where $K$ is a threshold that roughly captures the mixing of the Markov chain {$(S_k, A_k)$} induced by the behavior policy $\pi$. For $k \leq K$, we bound $p_{span}(Q_k - Q^\star)$ as
>
> $p_{span}(Q_k - Q^\star)\leq p_{span}(Q_k) + p_{span}( Q^\star )\leq b_k+p_{span}(Q^\star)=m_k$,
>
> where $b_k$ is the time-varying almost-sure bound of $p_{span}(Q_k)$ established in Proposition B.1, and $m_k$ is defined right before Theorem 3.1. Note that in the proof of Proposition B.1, we assume without loss of generality that $p_{span}(Q_1) = 0$, which will be made explicit in the next version of this work. This can be easily achieved by initializing $Q_1$ as a constant vector.
>
> As for $k \geq K$ (which is the more interesting case), the bound naturally depends on the "initial error" $p_{span}(Q_K - Q^\star)$. However, because $Q_K$ is inherently random, to maintain the mathematical rigor of our result, we apply the following inequality to ensure that the bound does not depend on random quantities:
>
> $p_{span}(Q_K - Q^\star) \leq p_{span}(Q_K) + p_{span}(Q^\star) \leq b_K + p_{span}(Q^\star) = m_K$,
>
> which is the term that multiplies the leading factor in all cases of Theorem 3.1. Moreover, to make the sample complexity bound instance-independent, we further upper bound $p_{span}(Q^*)$ by $1/(1 - \beta)$ in Corollary 3.1.1 when computing the sample complexity.
>
> To summarize, if one accepts having $p_{span}(Q_K - Q^\star)$ in the finite-time bound, one can simply replace $m_K$ by $\mathbb{E}[p_{span}(Q_K - Q^\star)^2]$ in Theorem 3.1 so that the theorem captures the dependence on the "initial error" $Q_K - Q^\star$. Moreover, if one also wants explicit dependence on $p_{span}(Q^\star)$ in the sample complexity, the result is
>
> $
> \mathcal{O}(|\mathcal{S}|^3 |\mathcal{A}|^3 D_{min}^{-2} \epsilon^{-2} (1 - \beta)^{-3} p_{span}(Q^*)^2).
> $
>
> See also the proof of Corollary 3.1.1 in Appendix B.5.

---

> > ### Comment · Reviewer_vKC5 · 2025-08-04
> >
> > I particularly appreciate the authors' comments regarding the solutions to the two Bellman equations, and I believe the paper will benefit from these additions. Overall I am satisfied with the comments provided by the authors and will maintain my score.

---

> > > ### Author Response · Authors · 2025-08-04
> > > **Acknowledgment of Reviewer Feedback**
> > >
> > > We thank the reviewer for appreciating the contributions of this work and for the valuable comments—especially those that led to the sensitivity analysis—which helped improve the paper.

---

### Official Review · Reviewer_i7gX · 2025-07-02

**Clarity:** 4
**Significance:** 3
**Originality:** 3
**Rating:** 5
**Confidence:** 4

**Summary:**

This paper aims to solve the infinite-horizon, undiscounted MDP problem by proposing a provably optimal asynchronous updating Q-learning algorithm. The algorithm adapts the RVI Q-learning [Abounadi et al., 2001] by using an adaptive stepsize that is related to the history trajectory per state-action pair. In measuring the optimality of a policy in the infinite-horizon, undiscounted MDP environment, the authors reformulate the Bellman equation by adapting it to the Seminorm Bellman equation with a seminorm $p_{\text{span}}$. Under the new proposed seminorm, the optimality Q-function $Q^\star$ should satisfy the equation $p_{\text{span}}(\mathcal{H}(Q^\star) - Q^\star) = 0$, where $\mathcal{H}(\cdot)$ is a Bellman operator. Solving this equation is equivalent to finding a fixed point/set of fixed points in the space measured by $p_{\text{span}}$.
The adaptive stepsize breaks the Markovian property of the trajectories and makes the standard analysis technique inapplicable.
The authors overcome this difficulty and derive the theoretical regret guarantee in the finite-time regime.
The authors also present the necessity of introducing an adaptive stepsize by showing that using a universal stepsize will lead the outcome to converge to the wrong target.

**Questions:**

1. For Proposition 4.1, it claims that without choosing a proper $D$, the set of solutions to the asynchronous Bellman equation is disjoint from the solutions to the original Bellman equation measured in the seminorm $p_{\text{span}}$. How to interpret the condition to ensure the two solution sets are identical: $D(s, a) = I/(|\mathcal{S}| |\mathcal{A} |)$? $D$ is supposed to be a diagonal matrix with diagonal entries $\{ \mu(s)\pi(a \mid s) \}_{(s, a) \in \mathcal{S} \times \mathcal{A}}$, does that mean that policy $\pi$ is a deterministic policy?

2. Another question about Proposition 4.1 is that although the sets of solutions are disjoint with some choice of $D$, are those two sets close to each other?

**Ethical Concerns:**

["NO or VERY MINOR ethics concerns only"]

**Final Justification:**

The author has resolved my questions during the rebuttal. I will keep my score and support this paper.

**Limitations:**

yes

**Paper Formatting Concerns:**

No obvious formatting concerns were found in this paper.

**Quality:**

3

**Strengths And Weaknesses:**

Strengths:

1. The paper is well-written with rigorous theory analysis and clear notations.

2. This is the first work to analyze an asynchronous updating algorithm in the setting of an average reward infinite-horizon undiscounted MDP environment.
The introduction of adaptive stepsizes breaks the Markovian property of the trajectories, making the algorithm a non-Markovian stochastic approximation scheme. The authors propose innovative techniques to address this challenge by incorporating the empirical frequencies of state-action pairs in the trajectories of the stochastic process.

3. For completeness, the authors justify the necessity of the adaptive stepsize by showing that using a universal stepsize like harmonic stepsize $\alpha_k \coloneqq \alpha / (k + h)$.

Weakness:

1. The analysis relies on Assumption 2.1, which states that the Bellman operator is a $\beta$-contraction mapping with respect to the span seminorm. While a sufficient condition is provided, the paper notes that analyzing Q-learning without such an assumption (i.e., under non-expansive operators) is an important direction for future research

2. While the paper acknowledges $h$ as a "tunable parameter" in the adaptive stepsize formula, it does not offer practical guidelines or methods for its selection. This leaves a gap for practitioners trying to implement and optimize the algorithm.

---

> ### Author Rebuttal · Authors · 2025-07-28
>
> We thank the reviewer for the encouraging comments regarding the contributions of this work. In the following, we provide a point-by-point response to all of the reviewer's comments.
>
> >Comment: The analysis relies on Assumption 2.1, which states that the Bellman operator is a contraction mapping with respect to the span seminorm. While a sufficient condition is provided, the paper notes that analyzing Q-learning without such an assumption (i.e., under non-expansive operators) is an important direction for future research.
>
> **Response:** We agree with the reviewer that Assumption 2.1 is a strong assumption. Removing this assumption would require completely abandoning the contractive viewpoint of the Bellman operator $\mathcal{H}(\cdot)$ and instead studying Q-learning as a stochastic approximation with a non-expansive operator, which is always true for $\mathcal{H}(\cdot)$. That said, as acknowledged by the reviewer, this paper has already overcome several major conceptual and technical challenges—such as identifying the necessity of using adaptive stepsizes, leveraging adaptive stepsizes as a means of implicit importance sampling, and handling non-Markovian noise—to achieve the main result. Relaxing Assumption 2.1 is an interesting future direction, but it is beyond the scope of this work.
>
> >Comment: While the paper acknowledges $h$ as a "tunable parameter" in the adaptive stepsize formula, it does not offer practical guidelines or methods for its selection. This leaves a gap for practitioners trying to implement and optimize the algorithm.
>
> **Response:** We first clarify the role of $h$ in the stepsize $\alpha_k(s,a) = \alpha / (N_k(s,a) + h)$. In the $Q$-learning algorithm we study, to achieve the optimal $1/k$ rate of convergence (see Theorem 3.1, third case), the constant $\alpha$ must exceed a certain threshold (specifically, $\alpha \geq 2 / (1 - \beta)$). The parameter $h > 0$ is introduced solely to ensure that $\alpha_k(s,a) \in (0,1)$ for all $(s,a)$. This phenomenon has also been observed in other stochastic iterative algorithms that do not use adaptive stepsizes. For example, in discounted $Q$-learning [44] (where it is referred to as rescaled linear stepsizes) and in stochastic gradient descent (SGD) for smooth and strongly convex objectives, the optimal convergence rate is achieved with $\alpha_k = \alpha / (k + h)$, where $\alpha$ must be above a threshold that depends on the condition number. The parameter $h$ is again introduced to ensure that $\alpha_k \in (0,1)$ for all $k$.
>
> In the next version of this work, we will add numerical simulations and discuss the choice of $h$ in that context.
>
> >Question: For Proposition 4.1, it claims that without choosing a proper $D$, the set of solutions to the asynchronous Bellman equation is disjoint from the solutions to the original Bellman equation measured in the seminorm $p_{span}$. How to interpret the condition to ensure the two solution sets are identical: $D=I/(|S||A|)$? $D$ is supposed to be a diagonal matrix with diagonal entries {$\mu(s)\pi(a|s)$}$_{(s,a)\in\mathcal{S}\times \mathcal{A}}$, does that mean that policy $\pi$ is a deterministic policy?
>
> **Response:** To clarify, $D$ is the $|\mathcal{S}||\mathcal{A}| \times |\mathcal{S}||\mathcal{A}|$ diagonal matrix whose entries correspond to the stationary distribution of the Markov chain {$(S_k, A_k)$} induced by the behavior policy $\pi$, where $|\mathcal{S}|$ and $|\mathcal{A}|$ denote the cardinalities of the state and action spaces, respectively. Proposition 4.1 states that a necessary and sufficient condition for $Q$-learning with a universal stepsize to converge to the correct target is that $D = I / (|\mathcal{S}||\mathcal{A}|)$; that is, $\mu(s)\pi(a|s) = 1 / (|\mathcal{S}||\mathcal{A}|)$ for all $(s,a)$. This condition is equivalent to saying that the behavior policy $\pi$ must be uniform over actions for every state, i.e., $\pi(a|s) = 1/|\mathcal{A}|$ for all $(s,a)$, and the Markov chain {$S_k$} induced by $\pi$ must also have a uniform stationary distribution over the state space, i.e., $\mu(s) = 1/|\mathcal{S}|$ for all $s$.
>
> However, the stationary distribution (i.e., the matrix $D$) is jointly determined by the behavior policy $\pi$ and the unknown environment transition dynamics, and thus cannot be freely chosen to be $D = I / (|\mathcal{S}||\mathcal{A}|)$. This is why $Q$-learning with a universal stepsize is, in general, not guaranteed to converge to the correct target. This limitation motivates the use of importance sampling with the uniform distribution as the target distribution, leading to the development of $Q$-learning with adaptive stepsizes, as illustrated in Section 4.2.
>
> >Question: Another question about Proposition 4.1 is that, although the sets of solutions are disjoint with some choice of $D$, are those two sets close to each other?
>
> **Response:** This is an excellent question. Recall that the sets of solutions to the asynchonous Bellman equation $p_{span}(\bar{\mathcal{H}}(Q) - Q) = 0$ and the original Bellman equation $p_{span}(\mathcal{H}(Q) - Q) = 0$ are of the forms $\bar{Q}^\star + ${$ce \mid c \in \mathbb{R}$} and $Q^\star + ${$ce \mid c \in \mathbb{R}$}, which are identical when $D = I / (|\mathcal{S}||\mathcal{A}|)$ (cf. Proposition 4.1). Therefore, to conduct the sensitivity analysis, it is enough to bound $p_{span}(\bar{Q}^\star - Q^\star)$ as a function of $D$.
>
> Since the asynchronous Bellman operator $\bar{\mathcal{H}}(\cdot)$ is defined as $\bar{\mathcal{H}}(Q) = (I-D)Q + D\mathcal{H}(Q) $ (see Eq. (5) of the submission), if $p_{span}(\bar{\mathcal{H}}(\bar{Q}^\star) - \bar{Q}^\star) = 0$, there must exist some $c \in \mathbb{R}$ such that
>
> $\bar{\mathcal{H}}(\bar{Q}^\star) - \bar{Q}^\star=D(\mathcal{H}(\bar{Q}^\star) - \bar{Q}^\star) = ce \quad \Leftrightarrow \quad \mathcal{H}(\bar{Q}^\star) - \bar{Q}^\star = c D^{-1} e$,
>
> where $e$ is the all-ones vector. On the other hand, we know that
>
> $\mathcal{H}(Q^\star) - Q^\star = r^\star e$.
>
> Taking the difference of the above two equations, we obtain
>
> $\bar{Q}^\star - Q^\star = \mathcal{H}(\bar{Q}^\star) - \mathcal{H}(Q^\star) + r^\star e - c D^{-1} e$.
>
> Taking the span seminorm on both sides and using the facts that $\mathcal{H}(\cdot)$ is a $\beta$-contraction with respect to $p_{span}(\cdot)$ and $\text{ker}(p_{span}) = ${$ce \mid c \in \mathbb{R}$}, we conclude that
>
> $p_{span}(\bar{Q}^\star - Q^\star) \leq \frac{|c| \cdot p_{span}(D^{-1} e)}{1 - \beta}$.
>
> To interpret the result, consider the special case where $D = I / (|\mathcal{S}||\mathcal{A}|)$, in which case the right-hand side vanishes and we conclude that $p_{span}(\bar{Q}^\star - Q^\star) = 0$, implying that the solution sets to $p_{span}(\bar{\mathcal{H}}(Q) - Q) = 0$ and $p_{span}(\mathcal{H}(Q) - Q) = 0$ are the same. This agrees with Proposition 4.1. More generally, when $D \neq I / (|\mathcal{S}||\mathcal{A}|)$, the above inequality provides a sensitivity analysis. We will add the above result to the next version of this work.
>
> As a follow-up to this comment, we note that the ultimate goal of reinforcement learning is not to find $Q^\star$ per se, but to identify an optimal policy $\pi^\star$. Therefore, even if $p_{span}(\bar{Q}^\star - Q^\star) \neq 0$, Q-learning with universal stepsizes could still be acceptable if the greedy policy induced by $\bar{Q}^\star$ is optimal (or close to optimal). Unfortunately, we have constructed a two-state, two-action MDP that demonstrates this is, in general, not the case. Due to space limitations, we are unable to include the example here, but we will add it in the next version of the paper (or during the discussion phase, if the reviewer is particularly interested).

---

> > ### Comment · Reviewer_i7gX · 2025-08-04
> >
> > I thank the authors detailed feedback, especially the discussion on the last question. I will keep my position to support this paper and maintain my score.

---

### Author Response · Authors · 2025-08-07
**Appreciation and Summary of Contributions and Vision**

We would like to sincerely thank all the reviewers for their thoughtful feedback, detailed engagement, and positive assessments of our paper, as well as the area chair for overseeing the review process. Based on the reviews and ensuing discussions, we would like to respectfully emphasize several points that, in our view, highlight the significance of this work and its potential impact on the community.

**Key Contributions:**

*This paper presents the first finite-time analysis of average-reward Q-learning with asynchronous updates, addressing a foundational problem in reinforcement learning*. Importantly, we show that adaptive stepsizes are not only sufficient but also necessary: without them, the algorithm fails to converge to the correct target. This insight, along with the connection to implicit importance sampling, is of both theoretical and conceptual significance. Thanks to several reviewers’ suggestions, we also provided a sensitivity analysis comparing the limit points of Q-learning with adaptive versus universal stepsizes, which further strengthens the paper’s contribution to the broader understanding of learning dynamics in average-reward settings.

From a technical perspective, this work tackles a major challenge: non-Markovian stochastic approximation arising from adaptive stepsizes that depend on the full sample history. To address this, we introduce a novel framework that reformulates the algorithm as a time-inhomogeneous Markovian stochastic approximation scheme, and we develop a set of tools—a combination of almost-sure time-varying bounds, conditioning arguments, and Markov chain concentration inequalities—to break the strong correlations between the iterates and the noise. Several reviewers explicitly acknowledged that our extension of classical Lyapunov-based methods to such non-Markovian settings appears to be new and broadly applicable to other algorithms with history-dependent updates.

**Future Vision:**

Looking ahead, this work opens the door to several promising future directions—such as analyzing Q-learning with alternative adaptive stepsize schemes; relaxing Assumption 2.1 by studying stochastic approximation with non-expansive operators and non-Markovian noise; and, more broadly, developing a theoretical framework for the non-asymptotic analysis of stochastic approximation algorithms with history-dependent or state-dependent updates. We hope these insights will inspire further exploration into adaptive learning dynamics in reinforcement learning.

---

### Note · Authors · 2025-08-11

We would like to sincerely thank all the reviewers for their thoughtful feedback, detailed engagement, and positive assessments of our paper, as well as the area chair for overseeing the review process. Based on the reviews, we would like to respectfully emphasize several points that, in our view, highlight the significance of this work and its potential impact on the community.

**Key Contributions:**

*This paper presents the first finite-time analysis of average-reward Q-learning with asynchronous updates, addressing a foundational problem in reinforcement learning.* Importantly, we show that adaptive stepsizes are not only sufficient but also necessary: without them, the algorithm fails to converge to the correct target. This insight, along with the connection to implicit importance sampling, is of both theoretical and conceptual significance. Thanks to several reviewers’ suggestions, we also provided a sensitivity analysis comparing the limit points of Q-learning with adaptive versus universal stepsizes, which further strengthens the paper’s contribution to the broader understanding of learning dynamics in average-reward settings.

From a technical perspective, this work tackles a major challenge: non-Markovian stochastic approximation arising from adaptive stepsizes that depend on the full sample history. To address this, we introduce a novel framework that reformulates the algorithm as a time-inhomogeneous Markovian stochastic approximation scheme, and we develop a set of tools—a combination of almost-sure time-varying bounds, conditioning arguments, and Markov chain concentration inequalities—to break the strong correlations between the iterates and the noise. Several reviewers explicitly acknowledged that our extension of classical Lyapunov-based methods to such non-Markovian settings appears to be new and broadly applicable to other algorithms with history-dependent updates.

**Future Vision:**

Looking ahead, this work opens the door to several promising future directions—such as analyzing Q-learning with alternative adaptive stepsize schemes; relaxing Assumption 2.1 by studying stochastic approximation with non-expansive operators and non-Markovian noise; and, more broadly, developing a theoretical framework for the non-asymptotic analysis of stochastic approximation algorithms with history-dependent or state-dependent updates. We hope these insights will inspire further exploration into adaptive learning dynamics in reinforcement learning.

---

### Decision · Program_Chairs · 2025-09-17

**Decision:**

Accept (poster)

**Comment:**

The paper develops finite-time analysis of average-reward Q-learning in the asynchronous setting with adaptive stepsizes. A highlight of the paper is to demonstrate that adaptive stepsizes provably help, and without adaptation, the algorithm might fail, thus providing a more complete understanding. The reviewers are all positive about the paper, and therefore it warrants acceptance. However, it is not recommended for more recognition due to the caveat that 1) the rate depends pessimistically with the size of the state-action space at a polynomial order, which diminishes the value of the faster convergence rate; 2) the faster rate requires a stringent assumption (Assumption 2.1), which eliminates its applicability to many standard MDPs.